# AUTONOMY-AWARE CLUSTERING: WHEN LOCAL DECISIONS SUPERSEDE GLOBAL PRESCRIPTIONS

## ABSTRACT

Clustering arises in a wide range of problem formulations, yet most existing approaches assume that the entities under clustering are passive and strictly conform to their assigned groups. In reality, entities often exhibit local autonomy, overriding prescribed associations in ways not fully captured by feature representations. Such autonomy can substantially reshape clustering outcomes—altering cluster compositions, geometry, and cardinality—with significant downstream effects on inference and decision-making. We introduce autonomy-aware clustering, a reinforcement (RL) learning framework that learns and accounts for the influence of local autonomy without requiring prior knowledge of its form. Our approach integrates RL with a deterministic annealing (DA) procedure, where, to determine underlying clusters, DA naturally promotes exploration in early stages of annealing and transitions to exploitation later. We also show that the annealing procedure exhibits phase transitions that enable design of efficient annealing schedules. To further enhance adaptability, we propose the Adaptive Distance Estimation Network (ADEN), a transformer-based attention model that learns dependencies between entities and cluster representatives within the RL loop, accommodates variable-sized inputs and outputs, and enables knowledge transfer across diverse problem instances. Empirical results show that our framework closely aligns with underlying data dynamics: even without explicit autonomy models, it achieves solutions close (within ∼3–4% gap) to the ground truth (where autonomy is known explicitly), whereas ignoring autonomy leads to substantially larger gaps (∼35–40%).

## 1 INTRODUCTION

Clustering, the task of grouping similar entities, underpins a wide range of applications and methodological pursuits, including computer vision, genomics, matrix factorization, and data mining (Karim et al., 2021; Singh & Singh, 2024; Basiri et al., 2025). This process helps reveal the underlying structure of the data and provides insights that can inform decision-making. Formally, given a set $\mathcal{I}$ of $N$ entities, clustering aims to partition them into $K$ clusters by solving

$$\min_{\{\mu(j|i)\},\{C_j\}} \sum_{i=1}^{N} \rho(i) \sum_{j=1}^{K} \mu(j|i)\,\Delta(i, C_j), \text{ subject to } \sum_{j=1}^{K} \mu(j|i) = 1 \,\forall\, 1 \leq i \leq N, \quad \text{(P1)}$$

where $\rho(i)$ denotes the relative importance of the $i^{\text{th}}$ entity, $C_j$ denotes the $j^{\text{th}}$ cluster, and $\mu(j|i) \in \{0, 1\}$ indicates membership of $i^{\text{th}}$ entity in the $j^{\text{th}}$ cluster $C_j$ ($\mu(j|i) = 1$ if $i \in C_j$, and 0 otherwise). The constraint $\sum_{j=1}^{K} \mu(j|i) = 1$ enforces exclusivity — each entity belongs exactly to one cluster. The cost function $\Delta(i, C_j)$ measures the dissimilarity between entity $i$ and those in cluster $C_j$, typically defined in terms of feature vectors $\mathcal{X} = \{x_i\}_{i=1}^{N}$, where $x_i \in \mathbb{R}^d$ represents the attributes of the entity $i$. The search space of partitions grows combinatorially with $N$ and $K$, making clustering a computationally challenging problem.

Clustering is also closely related to *resource allocation* problems, such as facility location, data quantization, and graph aggregation (Rose, 1998; Xu et al., 2014). In these problems, the objective is to assign $K$ *resources* to the entities in $\mathcal{I}$ so that the resources adequately service the entities. This can be viewed as a special case of the clustering problem. For example, the facility location problem can be formulated as $\min_{\{\mu(j|i)\},\{y_j\}} \sum_{i=1}^{N} \sum_{j=1}^{K} \mu(j|i)\,d(x_i, y_j)$, where $x_i$ and $y_j$ denote

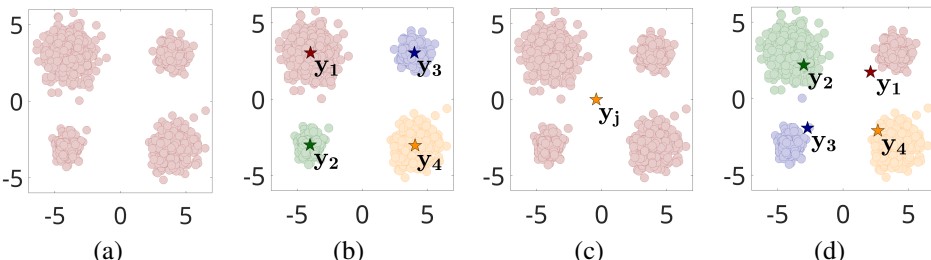

Figure 1: (a) Dataset, (b) No local autonomy - $y_j$'s at cluster centroid, (c) $p(k|j,i) = 0.25$, all $y_j$'s at the centroid of the dataset, and (d) $p(k|j,i) = 0.083$ if $k \neq j$ and $p(k|j,i) = 0.75$, $y_j$'s shifted towards the centroid of the dataset

the feature vectors of the $i^{\text{th}}$ client entity and the $j^{\text{th}}$ resource facility, respectively. Resource allocation problems can thus be interpreted as clustering tasks in which the dissimilarity function takes the form $\Delta(i, C_j) = d(x_i, y_j)$, with each $y_j$ serving as the feature vector of the representative point (or cluster center) of cluster $C_j$. Existing techniques in the literature typically treat the entities as *passive*, meaning that they strictly follow the assignments dictated by the policy $\mu$. However, in many real-world settings, entities exhibit some degree of *autonomy*, allowing them to override the prescribed assignment and behave as *active* rather than passive entities. For instance, in decentralized sensing, distributed sensors (entities) are grouped into clusters, each of which communicates its data to an assigned processing unit (resource). Individual sensors, however, may exercise autonomy and transmit their data to a processing unit associated to a different cluster. Such deviations can result from several factors in the network such as signal interference, congestion at the processing unit, energy constraints, or intentional redundancy (Rusu et al., 2018; Yadav & Ujjwal, 2021).

In this work, we introduce the class of *autonomy-aware clustering* problems, where an entity's cluster membership is determined by two complementary factors: (i) a global assignment policy $\mu(\cdot|i)$, which prescribes the $j^{\text{th}}$ cluster for the $i^{\text{th}}$ entity when $\mu(j|i) = 1$, and (ii) a local autonomy term $p(k|j,i) \in [0,1]$, which probabilistically reassigns the $i^{\text{th}}$ entity to the $k^{\text{th}}$ cluster given the prescription $j \sim \mu(\cdot|i)$. Existing clustering methods can be seen as a degenerate special case where $p(k|j,i) = 1$ if $k = j$ and 0 otherwise, strictly enforcing the policy-prescribed assignment without any autonomy. The local autonomy term $p(k|j,i)$ encodes latent behavioral tendencies of entities that are either not captured at all or only partially reflected in the feature vector $x_i \in \mathbb{R}^d$. For example, in decentralized sensing case, $x_i$ may include attributes such as sensor location, recorded data, and signal strength, but it does not capture network uncertainties such as current battery charge, interference, congestion, or path loss — which are instead reflected through $p(k|j,i)$ (see Appendix A for details). Similarly, in recommender systems, a user's feature vector $x_i$ may represent demographic or historical preferences, yet spontaneous choices, mood, or context-dependent behavior are better captured through the local autonomy. This leads to an important observation: while it is often straightforward to construct feature vectors from available information, quantifying local autonomy is considerably more challenging. In practice, this autonomy — driven by an entity's latent behavior — is rarely known explicitly, and this hidden nature constitutes one of the key challenges for autonomy-aware clustering.

The introduction of local autonomy impacts clustering solutions in multiple ways, including altering cluster assignments, cluster sizes and shapes, and the representative feature vectors of cluster centers $y_j \in \mathbb{R}^d$. For illustration, consider the dataset shown in Figure 1(a). Figures 1(b)-(d) depict clustering solutions for different levels of local autonomy. In Figure 1(b), there is no autonomy, and the cluster centers $y_j$ exactly coincide with the respective cluster centroids. In Figure 1(c), the entities have full autonomy: each entity associates itself to each of the four clusters with equal probability, $p(k|j,i) = 0.25$ for all $i,j,k$. Consequently, all cluster centers $\{y_j\}$ collapse to the centroid of the *entire* dataset. For intermediate levels of autonomy, the cluster centers $y_j$ tend to shift toward the global centroid, increasing accessibility to entities not assigned to them under the policy $\mu$. For example, in Figure 1(d), $p(k|j,i) = 0.75$ if $k = j$ and $p(k|j,i) = 0.083$ otherwise. The algorithm used to generate these solutions is described in Section 3.

Changes in clustering solutions due to local autonomy can significantly affect downstream inference and decision-making. For instance, consider the design of decentralized sensing networks discussed above. Here $y_j$ determine the location of the $j^{\text{th}}$ processing unit (UAV). Due to local conditions such

as congestion at the assigned UAV, intermittent fading or blockage caused by buildings and vehicles, and battery-saving heuristics, a sensor may choose to share its data with the $k^{\text{th}}$ UAV instead of the $j^{\text{th}}$ that could be closest to it. To reduce communication cost $d(x_i, y_k)$ in such instances, it is preferred to have the $k^{\text{th}}$ UAV closer to the sensor location $x_i$. This exactly translates to shift of the cluster centers (or representatives) $y_j$ in Figure 1(d) from their location in Figure 1(a) (see Appendix A for details). Similarly, in recommender systems, cluster centers are used to characterize user preference profiles, and shifts induced by local autonomy may lead to different insights for designing new products. So, ignoring autonomy risks producing misleading conclusions from clustering outcomes.

One of the key contributions of this work is the development of a framework that captures the effect of local autonomy on clustering solutions. The framework is presented in two stages. We first consider the case where the autonomy models are known. Here, we build upon the Maximum Entropy Principle (MEP)–based Deterministic Annealing (DA) algorithm for data clustering (Rose, 1991), which has proven itself to be effective in addressing major challenges such as combinatorial complexity, non-convexity, poor local minima, and sensitivity to initialization. Our reformulation of DA with local autonomy models inherits these advantages. In particular, our modified DA algorithm maintains *soft* assignment distributions $\pi(j|i) \in [0, 1]$, where the annealing parameter $\beta$ controls the entropy (or "softness") of the assignments. Reformulating the problem in terms of these soft distributions yields explicit solutions for $\pi(j|i)$ at each $\beta$, thereby significantly reducing problem complexity. The annealing process, in which these relaxed problems are solved successively as $\beta$ increases, helps avoid shallow minima and reduces sensitivity to initialization. A key feature of DA is its *phase transition* behavior: cluster centers $y_j$ remain stable over ranges of $\beta$ and tend to change significantly only at certain critical values. This property enables efficient annealing schedules in which $\beta$ is increased exponentially.

We then turn to the more practical case where autonomy models are unknown. In this setting, we view clustering as a Markov Decision Process (MDP) with *unit* horizon: the state space consists of the data points $\{x_i\}$ and cluster centers $\{y_j\}$, the action space corresponds to the set of clusters $\{C_j\}$, local autonomy defines the transition probabilities, the instantaneous cost reflects the cluster-association cost, and the policy $\mu(j|i)$ specifies the action at each state $i$. This formulation allows us to leverage reinforcement learning (RL), which can determine cluster assignments without explicit knowledge of the transition probabilities. Accordingly, we propose an RL-based framework that jointly *learns* both the assignment policy $\mu$ and the cluster representatives $y_j$, effectively tracking the solutions from the known-model case. We develop algorithms for both scenarios: when the local autonomy model $p(k|j, i)$ is independent of the decision variables $y_j$, and when it depends on them. A further contribution of this work is the Adaptive Distance Estimation Network (ADEN), an attention-based deep model built on a transformer backbone (Vaswani et al., 2017). ADEN enables model-free learning by leveraging the attention mechanism to capture dependencies between entity properties $x_i$ and cluster representatives $y_j$ — dependencies that subsequently determine the assignment policy. Its flexibility in handling inputs and outputs of varying sizes facilitates knowledge transfer across diverse problem instances. Crucially, ADEN takes in the entire set of cluster representatives as input, which is essential to address the scenario where the local autonomy distribution $p(k|j, i)$ depends on all $y_j$. Such global dependencies are common in applications such as decentralized sensing and recommender systems. The proposed ADEN architecture also allows exploiting hardware parallelism for large-scale datasets.

Empirical evaluations were conducted on a suite of synthetic scenarios where autonomy distributions were governed by scenario parameters, on a decentralized sensing application using the UDT19 London Traffic dataset (Loder et al., 2019), where the problem is posed as optimal UAV placement to maximize coverage of roadside sensors, and on a movie recommendation task using MovieLens dataset (Harper & Konstan, 2015). These studies show that our framework produces solutions that closely reflect the underlying data dynamics: even without explicit autonomy models, the performance of our method on average remains within $\sim 3 - 4\%$ of model-based solutions (ground truth — where autonomy is known explicitly). Interestingly, on some instances of the large-scale decentralized sensing problem, our proposed learning-based algorithm achieves up to a $10\%$ improvement over the case where the local autonomy model is explicitly known, underscoring its inherent capacity to escape poor minima. Note that, the reinforcement learning foundation of our framework equips it with the ability to operate in an online manner, where solutions are not only computed once but can be progressively improved as more information becomes available.

## 2  RELATED WORK

Probabilistic model-based clustering has been widely studied in classical machine learning and applied to diverse domains including image segmentation and topic modeling (Deng & Han, 2018). Fundamental approaches include mixture models such as Gaussian Mixture Models (GMM) and Bernoulli Mixture Models (BMM) (McLachlan & Basford, 1988; McLachlan & Peel, 2000; Figueiredo & Jain, 2002; Zhang et al., 2021), as well as algorithms like Expectation-Maximization (EM) (McLachlan & Krishnan, 2008), probabilistic topic models (Hofmann, 2013) and offline/online Deterministic Annealing (Rose, 1998; Mavridis & Baras, 2022). In such frameworks, data points are not deterministically assigned to clusters; instead, they have soft assignments $\pi(j|i) \in [0, 1]$, with $\sum_j \pi(j|i) = 1$ for each data point $i$. These soft assignments capture stochasticity at the *policy level*: the algorithm expresses uncertainty about which cluster representative best explains a data point based solely on its observed features $x_i$. In other words, the probabilistic assignments $\pi(j|i)$ arise because multiple cluster centers fit the feature vector $x_i$ comparably well. However, this type of stochasticity does not account for *local autonomy*—the possibility that a data point may override the prescribed assignment due to latent, unobserved behavioral or contextual factors that are not encoded in $x_i$. Autonomy therefore introduces an additional layer of stochastic behavior at the level of the data point itself, modeled through $p(k|j, i)$, which is distinct from and complementary to the soft assignment probabilities produced by standard probabilistic clustering methods.

A line of theoretical work in (Harris et al., 2019; Brubach et al., 2024; Negahbani & Chakrabarty, 2021) have formulated clustering directly under stochastic assignment policy and developed approximation algorithms with provable guarantees. For example, (Harris et al., 2019) defines a probability distribution, termed as "$k$-lottery," over possible sets of $K$ centers rather than deterministically selecting a fixed set. The users however, are still passive recipients of assignments, highlighting a gap between these approaches and applications where agents may autonomously accept or reject assignments. In parallel, researchers have cast clustering as a reinforcement learning (RL) problem. One of the earliest examples is Reinforcement Clustering (RC) (Likas, 1999) where each data point's (assumed passive) assignment to a cluster is treated as an action and the distortion/error acts as the reward signal. More recent work in (Li et al., 2022; Gowda et al., 2022; Zhu et al., 2025), provide a deep reinforcement learning variant of the approach presented in (Likas, 1999).

This line of work also connects to clustering in environments where human behavior introduces uncertainty. For instance, (Banerjee & Veltri, 2024; Ji et al., 2023) highlight how policies assign individuals to behavioral "clusters," yet real-world deviations due to human unpredictability and information asymmetry necessitate probabilistic post-adjustments. Similarly, *The Ethical Algorithm* (Kearns & Roth, 2019) discusses fairness-aware clustering and allocation under uncertainty, underscoring challenges when algorithmic groupings diverge from intended impact.

Unlike prior works, our framework allows stochasticity not only in the cluster-assignment policy but also in the behavior of the entities themselves conditioned on the prescribed cluster. In fact, we explicitly accounts for the latter in the underlying optimization problem that we pose. Building on the DA framework-where the annealing parameter governs the softness of the assignment policy—we introduce a notion of local autonomy: for each entity, a probability distribution governs the realized action conditioned on the assigned policy action. To our knowledge, incorporating such entity-level autonomy into clustering is novel and opens a promising direction for applications where policy adaptation to locally stochastic behavior is essential.

## 3  PROBLEM FORMULATION AND SOLUTION METHODOLOGY

In this article, we modify the classical clustering (resource allocation) problem (P1) to include local autonomy. For ease of reference, a consolidated summary of all relevant notations is provided in Appendix L. Let $x_i \in \mathcal{X} \subseteq \mathbb{R}^d$ denote the feature vector of the $i$-th entity, with relative weight $\rho(i)$ such that $\sum_{i=1}^N \rho(i) = 1$. In the autonomy-aware setting, each entity $i$ may override its prescribed cluster assignment. Specifically, if entity $i$ is assigned to cluster $C_j$, it may instead select cluster $C_k$ with probability $p(k|j, i)$. The objective is to determine a set of representative feature vectors $\mathcal{Y} := \{y_j\}_{j=1}^K$ (or the vector $Y = [y_1^\top \ y_2^\top \ \cdots \ y_K^\top]^\top \in \mathbb{R}^{Kd}$), corresponding to cluster centers, together with binary association variables $\mu(j|i) \in \{0, 1\}$, which indicate the assignment of entity $i$

to cluster $C_j$, such that the cumulative *expected cost* of assignment is minimized:

$$\min_{\{\mu(j|i)\},\{y_j\}} D := \sum_{i=1}^{N} \rho(i) \sum_{j=1}^{K} \mu(j|i) \sum_{k=1}^{K} p(k|j,i)\, d(x_i, y_k), \quad \text{subject to } \mu \in \Lambda, \quad \text{(P2)}$$

where $\Lambda = \{\mu : \mu(j|i) \in \{0,1\} \ \forall i,j, \ \sum_{j=1}^{K} \mu(j|i) = 1 \ \forall i\}$ denotes the set of feasible assignment policies as in (P1). We further define $d_{\mathrm{avg}}(x_i, y_j) := \sum_{k=1}^{K} p(k|j,i)\, d(x_i, y_k)$ as the average cost of associating the $i^{\text{th}}$ entity to the $j^{\text{th}}$ cluster.

To address the autonomy-aware clustering problem, we adapt the *maximum entropy principle (MEP)-based deterministic annealing (DA)* algorithm, originally developed for the classical formulation (P1). Instead of solving Problem (P2) directly, we introduce a family of parameterized problems (P($\beta$)), which are solved sequentially for an increasing sequence of annealing parameters $\{\beta_k\}$. At each stage, the solution of (P($\beta_{k-1}$)) is used to initialize (P($\beta_k$)), and the sequence is constructed such that the limiting solution of (P($\infty$)) provides a high-quality approximation to (P2).

In (P($\beta$)) the binary assignment policy $\mu(j|i) \in \{0,1\}$ is relaxed to $\pi(j|i)$, which takes values in the interval $[0,1]$. This relaxation enables *soft* rather than binary associations, and $\pi(\cdot|i)$ can be interpreted probabilistically as a distribution over cluster assignments for the $i^{\text{th}}$ entity. The parameterized problem is then formulated as

$$\min_{\{\pi(j|i)\},Y} F = D - \tfrac{1}{\beta} H \quad \text{subject to } \pi \in \Lambda_\beta, \quad \text{(P($\beta$))}$$

where $D := \sum_{i=1}^{N} \rho(i) \sum_{j=1}^{K} \pi(j|i) \sum_{k=1}^{K} p(k|j,i)\, d(x_i, y_k), \quad H := -\sum_{i=1}^{N} \rho(i) \sum_{j=1}^{K} \pi(j|i) \log \pi(j|i),$

denote the relaxed distortion term (corresponding to (P2)) and the conditional entropy of the assignment distribution $p(y_j|x_i) = \pi(j|i)$, respectively. The feasible set is defined as $\Lambda_\beta := \{\pi : \pi(j|i) \in [0,1] \ \forall i,j, \ \sum_j \pi(j|i) = 1 \ \forall i\}$, which is the natural relaxation of the feasible set $\Lambda$ in (P1) and (P2).

The entropy term serves two purposes: (i) it reduces sensitivity to initialization, and (ii) it helps avoid poor local minima. When $\beta$ is small, the entropy term dominates, encouraging high-entropy (near-uniform) assignments. In this regime, cluster centers $\{y_j\}$ are estimated using information from the entire dataset, producing more global solutions. As $\beta$ increases, the influence of the distortion term grows, gradually sharpening the assignments towards deterministic clustering. This contrasts with algorithms such as $k$-means, where cluster centers depend only on local memberships and are therefore highly sensitive to initialization.

For each fixed $\beta$, the cost function $F$ is convex with respect to the policy variables $\pi(j|i)$ (although it is not jointly convex in both $\{y_j\}$ and $\{\pi(j|i)\}$). The optimal assignment policy can therefore be obtained in closed form. Specifically, consider the unconstrained Lagrangian $F' = F + \sum_{i=1}^{N} \nu_i \left( \sum_{j=1}^{K} \pi(j|i) - 1 \right)$, where $\nu_i$ are the multipliers enforcing the normalization of $\pi(\cdot|i)$. Setting $\frac{\partial F'}{\partial \pi(j|i)} = 0$ yields the Gibbs distribution:

$$\pi_Y^\beta(j|i) = \operatorname{softmax}_j\big(-\beta\, d_{\mathrm{avg}}(x_i, y_j)\big) = \frac{\exp\{-\beta\, d_{\mathrm{avg}}(x_i, y_j)\}}{\sum_{\ell=1}^{K} \exp\{-\beta\, d_{\mathrm{avg}}(x_i, y_\ell)\}}, \quad (1)$$

where $d_{\mathrm{avg}}(x_i, y_j) := \sum_{k=1}^{K} p(k|j,i)\, d(x_i, y_k)$ represents the average cost of assigning entity $i$ to cluster $j$ under autonomy-aware reassignments. The Gibbs distribution (1) assigns higher probability to clusters with smaller average costs. The parameter $\beta$ acts as an *annealing factor* controlling the sharpness of assignments: when $\beta$ is small, $\pi_Y^\beta(j|i)$ approaches a uniform distribution over clusters (high entropy), encouraging exploration; as $\beta \to \infty$, assignments become increasingly deterministic, converging toward the hard clustering of Problem (P2).

Substituting the Gibbs distribution into (P($\beta$)) eliminates the policy variables and yields the *free energy* $F$ as a function of the cluster representatives $Y = [y_1^\top \ y_2^\top \ \dots \ y_K^\top]^\top \in \mathbb{R}^{Kd}$:

$$\min_Y \ F(Y) = -\frac{1}{\beta} \sum_{i=1}^{N} \rho(i) \log \left( \sum_{j=1}^{K} \exp\big\{ -\beta\, d_{\mathrm{avg}}(x_i, y_j) \big\} \right). \quad (\hat{\mathrm{P}}(\beta))$$

---

**Algorithm 1:** Autonomy-aware clustering - when local autonomy is known explicitly

**Input:** $\beta_{\min}, \beta_{\max}, \tau, K, \{x_i\}_{i=1}^N, \rho(i)$, and $p(k|j,i)$ for all $1 \leq j, k \leq K$ and $1 \leq i \leq N$;

**Output:** Assignment policy $\pi$, and cluster representatives $\{y_\ell\}_{\ell=1}^K$

**Initialize:** $\beta = \beta_{\min}, \pi_Y(j|i) = \frac{1}{K} \; \forall \; i, j$, and $\{y_\ell\}_{\ell=1}^K$ using (2).

**while** $\beta \leq \beta_{\max}$ **do**

     **while** *until convergence* **do**

         Compute $\{\pi_Y(j|i)\}$ in (1), $\{y_\ell\}$ in (2)

     $\beta \leftarrow \tau\beta$; set $y_\ell \leftarrow y_\ell + \epsilon_{\text{noise}}$ (to escape saddle) $\forall \; \ell$

---

The cluster representatives $\{y_\ell\}$ are obtained by minimizing $(\hat{P}(\beta))$, either by solving $\frac{\partial F}{\partial Y} = 0$ or via a descent method (Luenberger et al., 1984). For the commonly used squared Euclidean cost $d(x_i, y_k) = \|x_i - y_k\|_2^2$, the optimality condition yields the update rule:

$$y_\ell = \frac{\sum_{i=1}^N \sum_{j=1}^K \rho(i) \, p(\ell|j,i) \, \pi_Y^\beta(j|i) \, x_i}{\sum_{i=1}^N \sum_{j=1}^K \rho(i) \, p(\ell|j,i) \, \pi_Y^\beta(j|i)}, \quad \forall \, 1 \leq \ell \leq K. \tag{2}$$

Equations (1) and (2) are therefore coupled and must be solved iteratively at each $\beta$. Algorithm 1 summarizes the procedure for computing these solutions under the assumptions that the autonomy probabilities $p(k|j,i)$ are known and independent of $Y$; here the cost function is squared Euclidean (though the approach generalizes to other dissimilarity notions and cases where $p(k|j,i)$ depends on $Y$). In Section (4), we develop a framework for determining clustering solutions when the local autonomy is unknown (and possibly dependent on $Y$) — the more common scenario in practice.

**Remark.** (1) reinforces that annealing promotes insensitivity to initialization: for small $\beta$, $\pi(j|i) \approx 1/K$, producing nearly uniform assignments, while increasing $\beta$ gradually emphasizes the distortion term $D$ in $(P(\beta))$, breaking uniformity. In the limit $\beta \to \infty$, $\pi_Y^\beta$ collapses to hard assignments, recovering the solution of (P2). This annealing induces a homotopy from the convex surrogate $-H$ to the original non-convex objective $D$, a feature of maximum-entropy methods (Rose, 1998; Xu et al., 2014) that helps avoid poor local minima (Srivastava & Salapaka, 2020; 2021).

We analyze Algorithm 1 by separating the *inner-loop* and *outer-loop* convergence. In the inner loop, the coupled equations (1) and (2) can be solved via fixed-point iterations. These iterations can be interpreted as gradient descent steps, which ensures convergence under mild conditions. We formalize this as follows:

**Theorem 1 (Inner-Loop Convergence).** *The fixed-point iteration defined by (1) and (2) is equivalent to gradient descent iteration of the form*

$$Y(t+1) = Y(t) - \frac{1}{2}\big(\hat{P}_{\pi_\rho}^{Y(t)}\big)^{-1} \nabla F(Y(t)), \tag{3}$$

*where $\hat{P}_{\pi_\rho}^{Y(t)} = P_{\pi_\rho}^{Y(t)} \otimes \mathbb{I}_d$, $\mathbb{I}_d$ is the $d \times d$ identity, $\otimes$ is the Kronecker product, and $P_{\pi_\rho}^{Y(t)} \in \mathbb{R}^{K \times K}$ is diagonal with $[P_{\pi_\rho}^{Y(t)}]_{\ell\ell} = p_{\pi_\rho}^{Y(t)}(\ell) := \sum_{i,j} \rho(i) \, \pi_{Y(t)}(j|i) \, p(\ell|j,i)$, representing the effective mass of cluster $\ell$. The iterations (3) converge to a stationary point under the following mild assumptions:*

*(i)* Non-degenerate clusters: *There exists $c > 0$ such that $p_{\pi_\rho}^{Y(t)}(\ell) \geq c$ for all $\ell$; i.e., every cluster has non-zero mass. This is trivially satisfied for $\pi_Y^\beta$ in (1) at $\beta < \infty$.*

*(ii)* No abrupt shift in cluster mass: *Let $Y_r(t+1) = Y(t) - \frac{r}{2}\big(\hat{P}_{\pi_\rho}^{Y(t)}\big)^{-1} \nabla F(Y(t)), \quad r \in (0,1]$ be the relaxed updates. Then the cluster mass change is bounded: $\max_{r \in (0,1]} p_{\pi_\rho}^{Y_r(t+1)}(\ell) < 4\,p_{\pi_\rho}^{Y(t)}(\ell), \; \forall \ell$, i.e., no cluster's mass increases by a factor of $4$ in a single update.*

*If these assumptions do not hold, there exist adaptive step-sizes $\sigma_t$ such that $Y(t+1) = Y(t) - \sigma_t \nabla F(Y(t))$ still converges to a stationary point.*

**Proof:** See Appendix B for details, including a modification of Algorithm 1 with adaptive step sizes that ensures convergence to a stationary point when assumptions are violated.

For the outer-loop ($\beta$) iterations, we highlight an important feature that motivates fast (geometric) annealing schedules. At $\beta \approx 0$, the Lagrangian $F$ is dominated by the convex term $-H$, and the fixed-point iterations in Algorithm 1 converge to a global minimum. As $\beta$ increases, the algorithm tracks the minimizer of $F(Y)$ until reaching a critical value $\beta_{cr}$, where the fixed point ceases to be a (local) minimum. Simulations show that the cluster representatives $y_\ell$ change significantly at $\beta_{cr}$, a phenomenon referred to as a *phase transition*, analogous to annealing processes in statistical physics. Between successive critical points, $\mathcal{Y} = \{y_\ell\}$ remains nearly constant. The following theorems quantify $\beta_{cr}$ and bound the change of $Y$ between phase transitions, enabling the design of efficient annealing schedules in Algorithm 1.

**Theorem 2** (**Phase Transitions**). *The critical value of the annealing parameter at which the fixed point of (1) and (2) is no longer a minimum is*

$$\beta_{cr} = \frac{1}{2\lambda_{\max}\left(\left(\hat{P}^Y_{\pi_\rho}\right)^{-\frac{1}{2}} \Delta \left(\hat{P}^Y_{\pi_\rho}\right)^{-\frac{1}{2}}\right)}, \text{ where } \Delta = \sum_{i=1}^{N}\Big(\sum_{j=1}^{K} P^{ij}_A z_i z_i^\top P^{ij} - \rho(i) P^i z_i z_i^\top P^i\Big) \quad (4)$$

*is a $Kd \times Kd$ matrix, $\lambda_{\max}(\cdot)$ denotes the maximum eigenvalue, $\hat{P}^Y_{\pi_\rho}$ is positive definite as defined in (3), and $P^{ij}_A$, $z_i$, $P^i$ are matrices determined by $\pi_Y$, $Y$, $\mathcal{X}$, and $\{p(k|j,i)\}$.*

**Proof:** See Appendix C.

**Theorem 3** (**Insensitivity in-between phase transitions**). *Let $\beta$ be sufficiently far from a critical value $\beta_{cr}$. Specifically, let $\delta > 0$ satisfy $\lambda_{\min}\left(\mathbb{I}_d - 2\beta\left(P^Y_{\pi_\rho}\right)^{-1/2}\Delta\left(P^Y_{\pi_\rho}\right)^{-1/2}\right) \geq \delta$, where $\lambda_{\min}(\cdot)$ is the minimum eigenvalue. Then, the cluster representatives evolve according to $\left\|\frac{dY}{d\beta}\right\| \leq \frac{N\sqrt{K}R_\Omega}{e\beta\delta}$, where $R_\Omega$ is the diameter of the space $\Omega$ containing $\mathcal{X}$. In particular, the sensitivity of $Y$ to $\beta$ decays as $\mathcal{O}(1/(\beta\delta))$, becoming smaller the farther $\beta$ is from $\beta_{cr}$.*

**Proof:** See Appendix D. The above theorems are stated for the squared Euclidean cost $d(x_i, y_k) = \|x_i - y_k\|_2^2$; similar results may be derived for other distance functions with suitable modifications.

**Annealing Schedule in Algorithm 1:** Theorems 2 and 3 show that significant changes in $Y$ occur only at critical points $\beta_{cr}$, while $Y$ remains nearly constant between successive critical points. This motivates an annealing schedule that steps from one $\beta_{cr}$ to the next. Since exact computation of $\beta_{cr}$ can be expensive, a practical alternative is a geometric schedule $\beta \leftarrow \tau\beta$, $\tau > 1$, which is computationally efficient. While Theorem 3 provides a conservative bound for small $\beta$, simulations indicate that $Y$ changes little for $\beta$ far from $\beta_{cr}$ (see Appendix E for details).

## 4 REINFORCEMENT-BASED METHOD FOR AUTONOMY-AWARE CLUSTERING

We first formalize the unit horizon MDP viewpoint of autonomy-aware clustering, which subsequently motivates an RL-based algorithm to solve it. Consider the MDP $\mathcal{M} = \langle \mathcal{S}, \mathcal{A}, c, p, \gamma \rangle$, where the state space $\mathcal{S}$ consists of data points $\{x_i\}_{i=1}^N$ and the current cluster representatives $\{y_j\}_{j=1}^K$ (i.e., $\mathcal{S} := \{x_1, \ldots, x_N, y_1, \ldots, y_K\}$), the action space $\mathcal{A}$ comprises the clusters $\{C_j\}_{j=1}^K$, the instantaneous cost $c : \mathcal{S} \times \mathcal{A} \times \mathcal{S} \to \mathbb{R}$ is defined as $c(x_i, C_j, y_k) := d(x_i, y_k)$ from (P2), the autonomy model $p(k|j,i)$ plays the role of the transition kernel $p(s'|s,a)$, and since we consider a unit horizon, the discount factor $\gamma = 0$. Let $\bar{\rho}$ denote the initial distribution over the states, such that $\bar{\rho}(x_i) = \rho(i)$ for all $1 \leq i \leq N$, and $\bar{\rho}(y_j) = 0$ for all $1 \leq j \leq K$. Within this formulation, the standard expected value function $J^Y_\pi = \sum_{i=1}^N \rho(i) J^Y_\pi(x_i)$, where

$$J^Y_\pi(x_i) := \mathbb{E}_{p_\pi}\Big[\sum_{t=0}^{\infty} \gamma^t c(s_t, a_t, s_{t+1})\Big|s_0 = x_i\Big], \quad (5)$$

coincides exactly with the clustering objective $D$ in (P2). Under entropy regularization (i.e., replacing instantaneous cost with $c(s_t, a_t, s_{t+1}) + \frac{1}{\beta}\log\pi(a_t|s_t)$), the above average value function coincides exactly with objective $F$ in (P($\beta$)). It is known (Fox et al., 2015; Grau-Moya et al., 2018) that the optimal policy under such regularization is given by

$$\pi^*(a|s) = \frac{\exp\{-\beta Q(s,a)\}}{\sum_{a'\in\mathcal{A}}\exp\{-\beta Q(s,a')\}}, \text{ where } Q(s,a) = \sum_{s'\in\mathcal{S}} p(s'|s,a)\big(c(s,a,s') + \gamma J^Y_{\pi^*}(s')\big). \quad (6)$$

**Algorithm 2:** Deep autonomy-aware clustering algorithm.

---

**Input:** data points $\mathcal{X} = \{x_i\}_{i=1}^N$, number of clusters $K$, annealing parameters $\beta_{\min}, \beta_{\max}, \tau > 1$, number of samples $L$, number of epochs $T_d, T_y$, learning rates $\eta_d, \eta_y$, number of batches $B$, batch size $S$, Exponential Moving Average (EMA) factor $\lambda \in (0, 1)$, perturbation spread $\sigma \ll 1$

**Output:** trained $\text{NN}_\theta$, optimized cluster representatives $Y$, and assignment policy $\pi_\theta$

**Initialize:** $\beta \leftarrow \beta_{\min}; \theta \leftarrow$ Xavier initialization; $Y \leftarrow \frac{1}{N} \sum_i x_i + \mathcal{N}(0, \sigma^2); \bar{d}_0(i, j) = 0 \, \forall \, i, j$

**while** $\beta \leq \beta_{\max}$ **do**

    **for** $t = 1$ **to** $T_d$ **do**

        Sample mini-batches $\{\mathcal{I}_b\}_{b=1}^B, \mathcal{I}_b = \{i_q : 1 \leq q \leq S, i_q \sim \rho\}; \tilde{Y}_b \leftarrow Y + \mathcal{N}(0, \sigma^2)$

        forward pass $\text{NN}_\theta$ to obtain predicted distances $\bar{D}_\theta(X_b, \tilde{Y}_b)$ for all mini-batches

        **for** *each* $i \in \mathcal{I}_b$ *(in parallel)* **do**

            $j \sim \epsilon$-greedy$(\pi_\theta(\cdot \mid i))$ with $\pi_\theta(j|i)$ in (7)

            draw $\hat{L}$ samples $k_\ell \sim p(k|j, i)$, and observe $d(i, k_\ell)$;

            compute the empirical mean $\hat{d}_t(i, j) = \frac{1}{\hat{L}} \sum_{\ell=1}^{\hat{L}} d(i, k_\ell)$

            update the estimate $\bar{d}_t(i, j) \leftarrow \lambda \bar{d}_{t-1}(i, j) + (1 - \lambda) \hat{d}_t(i, j)$; set $\mathcal{M}_b \leftarrow \mathcal{M}_b \cup (i, j)$

        update $\theta$ with one AdamW step on: $L(\theta) = \frac{1}{B} \sum_{b=1}^B \sum_{(i,j) \in \mathcal{M}_b} \left[ \bar{d}_t(i, j) - d_\theta(x_i, \tilde{y}_j) \right]^2$.

    **for** $t = 1$ *to* $T_y$ **do**

        Substitute $d_{\text{avg}}(x_i, y_j)$ in $F(Y)$ in (P̂($\beta$)) with $d_\theta(x_i, y_j)$; perform $Y \leftarrow Y - \eta_y \nabla_Y F(Y)$

    $\beta \leftarrow \tau \beta$

---

Substituting $\gamma = 0$, $s = x_i$, $a = C_j$, and $s' = y_k$, we obtain $Q(x_i, C_j) = \sum_{k=1}^K p(k|j, i)d(x_i, y_k)$, which is exactly the average association cost $d_{\text{avg}}(x_i, C_j)$ (or more precisely, $d_{\text{avg}}(x_i, y_j)$) in (P2) — establishing the $Q$-function analogy of the average association cost. Further, under this observation the policy $\pi^*$ in (6) is exactly the optimal policy $\pi_Y^\beta(j|i)$ in (1). Below we illustrate an RL-based algorithm for autonomy-aware clustering, which in addition to learning the policy $\pi_Y^\beta(j|i)$, also learns the cluster representatives $\{y_j\}$.

Structurally, our proposed RL-based learning algorithm parallels Algorithm 1, with the key difference that in the inner while loop (executed at fixed $\beta$), explicit expressions (or updates) of $\pi_Y^\beta(j|i)$ and $\{y_\ell\}$ are replaced with their learning counterparts. We can distinguish between two learning paradigms for autonomy-aware clustering, each motivated by a different reinforcement learning framework: (C1) when $\mathcal{X}$ contains a tractable number of data points, $d(x_i, y_k)$ is available in closed form, and $p(l|j, i)$ is independent of $y_\ell$; and (C2) when the dataset is large ($N \gg 1$), $d(x_i, y_k)$ is not available in closed form, or the local autonomy depends on $y_\ell$.

In (C1), we can estimate the policy $\pi_Y^\beta$ in (1) by learning $d_{\text{avg}}(x_i, y_j)$ through straightforward $Q$-learning–style stochastic iterative updates (Sutton & Barto, 2018), followed by stochastic gradient descent (SGD) iterations to update the cluster representatives $y_\ell$; see Appendix F for details. Here, we expound on the learning framework for the case (C2), which is more general, and subsumes (C1). In this learning framework, we learn a function approximator $d_\theta(x_i, y_j)$ to estimate the average cost $d_{\text{avg}}(x_i, y_j)$. Note that here the $Q$-learning type tabular method to estimate $d_{\text{avg}}(x_i, y_j)$ would fail to scale ($N \gg 1$), and SGD iterations would not be possible due to either the missing closed form of $d(x_i, y_k)$, or the dependence of $p(k|j, i)$ on $\{y_\ell\}$ (or both) — preventing the computation of stochastic gradients. We learn the function approximator $d_\theta(x_i, y_j)$ for $d_{\text{avg}}(x_i, y_j)$ similar to several deep RL frameworks (Mnih et al., 2015). In particular, we determine $\theta$ such that it minimizes

$$L(\theta) = \mathbb{E}_{\substack{i \sim \rho, j \sim \pi_\theta \\ k \sim p}} \left[ (d(x_i, y_k) - d_\theta(x_i, y_j))^2 \right], \text{ where } \pi_\theta(j|i) = \frac{e^{-\beta d_\theta(x_i, y_j)}}{\sum_{j'=1}^K e^{-\beta d_\theta(x_i, y_{j'})}}. \quad (7)$$

In practice, $L(\theta)$ is approximated using sampled mini-batches and optimized to obtain $\theta$. We then substitute the average cost $d_{\text{avg}}(x_i, y_j)$ with its closed-form approximator $d_\theta(x_i, y_j)$ in $F(Y)$ in (P̂($\beta$)), and update the representatives $\{y_\ell\}$ using a descent method. See Algorithm 2 for details. In Appendix G, we present the detailed description and architecture of Adaptive Distance Estimation Network (ADEN), our proposed attention-based deep neural approximator for $d_{\text{avg}}(x_i, y_j)$.

Table 1: $D$ gap (%) of the ADEN versus the setting that ignores local autonomy $p(k|j,i)$, relative to the ground truth (Algorithm 1) across scenarios where $p(k|j,i)$ depends on $Y$.

| $\kappa$ | $\bar{\gamma}$ | $\zeta$ | $T$ | ADEN | Ignored | | $\kappa$ | $\bar{\gamma}$ | $\zeta$ | $T$ | ADEN | Ignored |
|---|---|---|---|---|---|---|---|---|---|---|---|---|
| 0.1 | 0 | 1 | 0.01 | 2.11 | 10.73 | | 0.3 | 0.5 | 1 | 0.01 | 7.02 | 43.56 |
| 0.1 | 0 | 1 | 100 | 2.08 | 6.67 | | 0.3 | 0.5 | 1 | 100 | 3.24 | 24.69 |
| 0.1 | 0.5 | 1 | 0.01 | 1.87 | 10.01 | | 0.4 | 0 | 1 | 0.01 | 8.03 | 68.89 |
| 0.2 | 0 | 1 | 0.01 | 3.01 | 25.80 | | 0.4 | 0 | 1 | 100 | 3.36 | 35.92 |
| 0.2 | 0 | 1 | 100 | 1.79 | 15.08 | | 0.4 | 0.5 | 1 | 0.01 | 3.47 | 67.63 |
| 0.2 | 0.5 | 1 | 0.01 | 1.50 | 24.90 | | 0.4 | 0.5 | 1 | 100 | 2.68 | 35.89 |
| 0.2 | 0.5 | 1 | 100 | 1.40 | 15.06 | | 0.5 | 0 | 1 | 100 | 5.66 | 49.21 |
| 0.3 | 0 | 1 | 0.01 | 4.80 | 44.62 | | 0.5 | 0.5 | 1 | 0.01 | 1.72 | 100.20 |
| 0.3 | 0 | 1 | 100 | 3.25 | 24.71 | | 0.5 | 0.5 | 1 | 100 | 4.66 | 49.17 |

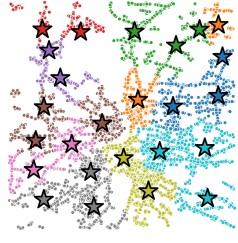 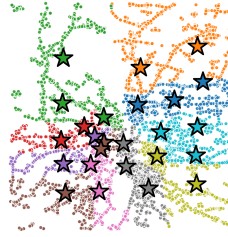 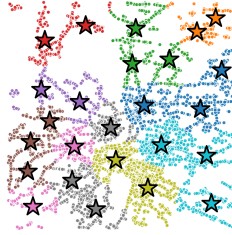 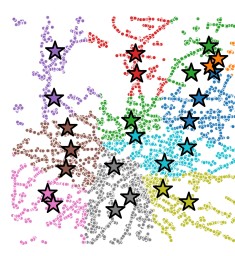

(a) $\kappa = 0.1$, $T = 0.1$    (b) $\kappa = 0.5$, $T = 0.1$    (c) $\kappa = 0.1$, $T = 0.01$    (d) $\kappa = 0.5$, $T = 0.01$

Figure 2: Clustering of the UDT19 dataset under parameterized autonomy for varying $\kappa$. UAVs are indicated by colored stars, and sensor (denoted by $*$) colors denote their associated UAV.

## 5  SIMULATIONS

To evaluate how well our framework accounts for local autonomy, we test it on the synthetic dataset in Figure 1(a), designing scenarios with varying autonomy levels. In some cases, the autonomy $p(k|j,i)$ explicitly depends on the parameters $\mathcal{X}$ and $\mathcal{Y}$; in others, it is independent. These settings capture realistic behaviors where an entity $i$ may reject its prescribed cluster $j$ and instead join another $k \neq j$. Here we choose the local autonomy model such that, with probability $1 - \kappa$, the entity $i$ accepts its prescribed cluster $j$; otherwise, with probability $\kappa$, it chooses an alternative cluster $k \neq j$ according to a softmax distribution, $p(k|j,i) = \kappa \frac{\exp[-c_k(j,i)/T]}{\sum_{t \neq j} \exp[-c_t(j,i)/T]}$, where the cost $c_k(j,i) = \zeta\, d(y_j, y_k) + \bar{\gamma}\, d(x_i, y_k)$ balances *cluster–cluster distance* ($\zeta$) and *cluster–entity distance* ($\bar{\gamma}$). Here, $\kappa$ controls override frequency, and $T$ regulates randomness (uniform as $T \to \infty$, deterministic as $T \to 0$). Varying $\{\kappa, T, \zeta, \bar{\gamma}\}$ yields diverse autonomy scenarios that affect both cluster locations $\{y_\ell\}$ and the central-planner policy $\pi_Y^\beta$. Full hyperparameters of ADEN in different scenarios appear in Appendix H.

We compare Algorithm 1 (ground truth), Algorithm 2 (ADEN-based), and a baseline that ignores autonomy. Table 1 reports objective gaps relative to ground truth, for the dataset in figure 1. The ADEN-based algorithm incurs only modest error: median 3.12%, mean 3.42%, with deviations from 1.40% (small $\kappa$) to 8.03% (intermediate $\kappa$). By contrast, ignoring autonomy produces severe degradation: median 30.84%, mean 36.26%, and up to 100.20%. Performance in this baseline worsens as $\kappa$ increases, while the performance of the ADEN-based algorithm is independent of it and can be further improved through hyperparameter tuning, underscoring its robustness across varying autonomy levels. See Appendix I for clustering solutions at various autonomy levels.

As a second illustrative example, we consider *decentralized sensing* in urban traffic monitoring. Using the UDT19 London Traffic dataset (Loder et al., 2019) (see Appendix I for details on the dataset), which provides geocoordinates of roadside traffic sensors across Greater London, we pose a *facility-location* problem: determine optimal UAV positions to maximize coverage of the sensor network. In practice, sensors may occasionally fail to transmit data to their assigned UAV due to network uncertainties such as packet loss or congestion (Psannis, 2016). When this occurs, a

sensor forwards its measurements to a different UAV, with higher probability for UAVs in adjacent clusters—naturally introducing local autonomy.

We model this behavior using the same transition distribution $p(k|j,i)$ described earlier, setting $\bar{\gamma} = 0$, $\zeta = 1$, and varying $\kappa \in \{0.1, 0.5\}$ and $T \in \{0.1, 0.01\}$. The temperature $T$ controls the sharpness of the softmax: lower values create a strong preference for selecting nearby UAVs whenever the assigned policy cannot be satisfied, while the two $\kappa$ values represent low and high rates of network faults. Figure 2 reports the solutions obtained by ADEN (Algorithm 2).

The results demonstrate a consistent pattern. Increasing $\kappa$ causes the UAV (cluster) representatives to move closer together (closer to the centroid of the entire dataset), reflecting stronger cross-cluster autonomy (Figure 2(b),2(d)). Whereas, in case of low $\kappa$, the UAVs are fairly spread out (Figure 2(a),2(c)). This intuitive outcome is accurately captured by our learned model (ADEN).

Notably, when $\kappa = 0.1, T = 0.1$, ADEN matches the performance of the model-based baseline (ground truth), and for $\kappa = 0.5, T = 0.1$ it achieves approximately a $10\%$ improvement over the ground truth, despite the absence of an explicit autonomy model. These results underscore the ability of our approach to remain competitive with, and in some cases surpass, model-aware methods while scaling to large, high-dimensional decentralized sensing problems.

The case $T = 0.01$ is particularly challenging: the distribution $p(k|j,i)$ becomes sharply peaked, and the large number of entities and clusters add to this challenge. Even in this setting, our model-free solution attains average optimality gaps of only $18.37\%$ and $24.82\%$ for $\kappa = 0.1$ and $\kappa = 0.5$, respectively, relative to a model-based oracle. These gaps can possibly be further reduced through standard hyperparameter tuning and extended training. We summarize these results in Table 3 in the Appendix I. We also study the effect of autonomy on a movie recommendation system; full details provided in Appendix I. Across 40 distinct autonomy levels on the MovieLens dataset (Harper & Konstan, 2015), Algorithm 2 achieves an average percentage gap in $D$ of only $2.81\%$ relative to the ground truth, whereas ignoring autonomy results in a substantially larger average gap of $30.41\%$. Further, the Appendix K analyses ADEN's stability and sensitivity to hyperparameters.

## 6  ANALYSIS AND DISCUSSION

**Scope, limitations, and applicability:** Unlike probabilistic clustering models (e.g., GMMs) that incorporate stochasticity at the level of the assignment policy, local autonomy models stochasticity at the level of the entities themselves. Our proposed framework is general and applies whenever entities exhibit autonomy. When data points are passive and adhere to prescribed assignments, classical clustering methods remain appropriate. A limitation of the class of problems we address is the computational challenge of learning autonomy when it is unknown, especially in large-scale settings or when the autonomy model is complex. The Algorithm 2 resulting from our proposed framework is flexible; for instance, simpler estimators can be used to replace ADEN when autonomy is weak, reducing the computational overhead in such scenarios.

**Unit horizon MDP viewpoint:** The unit-horizon MDP viewpoint of autonomy-aware clustering allows us to motivate the structure of Algorithm 2 from the components of a standard RL pipeline. We begin by sampling a data point, analogous to sampling an initial state in RL. The cluster assignment $\epsilon$-greedy policy plays the role of an $\epsilon$-greedy action selection mechanism in RL. Once a cluster $j \sim \epsilon - \text{greedy} \ \pi_Y^\beta(\cdot|i)$ is prescribed, the data point may associate with a different cluster $k \sim p(\cdot|j,i)$, paralleling a transition from state $s$ to $s'$ under the dynamics $p(s'|a,s)$. The function $d_\theta(x_i, y_j)$ serves as a learned estimate of the autonomy-aware cost, and is updated by minimizing the loss $L(\theta)$ in a manner similar to learning a $Q$-function in deep RL. The key distinction from standard RL is that the learned estimate $d_\theta(x_i, y_j)$ is used not only to update the policy $\pi_Y^\beta(j|i)$, but also to construct $F(Y)$ in ($\hat{P}(\beta)$), which is then optimized over the cluster representatives $Y$.

**Future Work:** There are several natural extensions of our work: (i) extending the autonomy formulation to settings (for instance, density or connectivity based clustering) that operate on distances $d(x_i, x_j)$ between entities and not distance $d(x_i, C_j)$ between the entity and clusters, (ii) addressing problem settings with large, noisy, and correlated entity feature vectors $x_i$ — this requires integrating our framework with deep autoencoder-based clustering setup (Lu & Li, 2021), and (iii) applying robust RL (Panaganti et al., 2022) techniques to study the sensitivity to noise in autonomy models.

## REPRODUCIBILITY STATEMENT

1. For all the theorems presented in this work, the complete details of the underlying assumptions (if any) and the full proofs are provided in the Appendix, with appropriate references made in the main text.

2. The codes were executed on a GPU Cluster, specifically utilizing the `ghx4` partition. This system is comprised of NVIDIA Grace Hopper Superchip nodes, each equipped with an NVIDIA H100 GPU and a Grace CPU. The node provided 16 CPU cores and 1 GPU, with a total of 64 GB of memory allocated for the job.

3. The random seeds / hyperparameter settings for all the simulations in Section 5 are provided in the Appendix H.

4. For the simulations reported in Section 5, the corresponding datasets and implementation codes are included in the supplementary material. Also, the code and data will be made publicly available on Github once review is completed.

5. Similarly, for the simulations described in Appendix E, the relevant dataset (data point locations and their autonomy levels) is also included in the same supplementary material. The code to reproduce the plots here are straightforward and based on Algorithm 1. The inputs involved are $\beta_{\min} = 10^{-3}$, $\beta_{\max} = 10^3$, $\tau = 0.99$, $\rho(i) = \frac{1}{N}$, $\epsilon_{\text{noise}} \sim (10^{-4}) \mathcal{N}(0, 1)$.

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

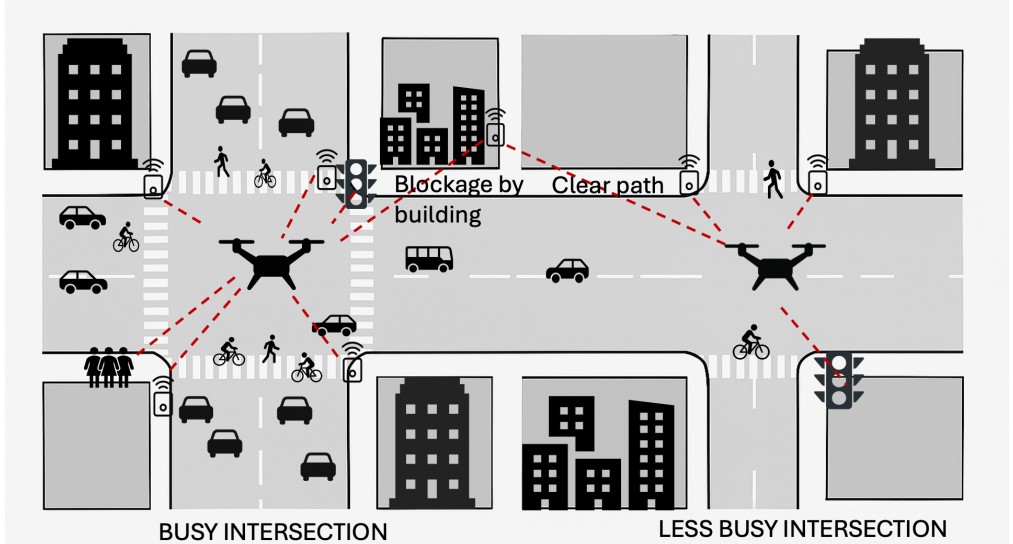

Figure 3: The figure illustrates a decentralized sensing setup in which multiple ground sensors transmit data to their allocated UAVs. One intersection is shown to be busy, with many sensors generating high traffic, while another intersection is less busy. Local autonomy may arise when sensors deviate from their nominal UAV assignment due to congestion at the UAV serving the busy intersection, or due to temporary hindrances such as buildings or moving vehicles that block or degrade the communication link.

## A  LOCAL AUTONOMY IN DECENTRALIZED SENSING

Figure 3 illustrates a representative decentralized sensing scenario in which ground sensors communicate their measurements to nearby UAVs acting as data-collection units. Different regions—such as intersections in an urban environment—may exhibit significantly different sensing loads; one intersection in the figure contains many active sensors (a high-traffic area), while another contains substantially fewer. Even though each sensor is nominally assigned to a specific UAV based on proximity, a variety of local operational conditions may cause deviations from this assignment. For example, the UAV serving the busy intersection may experience congestion or queue buildup due to multiple simultaneous uplink requests. Similarly, intermittent fading, blockage from buildings or passing vehicles, antenna-pattern effects, or energy-saving heuristics at the sensor can cause the effective communication link to a different UAV to become more favorable. In some settings, security or privacy considerations may also prompt a sensor to route its data to an alternative UAV that is temporarily more trusted or more reliable.

These factors collectively give rise to *local autonomy*, whereby a sensor at location $x_i$ may choose to communicate with a UAV $k \neq j$ even if $j$ is its nominally assigned processing unit. As a result, the effective communication cost for the system is not solely determined by the distance $d(x_i, y_j)$ to the assigned UAV, but rather by the autonomy-adjusted cost

$$\mathbb{E}_{k \sim p(\cdot|j,i)}\, d(x_i, y_k), \tag{8}$$

which averages over the UAVs that the sensor might actually use under such variability. Consequently, the UAV locations $y_j$ (cluster representatives) may need to adapt to these behavioral deviations to reduce the effective communication cost and improve reliability. For instance, the UAV at the less busy intersection in Figure 3 may shift towards the left to also cater to the sensors from the busy intersection and the sensor whose view to its closest UAV is blocked by a building. In other words, autonomy not only influences sensor-to-UAV associations, but also reshapes the optimal positioning of UAVs themselves to better accommodate the stochastic communication patterns that arise in realistic decentralized sensing environments.

## B CONVERGENCE PROOF

### B.1 CASE A: WHEN THE ASSUMPTIONS IN THEOREM 1 HOLD TRUE

Parts of the proof below require hessian of $F(Y)$ in ($\hat{P}(\beta)$), which is computed in the Appendix C. We refer to the relevant equations from Appendix C wherever required.

We have the following assumptions:

(i) Non-degenerate clusters - there exists $c > 0$ such that $p_{\pi_\rho}^{Y(t)}(l) \geq c$ for all $l$. In other words, each cluster is a of non-zero mass. Note that this is trivially true for $\pi_Y^\beta$ in (1) at $\beta < \infty$.

(ii) No abrupt shift in cluster mass - let $Y_r(t+1) = Y(t) - \frac{r}{2}\big(\hat{P}_{\pi_\rho}^{Y(t)}\big)^{-1}\nabla F(Y(t))$ be the relaxed update of the iteration in (3), where $r \in (0, 1]$. Then the change in cluster mass from $t$ and $t+1$ is upper bounded. In particular, $\max_{r \in (0,1]} p_{\pi_\rho}^{Y_r(t+1)}(\ell) < 4 p_{\pi_\rho}^{Y(t)}(\ell)$ for all $\ell$, i.e., the $\ell^{\text{th}}$ cluster mass at $t+1$ does not increase by more than 4 times of that at $t$ for all possible values of $r \in (0, 1]$.

Consider fixed point iterations in the inner-loop of the Algorithm 1. Substituting the policy $\pi_Y^\beta(j|i)$ in (1) into the expression for the representative features $y_\ell$ in (2) results into

$$y_\ell = \frac{\sum_{i=1}^N \sum_{j=1}^K \rho(i) p(l|j,i) \frac{\exp\{-\beta d_{\text{avg}}(x_i, y_j)\}}{\sum_{j'=1}^K \exp\{-\beta d_{\text{avg}}(x_i, y_{j'})\}} x_i}{\sum_{i=1}^N \sum_{j=1}^K \rho(i) p(l|j,i) \frac{\exp\{-\beta d_{\text{avg}}(x_i, y_j)\}}{\sum_{j'=1}^K \exp\{-\beta d_{\text{avg}}(x_i, y_{j'})\}}}. \tag{9}$$

Thus, one pass over the equations in (1) and (2) is analogous to the iteration

$$y_\ell(t+1) = \frac{\sum_{i=1}^N \sum_{j=1}^K \rho(i) p(l|j,i) \frac{\exp\{-\beta d_{\text{avg}}(x_i, y_j(t))\}}{\sum_{j'=1}^K \exp\{-\beta d_{\text{avg}}(x_i, y_{j'}(t))\}} x_i}{\sum_{i=1}^N \sum_{j=1}^K \rho(i) p(l|j,i) \frac{\exp\{-\beta d_{\text{avg}}(x_i, y_j(t))\}}{\sum_{j'=1}^K \exp\{-\beta d_{\text{avg}}(x_i, y_{j'}(t))\}}} \quad \forall\, l, \tag{10}$$

$$\Rightarrow y_\ell(t+1) p_{\pi_\rho}^{Y(t)}(l) = \sum_{i=1}^N \sum_{j=1}^K \rho(i) p(l|j,i) \pi_{Y(t)}^\beta(j|i) x_i, \text{ where} \tag{11}$$

$\pi_{Y(t)}^\beta(j|i) = \frac{\exp\{-\beta d_{\text{avg}}(x_i, y_j(t))\}}{\sum_{j'=1}^K \exp\{-\beta d_{\text{avg}}(x_i, y_{j'}(t))\}}$ and $p_{\pi_\rho}^{Y(t)}(l) = \sum_{i=1}^N \rho(i) \sum_{j=1}^K p(l|j,i) \pi_{Y(t)}^\beta(j|i)$. Subtracting $p_{\pi_\rho}^{Y(t)}(l) y_\ell(t)$ from both sides of the equation (11), we obtain

$$\big(y_\ell(t+1) - y_\ell(t)\big) p_{\pi_\rho}^{Y(t)}(l) = -\sum_{i=1}^N \sum_{j=1}^K \rho(i) p(l|j,i) \pi_{Y(t)}^\beta(j|i) \big(y_\ell(t) - x_i\big) \tag{12}$$

$$\Rightarrow \big(y_\ell(t+1) - y_\ell(t)\big) p_{\pi_\rho}^{Y(t)}(l) = -\frac{\partial F(Y(t))}{\partial y_\ell}, \tag{13}$$

which in the stacked vector notation can be re-written as

$$Y(t+1) = Y(t) - \frac{1}{2}\big(\hat{P}_{\pi_\rho}^{Y(t)}\big)^{-1}\nabla F(Y(t)) =: Y(t) + S(t). \tag{14}$$

Here, $\hat{P}_{\pi_\rho}^{Y(t)} = P_{\pi_\rho}^Y(t) \otimes I_d$, $I_d$ is a $d \times d$ Identity matrix, $\otimes$ denotes the Kronecker product, $P_{\pi_\rho}^{Y(t)} \in \mathbb{R}^{K \times K}$ is a positive definite diagonal matrix with $\big[P_{\pi_\rho}^{Y(t)}\big]_{ll} = p_{\pi_\rho}^{Y(t)}(l) \geq c > 0$, $\nabla F(Y(t)) = \Big[\frac{dF(Y(t))}{dy_1}^\top, \cdots, \frac{dF(Y(t))}{dy_K}^\top\Big]^\top \in \mathbb{R}^{Kd}$. At every time instant $t$, we define

$$L(Y(t)) := \sup_{r \in [0,1]} \lambda_{\max}\Big(\big(\hat{P}_{\pi_\rho}^{Y(t)}\big)^{-1/2}\nabla^2 F(Y(t) + rS(t))\big(P_{\pi_\rho}^{Y(t)}\big)^{-1/2}\Big), \tag{15}$$

Let $Y_r(t+1) := Y(t) + rS(t)$, and $g(r) := F(Y_r(t+1))$. Then $g'(r) = \nabla F(Y_r(t+1))^\top S(t)$, $g''(r) = S(t)^\top \nabla^2 F(Y_r(t+1))^\top S(t)$. To avoid notational clutter, let $M_t := \big(\hat{P}_{\pi_\rho}^{Y(t)}\big)^{-1/2}\nabla^2 F(Y_r(t+1))\big(P_{\pi_\rho}^{Y(t)}\big)^{-1/2}$. Then

$$g''(r) = \Big(\big(\hat{P}_{\pi_\rho}^{Y(t)}\big)^{1/2} S(t)\Big)^\top M_t \Big(\big(\hat{P}_{\pi_\rho}^{Y(t)}\big)^{1/2} S(t)\Big) \tag{16}$$

$$\Rightarrow g''(r) \le \lambda_{\max}(M_t)\Big(S(t)^\top\big(\hat{P}_{\pi_\rho}^{Y(t)}\big)S(t)\Big) \tag{17}$$

$$\Rightarrow g''(r) \le \lambda_{\max}(M_t)\|S\|^2_{P_{\pi_\rho}^{Y(t)}} \le L(Y(t))\|S(t)\|^2_{P_{\pi_\rho}^{Y(t)}} \tag{18}$$

Integrating both sides we obtain

$$g(1) = g(0) + g'(0) + \int_0^1 (1-t)g''(t)dt \tag{19}$$

$$\Rightarrow F(Y(t+1)) \le F(Y(t)) + \nabla F(Y(t))^\top S(t) + L(Y(t))\|S(t)\|^2_{P_{\pi_\rho}^{Y(t)}} \int_0^1 (1-t)dt \tag{20}$$

$$\Rightarrow F(Y(t+1)) \le F(Y(t)) + \nabla F(Y(t))^\top S(t) + \frac{1}{2}L(Y(t))\|S\|^2_{P_{\pi_\rho}^{Y(t)}} \tag{21}$$

Substituting $S(t) = -\frac{1}{2}\big(\hat{P}_{\pi_\rho}^{Y(t)}\big)^{-1}\nabla F(Y(t))$ in (21), we get:

$$F(Y(t+1)) \le F(Y(t)) - \frac{1}{2}\nabla F(Y(t))^\top\big(P_{\pi_\rho}^{Y(t)}\big)^{-1}\nabla F(Y(t))$$
$$+ \frac{1}{8}L(Y(t))\Big(\nabla F(Y(t))^\top\big(P_{\pi_\rho}^{Y(t)}\big)^{-1}\Big)\nabla F(Y(t))\Big) \tag{22}$$

$$\Rightarrow \Big(\frac{1}{2} - \frac{1}{8}L(Y(t))\Big)\|\nabla F(Y(t))\|_{\big(P_{\pi_\rho}^{Y(t)}\big)^{-1}} \le F(Y(t)) - F(Y(t+1)) \tag{23}$$

Telescopic summation over all $t \in \{0,1,\dots,T\}$ gives us

$$\sum_{t=0}^T \nu_t\|\nabla F(Y(t))\|_{\big(P_{\pi_\rho}^{Y(t)}\big)^{-1}} \le F(Y(0)) - F(Y(T)), \text{ where } \nu_t = \frac{1}{2} - \frac{1}{8}L(Y(t)) \tag{24}$$

$$\Rightarrow \sum_{t=0}^T \nu_t\|\nabla F(Y(t))\|_{\big(P_{\pi_\rho}^{Y(t)}\big)^{-1}} \le F(Y(0)) - F_{\min} < \infty \tag{25}$$

There always exists a minimum $F_{\min}$ such that $F(Y(T)) \ge F_{\min}$ for all $Y$; note that for $0 < \beta < \infty$, the Log-Sum-Exponential function $F(Y(T))$ in ($\hat{P}(\beta)$) is always lower bounded. Thus, if $\nu_t > 0$ $\forall\, t$, then, as $T \to \infty$, $\|\nabla F(Y(T))\|_{\big(P_{\pi_\rho}^{Y(T)}\big)^{-1}} \to 0$.

The condition $\nu_t > 0$ holds true if $L(Y(t)) < 4$. We have from (52) in Appendix C that

$$\nabla^2 F(Y_r(t+1)) = \hat{P}_{\pi_\rho}^{Y_r(t+1)} - 2\beta\underbrace{\sum_{i=1}^N\Big(\sum_{j=1}^K P_A^{ij}z_iz_i^\top P^{ij} - \rho(i)P^iz_iz_i^\top P^i\Big)}_{=\Delta}, \tag{26}$$

where $\hat{P}_{\pi_\rho}^{Y_r(t+1)}$ is positive definite by definition in Appendix C, and the matrix $\Delta$ is positive semi-definite too under the definition of the matrices $P_A^{ij}, z, P^{ij}, P^i$ detailed in Appendix C. Actually, the latter follows from the fact that for any $\Psi = [\psi_1^\top \dots \psi_K^\top]^\top \in \mathbb{R}^{Kd}$, $\Psi^\top\Delta\Psi = \sum_{i=1}^N \rho(i)\delta_i$, where

$$\delta_i = \sum_{j=1}^K \pi_Y^\beta(j|i)\Big(\sum_{k=1}^K p(k|j,i)[y_k - x_i]^\top\psi_k\Big)^2 - \Big(\sum_{j,k=1}^K \pi_Y^\beta(j|i)p(k|j,i)[y_k - x_i]^\top\psi_k\Big)^2. \tag{27}$$

Note that $\delta_i \ge 0$, because variance of $\Big(\sum_{k=1}^K p(k|j,i)[y_k - x_i]^\top\psi_k\Big)$ computed with respect to the distribution $\pi_Y^\beta(\cdot|i)$ is always non-negative. Thus, we can say that $\nabla^2 F(Y_r(t+1)) \preceq P_{\pi_\rho}^{Y_r(t+1)}$, in other words $P_{\pi_\rho}^{Y_r(t+1)} - \nabla^2 F(Y_r(t+1))$ is positive semi-definite. Thus, we have that

$$\lambda_{\max}\Big(\big(\hat{P}_{\pi_\rho}^{Y(t)}\big)^{-1/2}\nabla^2 F(Y_r(t+1))\big(P_{\pi_\rho}^{Y(t)}\big)^{-1/2}\Big) \le \lambda_{\max}\Big(\big(\hat{P}_{\pi_\rho}^{Y(t)}\big)^{-1}\hat{P}_{\pi_\rho}^{Y_r(t+1)}\Big) \tag{28}$$

$$\Rightarrow L(Y(t)) \le \max_{r\in[0,1]}\lambda_{\max}\Big(\big(\hat{P}_{\pi_\rho}^{Y(t)}\big)^{-1}\hat{P}_{\pi_\rho}^{Y_r(t+1)}\Big) = \max_{r\in[0,1]}\max_{1\le l\le K}\frac{p_{\pi_\rho}^{Y_r(t+1)}(l)}{p_{\pi_\rho}^{Y(t)}(l)}. \tag{29}$$

Under the assumption that for any cluster, its mass does not drastically change, i.e., $\frac{p_{\pi_\rho}^{Y_r(t+1)}}{p_{\pi_\rho}^{Y(t)}} < 4$ for all $r \in (0,1]$, we obtain that $L(Y(t)) < 4$. Thus $\nu_t > 0$, and $\|\nabla F(Y(T))\|_{\left(P_{\pi_\rho}^{Y(T)}\right)^{-1}} \to 0$ as $T \to \infty$. This is equivalent to $\nabla F(Y(T)) \to 0$, which implies that the iterations (14) converge to a stationary point.

## B.2 CASE B: WHEN THE ASSUMPTIONS IN THEOREM 1 DO NOT HOLD

Here, we replace the gradient descent steps in (3) with descent steps of the form

$$Y(t+1) = Y(t) - \sigma_t \nabla F(Y(t)) =: Y(t) + \sigma_t S(t), \tag{30}$$

where the step-size $\sigma_t$ is designed using Armijo's rule (Luenberger et al., 1984). More precisely, we follow the following steps:

1. Let $m = 0$, $\sigma_{m,t} = s$, $\varrho \in (0,1), \xi \in (0,1)$ be Armijo's parameter.
2. Check if

$$F\big(Y(t) - \sigma_{m,t}\nabla F(Y(t))\big) - F\big(Y(t)\big) \leq -\varrho\sigma_{m,t}\|\nabla F(Y(t))\|_2^2 \tag{31}$$

3. If yes: $\sigma_t \leftarrow \sigma_{m,t}$ and exit. If not: $\sigma_{m+1,t} = \xi\sigma_{m,t}$, $m \leftarrow m + 1$. Go to step 2.

Note that if the above steps terminate, then we obtain a step size $\sigma_t$ that enables descent $F\big(Y(t) - \sigma_t\nabla F(Y(t))\big) \leq F\big(Y(t)\big)$. We next show that for the free-energy function $F(Y)$ in $(\hat{P}(\beta))$ the above steps always converge. In other words, we show that there always exists a $\sigma_t$ such that $F\big(Y(t) - \sigma_t\nabla F(Y(t))\big) \leq F\big(Y(t)\big)$.

$$L_\sigma(Y(t)) := \sup_{r \in [0,1]} \lambda_{\max}\Big(\nabla^2 F(Y(t) + r\sigma_t S(t))\Big) \tag{32}$$

Let $Y_{r,\sigma}^t = Y(t) + r\sigma_t S(t)$, $h(r) := F(Y_{r,\sigma}^t)$. Then, $h'(r) = \sigma_t \nabla F(Y_{r,\sigma}^t)^\top S(t)$, $h''(r) = \sigma_t^2 S(t)^\top \nabla^2 F(Y_{r,\sigma}^t) S(t)$. To avoid notational clutter, let $\hat{M}_t := \nabla^2 F(Y(t) + r\sigma_t S(t))$. Then

$$h''(r) = \sigma_t^2 \big(S(t)\big)^\top \hat{M}_t\big(S(t)\big) \tag{33}$$

$$\Rightarrow h''(r) \leq \sigma_t^2 \lambda_{\max}\big(\hat{M}_t\big)\Big(S(t)^\top S(t)\Big) \tag{34}$$

$$\Rightarrow h''(r) \leq \sigma_t^2 \lambda_{\max}\big(\hat{M}_t\big)\big\|S(t)\big\|_2^2 \leq \sigma_t^2 L_\sigma(Y(t))\big\|S(t)\big\|_2^2 \tag{35}$$

Integrating both sides we obtain

$$h(1) = h(0) + h'(0) + \int_0^1 (1-t)h''(t)dt \tag{36}$$

$$\Rightarrow F(Y(t) + \sigma_t S(t)) \leq F(Y(t)) + \sigma_t \nabla F(Y(t))^\top S(t) + \sigma_t^2 \frac{1}{2}L_\sigma(Y(t))\|S(t)\|_2^2 \tag{37}$$

Setting step-size at $\sigma_{m,t}$ in (37), we obtain

$$F\big(Y(t) - \sigma_{m,t}\nabla F(Y(t))\big) - F(Y(t)) \leq \sigma_{m,t}\nabla F(Y(t))^\top S(t) + \frac{\sigma_{m,t}^2}{2}L_\sigma(Y(t))\|S(t)\|_2^2 \tag{38}$$

The Armijo's condition in (31) will be true if

$$\sigma_{m,t}\nabla F(Y(t))^\top S(t) + \frac{\sigma_{m,t}^2}{2}L_\sigma(Y(t))\|S(t)\|_2^2 \leq -\varrho\sigma_{m,t}\|\nabla F(Y(t))\|_2^2 \tag{39}$$

Substituting $S(t) = -\nabla F(Y(t))$, we obtain

$$-\|\nabla F(Y(t))\|_2^2 + \frac{\sigma_{m,t}}{2}L_\sigma(Y(t))\|\nabla F(Y(t))\|_2^2 \leq -\varrho\|\nabla F(Y(t))\|_2^2 \tag{40}$$

**Algorithm 3:** Autonomy-aware clustering — when assumptions in Theorem 1 fail

**Input:** $\beta_{\min}, \beta_{\max}, \tau, K, \mathcal{X}, \rho(i), p(k|j,i) \,\forall\, i,j,k$, and Armijo's parameters $s, \varrho \in (0,1), \xi \in (0,1)$;
**Output:** Assignment policy $\pi$, and cluster representatives $\{y_\ell\}_{\ell=1}^K$
**Initialize:** $\beta = \beta_{\min}, \pi_Y^\beta(j|i) = \frac{1}{K} \,\forall\, i,j$, and $\{y_\ell\}_{\ell=1}^K$ using (2).
**while** $\beta \leq \beta_{\max}$ **do**
     **while** *until convergence* **do**
         $m = 0; \sigma_{m,t} = s$; **while** *True* **do**
             **if** $F(Y(t) - \sigma_{m,t}\nabla F(Y(t))) - F(Y(t)) \leq -\varrho\sigma_{m,t}\|\nabla F(Y(t))\|_2^2$ **then**
                 $\sigma_t \leftarrow \sigma_{m,t}$; break;
             **else**
                 $\sigma_{m+1,t} \leftarrow \xi\sigma_{m,t}; \quad m \leftarrow m+1$
         $Y(t+1) \leftarrow Y(t) - \sigma_t\nabla F(Y(t)); t \leftarrow t+1$;
     $\beta \leftarrow \tau\beta$; set $y_\ell \leftarrow y_\ell + \epsilon_{\text{noise}}$ (to escape saddle) $\forall\, \ell$

$$\Rightarrow -1 + \frac{\sigma_{m,t}}{2}L_\sigma(Y(t)) \leq -\varrho \qquad \Rightarrow \sigma_{m,t} \leq \frac{2}{L_\sigma(Y(t))}\big(1-\varrho\big) \tag{41}$$

$$\Rightarrow \sigma_{0,t}\xi^m \leq \frac{2}{L_\sigma(Y(t))}\big(1-\varrho\big) \tag{42}$$

Since $\xi < 1$, there exist a finite number of iterations $m$ beyond which the above inequality will be true. In other words, Armijo's condition in (31) will be satisfied. Thus, resulting into an appropriate step size $\sigma_t$. See Algorithm 3 for details.

# C  PHASE TRANSITION AND CRITICAL ANNEALING PARAMETER

## C.1  HESSIAN COMPUTATION

We define the following matrices:

1. $\hat{P}^{ij} \in \mathbb{R}^{K\times K}$ is a diagonal matrix, such that $[\hat{P}^{ij}]_{kk} = p(k|j,i)$, $P^{ij} = \hat{P}^{ij} \otimes \mathbb{I}_d$, where $\mathbb{I}_d$ is a $d \times d$ identity matrix,

2. $P^i = \sum_{j=1}^K \pi_Y^\beta(j|i)P^{ij}$,

3. $z_i = Y - X^i \in \mathbb{R}^{Kd}$, where $Y = \begin{bmatrix} y_1^\top & y_2^\top & \ldots & y_K^\top \end{bmatrix}^\top$, $X^i = \mathbf{1}_{Kd} \otimes x_i$, and $\otimes$ denotes the Kronecker product.

4. $\hat{P}_A^{ij} \in \mathbb{R}^{K\times K}$ is a diagonal matrix; $\left[\hat{P}_A^{ij}\right]_{kk} = \rho(i)\pi_Y^\beta(j|i)p(k|j,i)$, $P_A^{ij} = \hat{P}_A^{ij} \otimes \mathbb{I}_d \in \mathbb{R}^{Kd\times Kd}$,

5. $P_{\pi_\rho}^Y \in \mathbb{R}^{K\times K}$ is a diagonal matrix such that $\left[P_{\pi_\rho}^Y\right]_{kk} = \sum_{i=1}^N \sum_{j=1}^K \rho(i)\pi_Y^\beta(j|i)p(k|j,i)$, and $\hat{P}_{\pi_\rho}^Y = P_{\pi_\rho}^Y \otimes \mathbb{I}_d \in \mathbb{R}^{Kd\times Kd}$. Note that, under the Gibbs' distribution of $\pi_Y$ in (1), $\left[P_{\pi_\rho}^Y\right]_{kk} > 0$ for $\beta < \infty$, thus making $\hat{P}_{\pi_\rho}^Y$ and $P_{\pi_\rho}^Y$ positive definite matrices.

Phase transitions occur when the cluster representatives $\{y_\ell\}$ in (2), given by $\frac{\partial F}{\partial y_\ell} = 0$, are no longer the local minima. In other words, $\frac{\partial F(Y)}{\partial Y} = 0$, where $Y = [y_1^\top, \ldots, y_K^\top]^\top$ but, there exist some perturbation direction $\Psi = [\psi_1^\top, \ldots, \psi_K^\top]^\top \in \mathbb{R}^{Kd}$ such that the Hessian $\mathcal{H}(Y, \pi, \Psi, \beta) =$

$$\frac{d^2 F(Y+\epsilon\Psi)}{d\epsilon^2}\bigg|_{\epsilon=0} = \Psi^\top\left[\hat{P}_{\pi_\rho}^Y - 2\beta\left(\sum_{i=1}^N \sum_{j=1}^K P_A^{ij} z_i z_i^\top P^{ij} - \sum_{i=1}^N \rho(i)P^i z_i z_i^\top P^i\right)\right]\Psi, \tag{43}$$

is no longer positive definite.

Computing the Hessian in (43) $\frac{d^2 F(Y+\epsilon\Psi)}{d\epsilon^2}\big|_{\epsilon=0} =$

$$= \sum_{i=1}^N \rho(i)\sum_{j=1}^K \pi_Y^\beta(j|i)\left[\sum_{k=1}^K p(k|j,i)\psi_k^\top\psi_k - 2\beta\left(\sum_{k=1}^K p(k|j,i)[y_k - x_i]^\top\psi_k\right)^2\right] \tag{44}$$

$$+ 2\beta \sum_{i=1}^{N} \rho(i) \left[ \sum_{j,k=1}^{K} \pi_Y^\beta(j|i) p(k|j,i) [y_k - x_i]^\top \psi_k \right]^2 \tag{45}$$

$$= \sum_{i=1}^{N} \rho(i) \left[ \sum_{j=1}^{K} \pi_Y^\beta(j|i) \Psi^\top \left[ P^{ij} - 2\beta P^{ij} z_i z_i^\top P^{ij} \right] \Psi \right. \tag{46}$$

$$\left. + 2\beta \left[ \sum_{j,k=1}^{K} \pi_Y^\beta(j|i) p(k|j,i) [y_k - x_i]^\top \psi_k \right]^2 \right] \tag{47}$$

$$= \Psi^\top \left[ \hat{P}_{\pi_\rho}^Y - 2\beta \sum_{i=1}^{N} \sum_{j=1}^{K} P_A^{ij} z_i z_i^\top P^{ij} \right] \Psi + 2\beta \sum_{i=1}^{N} \rho(i) \left[ \sum_{j,k=1}^{K} \pi_Y^\beta(j|i) p(k|j,i) [y_k - x_i]^\top \psi_k \right]^2 \tag{48}$$

$$= \Psi^\top \left[ \hat{P}_{\pi_\rho}^Y - 2\beta \sum_{i=1}^{N} \sum_{j=1}^{K} P_A^{ij} z_i z_i^\top P^{ij} \right] \Psi + 2\beta \sum_{i=1}^{N} \rho(i) \left[ \sum_{j=1}^{K} \pi_Y^\beta(j|i) z_i^\top P^{ij} \Psi \right]^2 \tag{49}$$

$$= \Psi^\top \left[ \hat{P}_{\pi_\rho}^Y - 2\beta \sum_{i=1}^{N} \sum_{j=1}^{K} P_A^{ij} z_i z_i^\top P^{ij} \right] \Psi + 2\beta \sum_{i=1}^{N} \rho(i) \left[ z_i^\top P^i \Psi \right]^2 \tag{50}$$

$$= \Psi^\top \left[ \hat{P}_{\pi_\rho}^Y - 2\beta \sum_{i=1}^{N} \sum_{j=1}^{K} P_A^{ij} z_i z_i^\top P^{ij} \right] \Psi + 2\beta \Psi^\top \left[ \sum_{i=1}^{N} \rho(i) \left[ P^i z_i z_i^\top P^i \right] \right] \Psi \tag{51}$$

$$= \Psi^\top \left[ \hat{P}_{\pi_\rho}^Y - 2\beta \left( \sum_{i=1}^{N} \sum_{j=1}^{K} P_A^{ij} z_i z_i^\top P^{ij} - \sum_{i=1}^{N} \rho(i) P^i z_i z_i^\top P^i \right) \right] \Psi \tag{52}$$

## C.2  $\beta_{\text{CR}}$ - CRITICAL ANNEALING PARAMETER VALUE

The Hessian can be re-written as

$$\mathcal{H}(Y, \pi, \Psi, \beta) =$$

$$\Psi^\top (\hat{P}_{\pi_\rho}^Y)^{\frac{1}{2}} \left[ I - 2\beta (\hat{P}_{\pi_\rho}^Y)^{-\frac{1}{2}} \left( \sum_{i=1}^{N} \sum_{j=1}^{K} P_A^{ij} z_i z_i^\top P^{ij} - \sum_{i=1}^{N} \rho(i) P^i z_i z_i^\top P^i \right) (\hat{P}_{\pi_\rho}^Y)^{-\frac{1}{2}} \right] (\hat{P}_{\pi_\rho}^Y)^{\frac{1}{2}} \Psi.$$

As evident from the above expression and the fact that $\beta$ gets annealed from a small value to a large value, the critical $\beta_{\text{cr}}$ at which Hessian loses rank is given by $\dfrac{1}{2\lambda_{\max}\left( (\hat{P}_{\pi_\rho}^Y)^{-\frac{1}{2}} \Delta (\hat{P}_{\pi_\rho}^Y)^{-\frac{1}{2}} \right)}$.

# D  SENSITIVITY OF $Y$ TO THE ANNEALING PARAMETER $\beta$

Consider the expression of the cluster representatives $y_\ell$ in (2). We re-write this expression as

$$y_\ell = \frac{\sum_{i=1}^{N} p_{\pi_\rho}^Y(l, i) x_i}{p_{\pi_\rho}^Y(l)},$$

where $p_{\pi_\rho}^Y(l, i) = \sum_{j=1}^{K} \rho(i) \pi_Y^\beta(j|i) p(l|j,i)$ and $p_{\pi_\rho}^Y(l) = \sum_{i=1}^{N} p_{\pi_\rho}^Y(l, i)$. We have that

$$\frac{dy_\ell}{d\beta} = \frac{1}{p_{\pi_\rho}^Y(l)} \sum_{i=1}^{N} \frac{dp_{\pi_\rho}^Y(l, i) x_i}{d\beta} - \frac{1}{p_{\pi_\rho}^Y(l)^2} \sum_{i=1}^{N} p_{\pi_\rho}^Y(l, i) x_i \frac{dp_{\pi_\rho}^Y(l)}{d\beta}, \tag{53}$$

where $\dfrac{dp_{\pi_\rho}^Y(l, i)}{d\beta} = \sum_{j=1}^{K} \rho(i) p(l|j,i) \dfrac{d\pi_Y^\beta(j|i)}{d\beta}$, where $\pi_Y^\beta(j|i)$ is the Gibbs' distribution in (1).
We obtain that

$$\frac{d\pi_Y^\beta(j|i)}{d\beta} = -\pi_Y^\beta(j|i) \left[ d_{\text{avg}}(x_i, y_j) + 2\beta \sum_{k=1}^{K} p(k|j,i) (y_k - x_i)^\top \frac{dy_k}{d\beta} \right]$$

$$+ \pi_Y^\beta(j|i) \sum_{j'=1}^M \pi_Y^\beta(j'|i)\Big[d_{\mathrm{avg}}(x_i, y_{j'}) + 2\beta \sum_{k=1}^K p(k|j', i)(y_k - x_i)^\top \frac{dy_k}{d\beta}\Big] \tag{54}$$

Substituting (54) in (53), and algebraically simplifying, we obtain

$$\frac{dy_\ell}{d\beta} = \frac{1}{p_{\pi_\rho}^Y(l)} \sum_{i=1}^N \sum_{j=1}^K \rho(i)p(l|j,i)\pi_Y^\beta(j|i)d_{\mathrm{avg}}(x_i, y_j)\big(y_\ell - x_i\big)$$

$$+ \frac{2\beta}{p_{\pi_\rho}^Y(l)} \sum_{i=1}^N \sum_{j=1}^K \rho(i)p(l|j,i)\pi_Y^\beta(j|i) \sum_{k=1}^K p(k|j,i)(y_k - x_i)^\top \frac{dy_k}{d\beta}\big(y_\ell - x_i\big)$$

$$- \frac{1}{p_{\pi_\rho}^Y(l)} \sum_{i=1}^N \sum_{j=1}^K \rho(i)p(l|j,i)\pi_Y^\beta(j|i) \sum_{j'=1}^K \pi_Y^\beta(j'|i)d_{\mathrm{avg}}(x_i, y_{j'})\big(y_\ell - x_i\big)$$

$$- \frac{2\beta}{p_{\pi_\rho}^Y(l)} \sum_{i=1}^N \sum_{j=1}^K \rho(i)p(l|j,i)\pi_Y^\beta(j|i) \sum_{j'=1}^K \pi_Y^\beta(j'|i) \sum_{k=1}^K p(k|j',i)(y_k - x_i)^\top \frac{dy_k}{d\beta}\big(y_\ell - x_i\big) \tag{55}$$

Multiplying (55) by $p_{\pi_\rho}(l)\frac{dy_\ell}{d\beta}^\top$ on both the sides and summing up over all $l$, we obtain:

$$T_1 := \sum_{l=1}^K p_{\pi_\rho}^Y(l)\frac{dy_\ell}{d\beta}^\top \frac{dy_\ell}{d\beta} = \sum_{i=1}^N \sum_{j,l=1}^K \rho(i)p(l|j,i)\pi_Y^\beta(j|i)d_{\mathrm{avg}}(x_i, y_j)\frac{dy_\ell}{d\beta}^\top (y_\ell - x_i)$$

$$+ \underbrace{2\beta \sum_{i=1}^N \sum_{j=1}^K \rho(i)\pi_Y^\beta(j|i)\Big[\sum_{k=1}^K p(k|j,i)\frac{dy_k}{d\beta}^\top (y_k - x_i)\Big]^2}_{T_2}$$

$$- \sum_{i=1}^N \sum_{l=1}^K p_{\pi_\rho}^Y(l,i) \sum_{j'=1}^K \pi_Y^\beta(j'|i)d_{\mathrm{avg}}(x_i, y_{j'})\frac{dy_\ell}{d\beta}^\top (y_\ell - x_i)$$

$$- \underbrace{2\beta \sum_{i=1}^N \rho(i)\Big[\sum_{j=1}^K \pi_Y^\beta(j|i) \sum_{k=1}^K p(k|j,i)\frac{dy_k}{d\beta}^\top (y_k - x_i)\Big]^2}_{T_3}, \tag{56}$$

which when re-arranged gives

$$T_1 - T_2 + T_3 =$$

$$\underbrace{\sum_{i=1}^N \sum_{j,l,j'=1}^K \rho(i)p(l|j,i)\pi_Y^\beta(j|i)\pi_Y^\beta(j'|i)\Big[d_{\mathrm{avg}}(x_i, y_j) - d_{\mathrm{avg}}(x_i, y_{j'})\Big]\frac{dy_\ell}{d\beta}^\top (y_\ell - x_i)}_{=:T_4}. \tag{57}$$

Now, we'll bound some of the terms in the expression $T_4$. Note that, from the expression in (1), $\pi_Y^\beta(j|i) \leq \exp\big\{-\beta\big(d_{\mathrm{avg}}(x_i, y_j) - d_{\mathrm{avg}}(x_i, y_{j'})\big)\big\}$, which implies

$$\pi_Y^\beta(j|i)\big(d_{\mathrm{avg}}(x_i, y_j) - d_{\mathrm{avg}}(x_i, y_{j'})\big) \tag{58}$$

$$\leq \big(d_{\mathrm{avg}}(x_i, y_j) - d_{\mathrm{avg}}(x_i, y_{j'})\big)e^{\big\{-\beta\big(d_{\mathrm{avg}}(x_i,y_j)-d_{\mathrm{avg}}(x_i,y_{j'})\big)\big\}} \leq \frac{e^{-1}}{\beta}, \tag{59}$$

where the last inequality follows from the fact that $xe^{-\beta x} \leq \dfrac{e^{-1}}{\beta}$ for $\beta > 0$. Substituting this bound in (57), we obtain

$$T_4 \leq \frac{e^{-1}}{\beta} \sum_{i=1}^N \sum_{j,l,j'=1}^K \rho(i)p(l|j,i)\pi_Y^\beta(j|i)\pi_Y^\beta(j'|i)\frac{dy_\ell}{d\beta}^\top (y_\ell - x_i) \tag{60}$$

$$= \frac{e^{-1}}{\beta} \sum_{i=1}^{N} \sum_{l=1}^{K} p_{\pi_\rho}^Y(l,i) \frac{dy_\ell}{d\beta}^\top (y_\ell - x_i) \le \frac{e^{-1}}{\beta} \sum_{i=1}^{N} \sum_{l=1}^{K} \left\| \frac{dy_\ell}{d\beta} \right\| R_\Omega, \tag{61}$$

where $R_\Omega$ quantifies the size of the domain $\Omega$ (for instance, radius of the smallest sphere containing all the data points $\{x_i\}$). Further note from expression (45) that $T_1 - T_2 + T_3$ is essentially the Hessian $\mathcal{H}(Y, \pi, \Psi, \beta)$ where the perturbation $\Psi = \left[ \frac{dy_1}{d\beta}^\top \dots \frac{dy_K}{d\beta}^\top \right]^\top$. Thus, we have that

$$\Rightarrow \sum_{l=1}^{K} \frac{dy_\ell}{d\beta}^\top \frac{\partial^2 F}{\partial y_\ell^2} \frac{dy_\ell}{d\beta} \le \frac{e^{-1}}{\beta} N R_\Omega \sum_{l=1}^{K} \left\| \frac{dy_\ell}{d\beta} \right\| \tag{62}$$

$$\Rightarrow \sum_{l=1}^{K} \lambda_{\min}\left(\frac{\partial^2 F}{\partial y_\ell^2}\right) \left\| \frac{dy_\ell}{d\beta} \right\|^2 \le \frac{e^{-1}}{\beta} N R_\Omega \sum_{l=1}^{K} \left\| \frac{dy_\ell}{d\beta} \right\| \tag{63}$$

$$\Rightarrow \sum_{l=1}^{K} \delta \left\| \frac{dy_\ell}{d\beta} \right\|^2 \le \frac{e^{-1}}{\beta} N R_\Omega \sum_{l=1}^{K} \left\| \frac{dy_\ell}{d\beta} \right\|, \tag{64}$$

where $\delta = \min_l \left[ \lambda_{\min}\left(\frac{\partial^2 F}{\partial y_\ell^2}\right) \right]$, and $\lambda_{\min}(\cdot)$ is the minimum eigenvalue. Note that $\sum_{l=1}^{K} \left\| \frac{dy_\ell}{d\beta} \right\| \le \sqrt{K} \left\| \frac{dY}{d\beta} \right\|$ by Cauchy-Schwarz inequality, where

$$\left\| \frac{dY}{d\beta} \right\| = \sqrt{\left\| \frac{dy_1}{d\beta} \right\|^2 + \dots + \left\| \frac{dy_K}{d\beta} \right\|^2}. \tag{65}$$

Thus, from (64) we have that

$$\delta \left\| \frac{dY}{d\beta} \right\|^2 \le \frac{e^{-1}}{\beta} N R_\Omega \sqrt{K} \left\| \frac{dY}{d\beta} \right\| \tag{66}$$

$$\Rightarrow \left\| \frac{dY}{d\beta} \right\| \le \frac{e^{-1}}{\beta\delta} N R_\Omega \sqrt{K} \tag{67}$$

## E  CHANGE IN $Y$ VERSUS $\beta$ AND CRITICAL TEMPERATURES

Here we illustrate how $Y$ changes drastically near critical $\beta_{\mathrm{cr}}$, and remains largely unchanged between two consecutive $\beta_{\mathrm{cr}}$. Figure 4(a) illustrates the dataset that we consider for this illustration. It contains 3200 data points, and we divide it into $K = 16$ clusters, i.e., 16 cluster representatives $\{y_\ell\}$. Each data point has local autonomy, such that it honors the prescribed cluster 15 out 16 times, and remaining times it overrides the prescription and uniformly associates with the remaining 15 clusters. Figure 4(b) plots $\|\Delta Y(\beta)\|$ versus $\beta$. Note that change in $Y$ remains largely zero except at 3 instances at which critical $\beta_{\mathrm{cr}}$ was attained. Initially all the representatives $\{y_\ell\}$ are coincident, i.e., all have the same feature vector values. At first $\beta_{\mathrm{cr}}$, 4 distinct representative feature vectors value are formed, where each unique representative feature vector value is shared by 4 representatives. At the second $\beta_{\mathrm{cr}}$ each of the previous 4 unique representative vector values give rise to 2 unique representative vector value — making a total of 8 unique representative feature values at this point. Here, each unique representative feature vector value is shared by 2 representatives in $\{y_l\}$. At the third $\beta_{\mathrm{cr}}$ each of the previous 8 unique representative vectors give rise to 2 unique representative vector values — making a total of unique 16 representative vectors. See the .mp4 file "Submission-WithPT.mp4" (where plot title in every frame shows $\frac{1}{\beta}$) submitted as supplementary material for a clearer understanding.

## F  REINFORCEMENT-BASED LEARNING FOR CASE (C1)

Case (C1) - When $\mathcal{X}$ contains a tractable number of data points, $d(x_i, y_k)$ is available in closed form, and $p(l|j, i)$ is independent of $y_\ell$.

*Learning $\pi_Y^\beta$ (S1):* Our mechanism to learn the assignment policy $\pi_Y^\beta$ in (1) is akin to that of learning the *control policy* in reinforcement learning (RL) frameworks (Sutton & Barto, 2018).

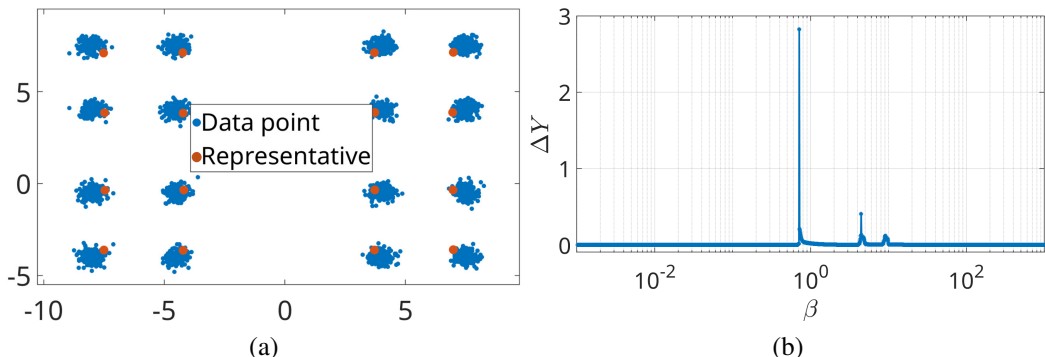

Figure 4: (a)Dataset, (b) Change in $Y$ versus $\beta$ demonstrates phenomenon of phase transitions

Let $q_t(x_i, y_j)$ be the estimate of the average cost $d_{\mathrm{avg}}(x_i, y_j)$ at a time instant $t$, and $\pi_t(j|i) = \mathrm{softmax}_j(-\beta q_t(x_i, y_j))$ be the estimate of the policy $\pi_Y^\beta$. At every $t$, we sample a data point $i \sim \rho(\cdot)$ and its prescribed cluster $j \sim \pi_t(\cdot|i)$. The data point associates itself to the cluster $k \sim p(\cdot|j, i)$ incurring a cost $d(x_i, y_k)$. We perform the following stochastic iteration to asynchronously update $q_t(x_i, y_j)$:

$$q_{t+1}(x_i, y_j) = (1 - \epsilon_{t,ij})q_t(x_i, y_j) + \epsilon_{t,ij}d(x_i, y_k), \tag{68}$$

and $q_{t+1}(x_{i'}, y_{j'}) = q_t(x_{i'}, y_{j'})$ for all $(i', j') \neq (i, j)$. These iterations, under the Robbins-Monro step-size conditions $\sum_t \epsilon_{t,ij} = \infty$, and $\sum_t \epsilon_{t,ij}^2 < \infty \; \forall \; i, j$, converge to the expected cost $d_{\mathrm{avg}}(x_i, y_j)$, and provide an estimate $\hat{\pi}_Y^\beta$ of the policy in (1). See (Borkar & Borkar, 2008) for proof.

*Learning $\{y_\ell\}$ (S2):* When the cost function $d(x_i, y_k)$ is known in the closed form and the local autonomy is not dependent on $\{y_\ell\}$, a straightforward way to learn the cluster representatives is via stochastic gradient descent (SGD). For instance, when $d(x_i, y_k) = \|x_i - y_k\|_2^2$ we execute the following SGD iterations

$$y_\ell(t + 1) = y_\ell(t) - \alpha_t\Big(\frac{1}{|\mathcal{S}|} \sum_{(i,j,k)\in\mathcal{S}} \big(y_\ell(t) - x_i\big)\delta_{\ell k}\Big), \tag{69}$$

where the mini-batch $\mathcal{S} = \{(i, j, k) : i \sim \rho, j \sim \hat{\pi}_Y^\beta, k \sim p\}$, and $\hat{\pi}_Y^\beta$ is the policy learnt in (S1).

## G  ADEN ARCHITECTURE

Given a set of data points $\mathcal{X}$ and representatives $\mathcal{Y}$, our attention-based deep neural network $\mathrm{NN}_\theta$ outputs the expected entity–cluster distance tensor

$$\bar{\mathbf{D}}_\theta(\mathcal{X}, \mathcal{Y}) \in \mathbb{R}_+^{|\mathcal{X}|\times|\mathcal{Y}|},$$

where $[\bar{\mathbf{D}}_\theta(\mathcal{X}, \mathcal{Y})]_{ij} = d_\theta(x_i, y_j)$. The design of $\mathrm{NN}_\theta$ allows inputs of variable sizes ($|\mathcal{X}|$ and $|\mathcal{Y}|$), enabling efficient transfer learning across problem instances without retraining from scratch. By training $d_\theta(x_i, y_j)$ to approximate a target distance function $d_{\mathrm{avg}}(x_i, y_j)$, the model implicitly encodes the influence of local autonomy $p(\cdot \mid j, i)$ on the full set of cluster representatives $\{y_\ell\}$, when such dependencies exist. This design ensures that gradients with respect to the cluster representatives are accurately propagated, allowing end-to-end optimization of both entity assignments and cluster representatives.

See Figure 5 for an architecture of our proposed Adaptive Distance Estimation Network (ADEN) that incorporates an internal Adaptive Distance Block (ADB). We employ a deep encoder to estimate autonomy-aware entity–cluster distances. Suppose there are $B$ mini-batch of data, each containing $S$ samples and all the $K$ clusters. We usually take $S \ll N$ to avoid computational overhead. The inputs are the data tensor $\mathbf{X} \in \mathbb{R}^{B\times S\times d}$ and the cluster tensor $\mathbf{Y} \in \mathbb{R}^{B\times K\times d}$. Both are first projected into a hidden space of dimension $d_h$ via

$$\hat{\mathbf{X}} = \mathcal{F}_X(\mathbf{X}), \hat{\mathbf{Y}} = \mathcal{F}_Y(\mathbf{Y}),$$

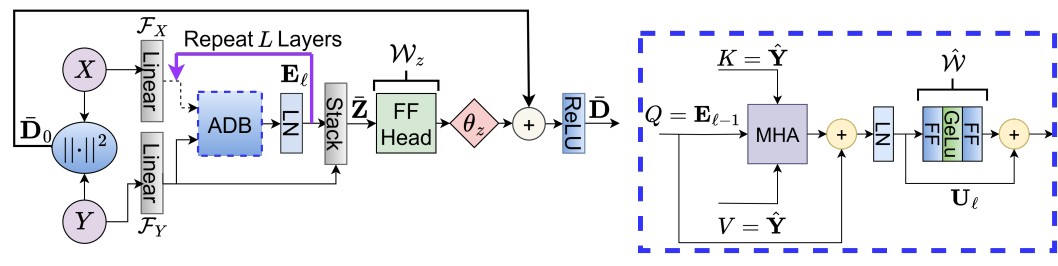

(a) Schematics of ADEN Architecture.    (b) ADB Module Used inside ADEN.

Figure 5: Overall Deep Architecture to predict autonomy-aware distances.

Table 2: Hyperparameters used for different scenarios.

| Parameter | Value |
|---|---|
| Seed | 0 |
| Hidden dim ($d_h$) | 64 |
| Feed Forward dim ($d_{FF}$) | 128 |
| ADB Layers ($L$) | 4 |
| Attention Heads | 8 |
| Batch Size ($B$) | 32 |
| Samples in Batch ($S$) | 128 |
| Learning Rate ($\eta_d$) | $10^{-4}$ |
| AdamW Weight Decay | $10^{-5}$ |
| Perturbation spread ($\sigma$) | 0.01 |
| Sampling size ($\hat{L}$) | 16 |
| EMA Filter rate ($\lambda$) | 0.95 |
| Epochs $Y$ ($T_y$) | 100 |
| Learning Rate $Y$ ($\eta_y$) | $10^{-4}$ |
| $\beta_{\min}$ | 10 |
| $\tau$ | 1.1 |

| **Hyperparameters with different values:** | | | |
|---|---|---|---|
| Epochs ADEN ($T_d$) | Fig. 1 Dataset: 1000 | — | Decentralized Sensing Fig 2: 2000 |
| $\beta_{\max}$ | Fig. 1 Dataset: 50,000 | — | Decentralized Sensing Fig 2: 10,000 |

where $\mathcal{F}_X, \mathcal{F}_Y : \mathbb{R}^d \to \mathbb{R}^{d_h}$ are learnable linear layers. Next, we apply $L$ layers of ADB encoding to $\hat{X}$ to obtain contextualized embeddings $\mathbf{E}_L \in \mathbb{R}^{B \times S \times d_h}$ (see Figure 5(b)):

$$\mathbf{U}_\ell = \mathrm{LN}\Big(\mathrm{MHA}\big(\mathbf{E}_{\ell-1}, \hat{\mathbf{Y}}, \hat{\mathbf{Y}}\big) + \mathbf{E}_{\ell-1}\Big), \tag{70}$$

$$\mathbf{E}_\ell = \mathrm{LN}\Big(\hat{\mathcal{W}}(\mathbf{U}_\ell) + \mathbf{U}_\ell\Big), \quad \ell = 1, \dots, L, \tag{71}$$

where $\mathbf{E}_0 = \hat{\mathbf{X}}$, MHA denotes standard multi-head attention (query, key, value), LN is layer normalization, and $\hat{\mathcal{W}} : \mathbb{R}^{d_h} \to \mathbb{R}^{d_{ff}} \to \mathbb{R}^{d_h}$ is a feed-forward module with expansion–contraction linear layers ($d_{ff} \gg d_h$) and GeLU activation. To form pairwise entity–cluster features, we broadcast the final point embeddings $\mathbf{E}_L \in \mathbb{R}^{B \times S \times d_h}$ and the cluster embeddings $\hat{\mathbf{Y}} \in \mathbb{R}^{B \times K \times d_h}$ across the cluster and entity dimensions, respectively:

$$\tilde{\mathbf{E}} = \mathbf{E}_L[:, :, \text{newaxis}, :] \in \mathbb{R}^{B \times S \times K \times d_h}, \qquad \tilde{\mathbf{Y}} = \hat{\mathbf{Y}}[:, \text{newaxis}, :, :] \in \mathbb{R}^{B \times S \times K \times d_h}.$$

We then pairwise-combine these representations by channel-wise stacking,

$$\bar{\mathbf{Z}} = [\,\tilde{\mathbf{E}} \| \tilde{\mathbf{Y}}\,] \in \mathbb{R}^{B \times S \times K \times 2d_h},$$

where $[\cdot \| \cdot]$ denotes feature-wise concatenation for each entity–cluster pair. A shared feed-forward "distance head" $\mathcal{W}_z : \mathbb{R}^{2d_h} \to \mathbb{R}$ then predicts autonomy-aware deviations:

$$\bar{\mathbf{D}} = \mathrm{ReLU}\big(\theta_z\,\mathcal{W}_z(\bar{\mathbf{Z}}) + \bar{\mathbf{D}}_0\big),$$

where $\theta_z$ is a learnable scalar, and $\bar{\mathbf{D}}_0$ is the baseline distance between $\mathbf{X}, \mathbf{Y}$. We define

$$\bar{\mathbf{D}}_0 = \big(\mathbf{X}[:,:,\text{newaxis},:] - \mathbf{Y}[:,\text{newaxis},:,:]\big)^{\odot 2}\mathbf{1}$$

as the batched, pairwise squared Euclidean distance between $\mathbf{X}$ and $\mathbf{Y}$, where $\odot 2$ denotes element-wise squaring and $\mathbf{1} \in \mathbb{R}^d$ is a vector of ones. Dropout layers within both the ADB blocks and the distance head mitigate overfitting. This design ensures permutation invariance across clusters, since $\mathcal{W}_z$ is applied identically to every entity–cluster pair.

**Complexity of one ADEN forward pass.** Let $d_h$ be the hidden dimension and $H$ the number of attention heads. The computational complexity is:

- **Projection layers:** $O(BSdd_h + BKdd_h)$.
- **ADB blocks:** Each multi-head attention in ADB processes queries of size $S$ and keys/values of size $K$, giving cost

$$O\big(L\,B\,H(SKd_h)\big),$$

which is linear in both $S$ and $K$.
- **Pairwise distance head:** After broadcasting, the pairwise tensor is of size $BSK \times 2d_h$, and the MLP head scales as

$$O\big(BSKd_h\big).$$

For the settings used in our experiments ($d_h = 64$, $H = 4$, $L = 2$, $S \leq 64$, $K \leq 50$), a full forward pass is on the order of *a few million* floating-point operations—well within the regime of small modern models (ADEN has $\approx$150k parameters). In practice, ADEN forward/backward passes constitute a small fraction of total runtime per RL iteration.

## H   ADEN TRAINING HYPERPARAMETERS

Table 2 provides the hyperparameters used for ADEN network in various simulations.

## I   SIMULATIONS WITH LOCAL AUTONOMY

We evaluate our method on the dataset shown in Fig. 1 under a controlled form of local autonomy. Specifically, each entity $i$ accepts its prescribed cluster $j$ with probability $1 - \kappa$; with probability $\kappa$ it selects an alternative cluster $k \neq j$ according to

$$p(k \mid j, i) = \kappa \frac{\exp[-c_k(j,i)/T]}{\sum_{t \neq j} \exp[-c_t(j,i)/T]}. \tag{72}$$

Here the cost $c_k(j,i) = \zeta\, d(y_j, y_k) + \bar{\gamma}\, d(x_i, y_k)$. We vary the parameters $\{\kappa, \bar{\gamma}, \zeta, T\}$ to study their influence on clustering outcomes. The results are shown in Fig. 6. In the visualizations, the color of each entity reflects its mixture of representative assignments, computed as a linear combination of representative colors weighted by $\pi_Y(j \mid i)$.

When $\kappa$ is small (Fig. 6(a1)-(a2)), cluster representatives remain close to their assigned entities. As $\kappa$ increases, deviations grow (Fig. 6(a3)-(d4)): the temperature $T$ controls the randomness of alternative selections—higher $T$ pulls all representatives toward a common location (Fig. 6(c1)-(c2)), whereas lower $T$ draws each representative toward the nearest entities (Fig. 6(a1) and 6(b1)). In the high-$\kappa$ regime (Fig. 6(d1)-(d4)), small $T$ produces pronounced shifts in which representatives migrate to the closest clusters, yielding nontrivial configurations (Fig. 6(d1), 6(d3), and 6(d4)). Note how the representative on top of a cluster is not of the same colour as the data points in that cluster — highlighting the non-trivial configurations under high $\kappa = 0.9$. The parameters $\bar{\gamma}$ and $\zeta$ further modulate assignment patterns, altering how entities distribute across representatives.

**UDT19 Dataset:** The UDT19 London Traffic dataset is curated from Transport for London (TfL) traffic sensor feeds and processed by (Loder et al., 2019) It contains **5,719 traffic sensors** distributed across London, each providing multiple spatio-temporal attributes such as geographic coordinates, average speed, flow, and occupancy. For our clustering and UAV-placement experiments, we use the

Table 3: UDT19 Dataset - $D$ gap (%) of the ADEN versus the setting that ignores local autonomy $p(k|j,i)$, relative to the ground truth (Algorithm 1) across scenarios where $p(k|j,i)$ depends on $Y$.

| $\kappa$ | $\bar{\gamma}$ | $\zeta$ | $T$ | ADEN |
|---|---|---|---|---|
| 0.1 | 0 | 1 | 0.1 | 0 |
| 0.5 | 0 | 1 | 0.1 | 10 (**improvement**) |

| $\kappa$ | $\bar{\gamma}$ | $\zeta$ | $T$ | ADEN |
|---|---|---|---|---|
| 0.1 | 0 | 1 | 0.01 | 18.37 |
| 0.5 | 0 | 1 | 0.01 | 24.82 |

**GPS coordinates**, which are the only relevant features for the sensing network design. We consider 4 different autonomy models, that are obtained by varying $\kappa, T$ in the softmax autonomy model discussed in the beginning of the Section 5. The Table 3 summarizes the details of the simulations on UDT19 Dataset. Give details about the dataset here UDT19 London Traffic Dataset.

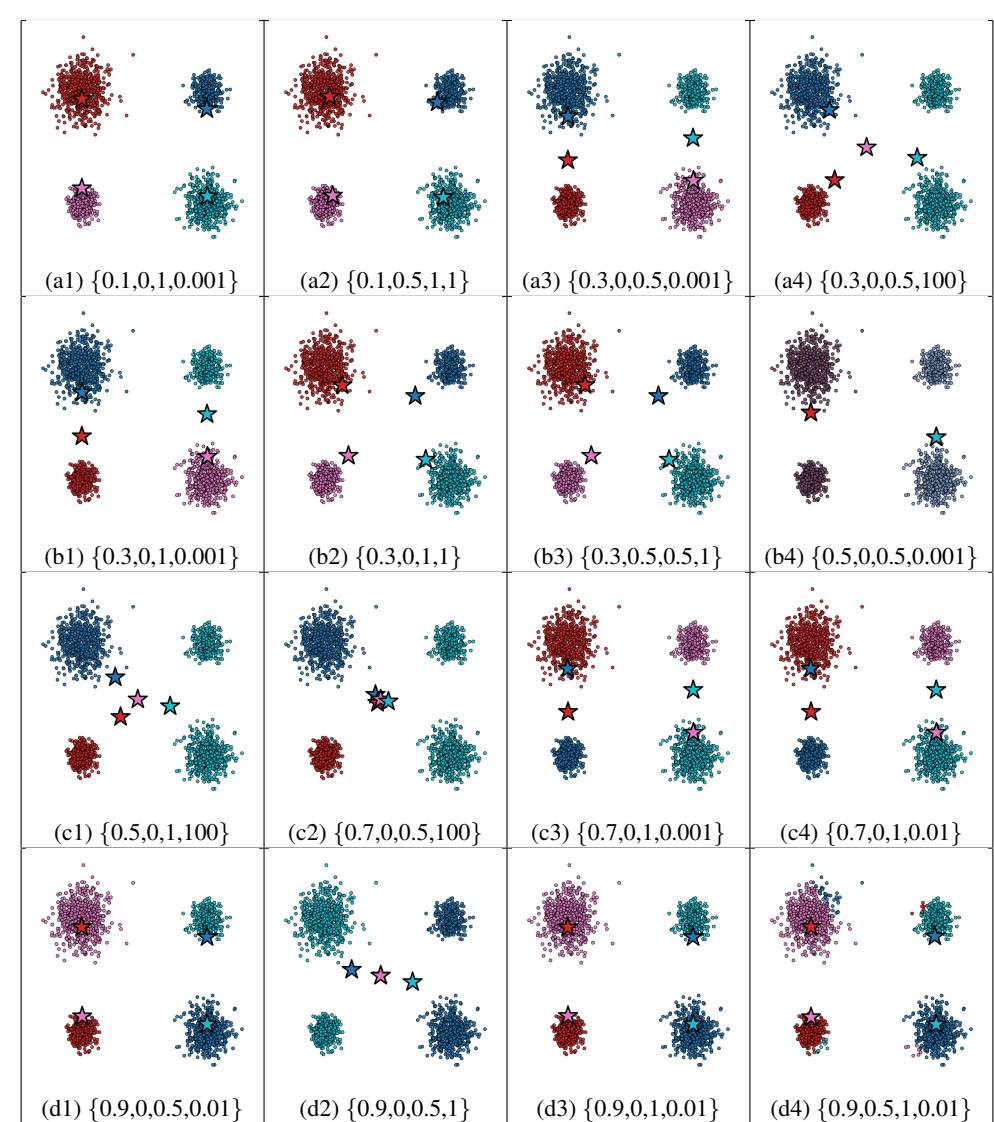

Figure 6: 4×4 grid of benchmark images. Each subcaption shows the tuple $\{\kappa, \bar{\gamma}, \zeta, T\}$ in that order.

**Movie Lens 1M Dataset:** We also conducted 40 additional experiment on a real-world dataset, called MovieLens-1M (Harper & Konstan, 2015). This dataset is a widely used benchmark in recommendation systems, containing $1,000,209$ anonymous ratings given by $6,040$ users to approximately $3,900$ movies. The ratings are made on a 5-star scale (whole-star ratings only). The data is primarily structured around two files: one for the ratings, including the UserID, MovieID, Rating, and Timestamp (with a guarantee that each user has at least 20 ratings); and another for the movie

Table 4: Benchmark Results for MovieLens-1M Dataset per $\{\kappa, \bar{\gamma}, \zeta, T\}$ Combination of Local Autonomy (Sorted by $\kappa$, Ascending).

| $\kappa$ | $\bar{\gamma}$ | $\zeta$ | $T$ | Err$_\text{ADEN}$ | Err$_\text{IGN}$ |
|---|---|---|---|---|---|
| 0.1 | 1.0 | 0.0 | 1.0 | -0.42 | 3.29 |
| 0.1 | 0.5 | 0.5 | 1.0 | -0.41 | 3.37 |
| 0.1 | 0.5 | 0.5 | 0.01 | 5.52 | 0.24 |
| 0.1 | 0.75 | 0.25 | 1.0 | -0.33 | 3.33 |
| 0.2 | 1.0 | 0.0 | 1.0 | 2.95 | 14.48 |
| 0.2 | 0.5 | 0.5 | 1.0 | 1.77 | 14.65 |
| 0.2 | 0.75 | 0.25 | 1.0 | 1.28 | 14.57 |
| 0.2 | 0.75 | 0.25 | 0.01 | 5.88 | 1.49 |
| 0.3 | 0.75 | 0.25 | 1.0 | 1.71 | 25.83 |
| 0.3 | 0.5 | 0.5 | 1.0 | 1.76 | 25.95 |
| 0.3 | 1.0 | 0.0 | 1.0 | 2.59 | 25.71 |
| 0.3 | 1.0 | 0.0 | 0.01 | 3.96 | 2.52 |
| 0.4 | 0.5 | 0.5 | 0.01 | 2.85 | 5.85 |
| 0.4 | 1.0 | 0.0 | 1.0 | 2.06 | 37.0 |
| 0.4 | 0.75 | 0.25 | 1.0 | 1.8 | 37.17 |
| 0.4 | 0.5 | 0.5 | 1.0 | 1.7 | 37.34 |
| 0.5 | 0.75 | 0.25 | 1.0 | 1.82 | 48.56 |
| 0.5 | 0.5 | 0.5 | 1.0 | 1.67 | 48.79 |
| 0.5 | 0.5 | 0.5 | 0.01 | 1.83 | 13.81 |
| 0.5 | 1.0 | 0.0 | 0.01 | 1.53 | 10.23 |
| 0.5 | 0.75 | 0.25 | 0.01 | 3.91 | 11.59 |
| 0.5 | 1.0 | 0.0 | 1.0 | 1.59 | 48.34 |
| 0.6 | 1.0 | 0.0 | 1.0 | 1.82 | 60.56 |
| 0.6 | 0.75 | 0.25 | 1.0 | 1.83 | 60.84 |
| 0.6 | 0.5 | 0.5 | 1.0 | 1.71 | 61.12 |
| 0.6 | 1.0 | 0.0 | 0.01 | 3.25 | 22.55 |
| 0.6 | 0.75 | 0.25 | 0.01 | 2.18 | 21.95 |
| 0.6 | 0.5 | 0.5 | 0.01 | 2.81 | 23.76 |
| 0.7 | 1.0 | 0.0 | 0.01 | 3.02 | 37.61 |
| 0.7 | 0.75 | 0.25 | 0.01 | 3.66 | 37.02 |
| 0.7 | 1.0 | 0.0 | 1.0 | 1.4 | 73.5 |
| 0.7 | 0.75 | 0.25 | 1.0 | 1.47 | 73.83 |
| 0.7 | 0.5 | 0.5 | 1.0 | 1.19 | 74.17 |
| 0.7 | 0.5 | 0.5 | 0.01 | 3.39 | 36.65 |
| 0.8 | 0.5 | 0.5 | 0.01 | 6.24 | 52.45 |
| 0.8 | 1.0 | 0.0 | 0.01 | 10.05 | 59.47 |
| 0.8 | 0.75 | 0.25 | 0.01 | 9.96 | 59.79 |
| 0.8 | 1.0 | 0.0 | 1.0 | 0.67 | 87.52 |
| 0.8 | 0.75 | 0.25 | 1.0 | 0.65 | 87.92 |
| 0.8 | 0.5 | 0.5 | 1.0 | 0.73 | 88.31 |

Table 5: Average performance across six distinct scenarios characterized by $\kappa$ and $T$ values.

| Autonomy Strength | Entropy level | Corresponding values of $\kappa$ and $T$ | Err$_\text{ADEN}$ (average) | Err$_\text{IGN}$ (average) | Difference |
|---|---|---|---|---|---|
| High | High | $\kappa \in \{0.6, 0.7, 0.8\}, T = 1.0$ | 1.27% | 74.20 % | 72.92% |
| High | Low | $\kappa \in \{0.6, 0.7, 0.8\}, T = 0.01$ | 4.95% | 39.03% | 34.08% |
| Moderate | High | $\kappa \in \{0.4, 0.5\}, T = 1.0$ | 1.77% | 42.87% | 41.10% |
| Moderate | Low | $\kappa \in \{0.4, 0.5\}, T = 0.01$ | 2.53% | 10.37% | 7.84% |
| Low | High | $\kappa \in \{0.1, 0.2, 0.3\}, T = 1.0$ | 1.21% | 14.58% | 13.36% |
| Low | Low | $\kappa \in \{0.1, 0.2, 0.3\}, T = 0.01$ | 5.12% | 1.42% | -3.70% |

metadata, which provides the title and pipe-separated genres selected from a list of 18 categories, such as Action, Drama, and Sci-Fi.

We created a feature vector for each user based on the weighted average of their ratings applied to the feature vectors of the movies they have rated. After obtaining an *18-dimensional feature vector* for each user (corresponding to the 18 MovieLens genres), we applied PCA (principal component analysis) to reduce this to a *10-dimensional vector* (preserving $\sim 95\%$ cumulative variance) for downstream task of demonstrating autonomy aware clustering simulations.

*Benefits of Autonomy-Aware Clustering for Movie Recommendation:* Clustering-based recommendation is a well-established direction in the literature (Ungar & Foster, 1998; Jannach et al., 2010), but existing approaches do not explicitly model and capture complex behavioral variability (autonomy) in user choices. In the following simulations, we construct a straightforward clustering-based movie recommendation system to illustrate the effect of user autonomy and the performance of Algorithm 2 when autonomy information is unavailable. The pipeline operates as follows: users are clustered based on their feature vectors $\{x_i\}$ together with autonomy effects encoded through $p(k|j, i)$, and for each user we recommend movies from a repository by selecting those closest (in feature space) to the representative of their assigned cluster. We demonstrate that (i) ignoring autonomy when forming clusters can lead to substantial errors (equivalently, poor recommendations), and (ii) Algorithm 2 successfully tracks the autonomy-aware clustering solution even without explicit knowledge of the autonomy model.

A further benefit of autonomy-aware clustering arises from the cluster representative vectors $\{y_k\}$. These representatives summarize dominant preference profiles within the learned clusters and—as illustrated in Figure 1—can shift significantly when autonomy is taken into account. Because these

representatives encode aggregate patterns of user taste, they serve as informative prototypes for downstream tasks such as informing movie design or aiding in the generation of future content that aligns with both the observed feature structure and the behavioral autonomy of user groups. In particular, Algorithm 2 yields these representative vectors while implicitly accounting for autonomy, even when the autonomy model is not explicitly provided. This allows the resulting cluster prototypes to accurately reflect both the observed user features and their latent behavioral tendencies, thereby supporting more reliable recommendation and content generation pipelines.

For the purpose of simulation, we cluster the $6040$ users (with 10-dimensional features as described above) into 10 clusters. We consider the autonomy model mathematically quantified in equation (72), similar to our previous simulations. Choosing different combinations of the parameters $\{\kappa, \bar{\gamma}, \zeta, T\}$ provides a comprehensive way to model a large variation of user behaviors when they are recommended a movie:

1. $\kappa$ (autonomy strength): Controls the tendency to follow the central policy. A small $\kappa$ suggests a user that mostly accepts the recommended movie, whereas a large $\kappa$ indicates a user that frequently deviates from the recommendation.

2. $\bar{\gamma}$ and $\zeta$ (deviation preference): Determine the preference when a user decides to deviate. A high $\bar{\gamma}$ suggests the user prefers to pick an alternative movie (from another cluster $k \neq j$, where $j$ is the prescribed cluster) that has a feature vector $y_k$ closer to their own feature vector $x_i$ (self-alignment). Similarly, a large $\zeta$ shifts this preference towards choosing a movie (from another cluster $k \neq j$, where $j$ is the prescribed cluster) that has a feature vector $y_k$ closer to the original feature vector $y_j$ of the prescribed cluster (system-alignment).

3. $T$ (entropy level): Quantifies the strictness of the deviation preference. At larger $T$ (higher temperature) the preference dictated by $\bar{\gamma}$ and $\zeta$ become weak, and the user is more likely to choose an alternative movie more randomly (high entropy). Conversely, at lower $T$ the preference dictated by $\bar{\gamma}$ and $\zeta$ become stronger, and the user preferences are less random (low entropy).

The results for 40 different combinations of $\{\kappa, \bar{\gamma}, \zeta, T\}$, modeling various user autonomy profiles, are presented in Table 4. The Err$_{\text{ADEN}}$ column reports the $\%$ gap in $D$ (P2) achieved by our proposed (ADEN/RL) Algorithm 2), while Err$_{\text{IGN}}$ column shows the $\%$ gap in $D$ observed when autonomy is ignored. Note that the both the $\%$ gaps are quantified with respect to the model-based approach (Algorithm 1), i.e., when the explicit knowledge of autonomy is available (ground truth).

Note that the average errors across all the scenarios in Table 4 are Err$_{\text{ADEN}}$ = $2.81\%$ Err$_{\text{IGN}}$ = $30.41\%$. This highlight two important aspects: (i) the cost of ignoring autonomy is exceptionally high $\approx 30.41\%$ deviation from the ground truth, and (ii) even with no explicit information about autonomy, our proposed Algorithm 2 performs well ($\approx 2.81\%$ deviation) and tracks the quality of solutions determined when the autonomy levels are explicitly known (ground truth).

To further summarize the findings from Table 4, we categorize the simulated autonomy behaviors into six distinct groups dictated by the values of $\kappa$ (autonomy strength) and $T$ (entropy level). In particular, we consider the following three autonomy strength ranges: (i) high $\kappa \in \{0.6, 0.7, 0.7\}$, (ii) moderate $\kappa \in \{0.4, 0.5\}$, and (iii) low $\kappa \in \{0.1, 0.2, 0.3\}$, and two entropy levels: (i) high $T = 1.0$ and (ii) low $T = 0.01$. Please see the Table 5 for details.

The data rows in Table 5 are arranged in the decreasing order of autonomy levels, i.e., the first row captures an extremely high autonomy level and the last row demonstrates a low autonomy level. Observe that across the entire spectrum of autonomy level, the Algorithm 2 performs consistently well — demonstrated by the low average Err$_{\text{ADEN}}$ ($\lesssim 5\%$). On the other hand, ignoring autonomy can be catastrophic as demonstrated by average Err$_{\text{IGN}}$ (as large as $74.20\%$). The above two observations, reinforce that efficacy of our proposed Algorithm 2 in the absence of autonomy information, at the same time, the perils of ignoring autonomy. Note that for the last data row, the Err$_{\text{IGN}}$ ($1.42\%$) is better than Err$_{\text{ADEN}}$ ($5.14\%$). This could be due to the fact that autonomy levels are quite low, and thus, ignoring it completely does not result into significant errors. However, this is not a general observation across all our simulation. For instance, in Table 1, across low autonomy strength $\kappa \in \{0.1, 0.2, 0.3\}$ and low entropy level $T = 0.01$, the $\%$ gap in $D$ is consistently low for the Algorithm 2 in comparison to the case where autonomy is entirely ignored.

## J  USE OF LARGE LANGUAGE MODELS

In preparing this manuscript, we made limited use of OpenAI's ChatGPT. Specifically, ChatGPT was employed to improve grammar, clarity, and readability of the text. On rare occasions, it was also used to aid in literature discovery; however, all references and citations included in the paper were independently verified and sourced directly from Google Scholar.

## K  STABILITY AND SENSITIVITY OF ADEN TRAINING

A central concern for any learning-based clustering method is the stability of training under changes in hyperparameters, initialization schemes, and environment parameters. To evaluate this, we conducted a comprehensive 32-way sweep over (i) environment parameters $(\kappa, \bar{\gamma}, \zeta, T)$, (ii) ADEN architectural choices $(d_{\mathrm{model}}, n_{\mathrm{layers}}, n_{\mathrm{heads}}, d_{\mathrm{ff}})$, (iii) training hyperparameters (learning rates, batch sizes, batch sampling strategy), and (iv) initialization protocols for the cluster centers $Y$ ("mean_noise" vs. "sample"). Each configuration was repeated over two random seeds, resulting in a total of 32 independent training runs.

**Overall stability.**  Out of the 32 experiments, *28 runs* exhibited highly stable training dynamics for both the ADEN prediction loss and the free-energy objective. In these runs, the ADEN loss decreased smoothly and monotonically, and the free-energy $F_\beta$ decreased consistently across all four tested inverse temperatures $\beta \in \{10, 100, 1000, 10000\}$. These results demonstrate that ADEN is remarkably robust to hyperparameter variations and initialization choices, and that the alternating optimization of $(d_{\mathrm{avg}}, Y)$ proceeds in a stable fashion across a wide range of regimes. Figures 7 and 8 show the loss and free-energy trajectories for all 28 stable experiments, aggregated over the four values of $\beta$.

A key reason for the robustness of ADEN training is that the optimization pipeline is tightly structured around the maximum entropy principle (MEP). As detailed in Algorithm 2, ADEN is trained across a sequence of entropy levels indexed by $\beta$. At $\beta \approx 0$, the free-energy landscape is extremely smooth because all effective distances $d_{\mathrm{avg}}(x_i, y_j)$ collapse to nearly the same scale. This makes the regression task trivial regardless of initialization of model weights, cluster representatives, or optimizer states. As $\beta$ is gradually increased, the targets become progressively more informative, but at every stage the model is warm-started with the parameters learned at the previous entropy level. This continuation scheme acts as a homotopy method: it avoids the abrupt introduction of sharp gradients, prevents large optimization shocks, and ensures that learning proceeds along a stable trajectory even under different random seeds, weight initializations, or environment parameters. The result is that ADEN reliably converges in 28 out of 32 tested configurations—including widely varying choices of initialization strategy, model depth, learning rate, and autonomy level—demonstrating that the stability of training is not accidental but an inherent consequence of the MEP-guided annealing process.

Only 4 out of the 32 runs displayed a qualitatively different pattern: the ADEN training loss remained approximately constant, and the free energy plateaued without meaningful improvement. These anomalous runs all correspond to the *same* configuration:

(i) $d_{\mathrm{model}} = 128$, $n_{\mathrm{layers}} = 6$, $n_{\mathrm{heads}} = 8$, $d_{\mathrm{ff}} = 256$, (ii) init = `mean_noise`, (iii) lr = $5 \times 10^{-4}$,

combined with *both* values of the autonomy strength $\kappa \in \{0.2, 0.5\}$ and *both* random seeds. The corresponding experiment identifiers are:

- `IDX4_M4_kappa0.2_seed0_D128_L6_H8_FF256_lr5e-4_initmean_noise`,
- `IDX4_M4_kappa0.2_seed1_D128_L6_H8_FF256_lr5e-4_initmean_noise`,
- `IDX4_M4_kappa0.5_seed0_D128_L6_H8_FF256_lr5e-4_initmean_noise`,
- `IDX4_M4_kappa0.5_seed1_D128_L6_H8_FF256_lr5e-4_initmean_noise`.

Figure 9 show the loss trajectories for these four runs, plotted for the two relevant $\beta$ values.

The behavior of these four cases is fully consistent with the structure of the autonomy model and the maximum-entropy formulation. Two effects explain the flat-loss phenomenon:

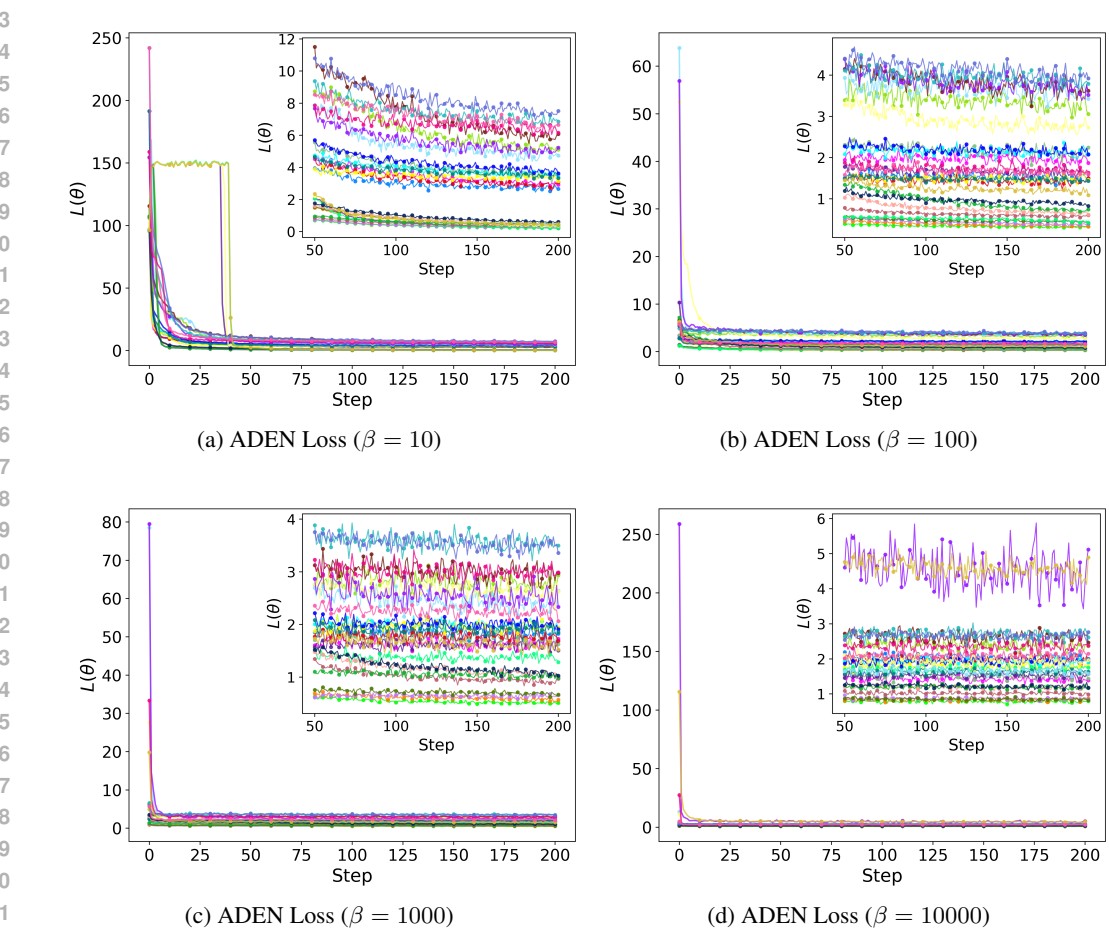

Figure 7: Loss trajectories for all 28 stable experiments, aggregated into four representative runs across the four values of $\beta$.

1. **High autonomy strength $\kappa$ reduces the informativeness of the central policy signals.** Larger $\kappa$ values correspond to stronger local autonomy and therefore larger systematic deviations from the soft assignments produced by the central planner. When ADEN attempts to learn $d_{\mathrm{avg}}(x_i, y_j)$ under such conditions, the observed entity–cluster outcomes become significantly more stochastic and less responsive to variations in $Y$. This directly reduces the signal-to-noise ratio in the gradients that shape both the ADEN training loss and the free-energy objective. When combined with high-capacity models (128d, 6 layers, 8 heads) and the `mean_noise` initialization, the system can enter a regime where the observed deviations are so dominated by autonomy effects that the RL signal contains insufficient structure to guide further improvement. The plateau therefore corresponds to a high-$\kappa$ *noise-dominated regime*.

2. **High learning rate + high-capacity ADEN + "mean_noise" initialization suppresses effective gradient signal at the start of training.** Unlike the other 28 experiments, the 4 anomalous runs exhibit *no meaningful decrease* in the ADEN loss from the very first epochs—the loss remains essentially identical to its initial value. This indicates not premature convergence, but rather a failure of the optimization dynamics to extract a usable gradient signal.

   All four problematic cases combine three factors simultaneously: (i) the largest ADEN architecture tested ($d_{\mathrm{model}}$=128, $n_{\mathrm{layers}}$=6, $n_{\mathrm{heads}}$=8, $d_{\mathrm{ff}}$=256), (ii) the higher learning rate $5 \times 10^{-4}$, and (iii) the `mean_noise` initialization of cluster centers. Under this combination, the model begins training in a regime where the autonomy-driven variability (controlled by $\kappa$) is large relative to the gradient signal induced by changes in $Y$ and ADEN's parameters. The high-capacity network rapidly absorbs the random structure of the initial `mean_noise` embeddings, while the

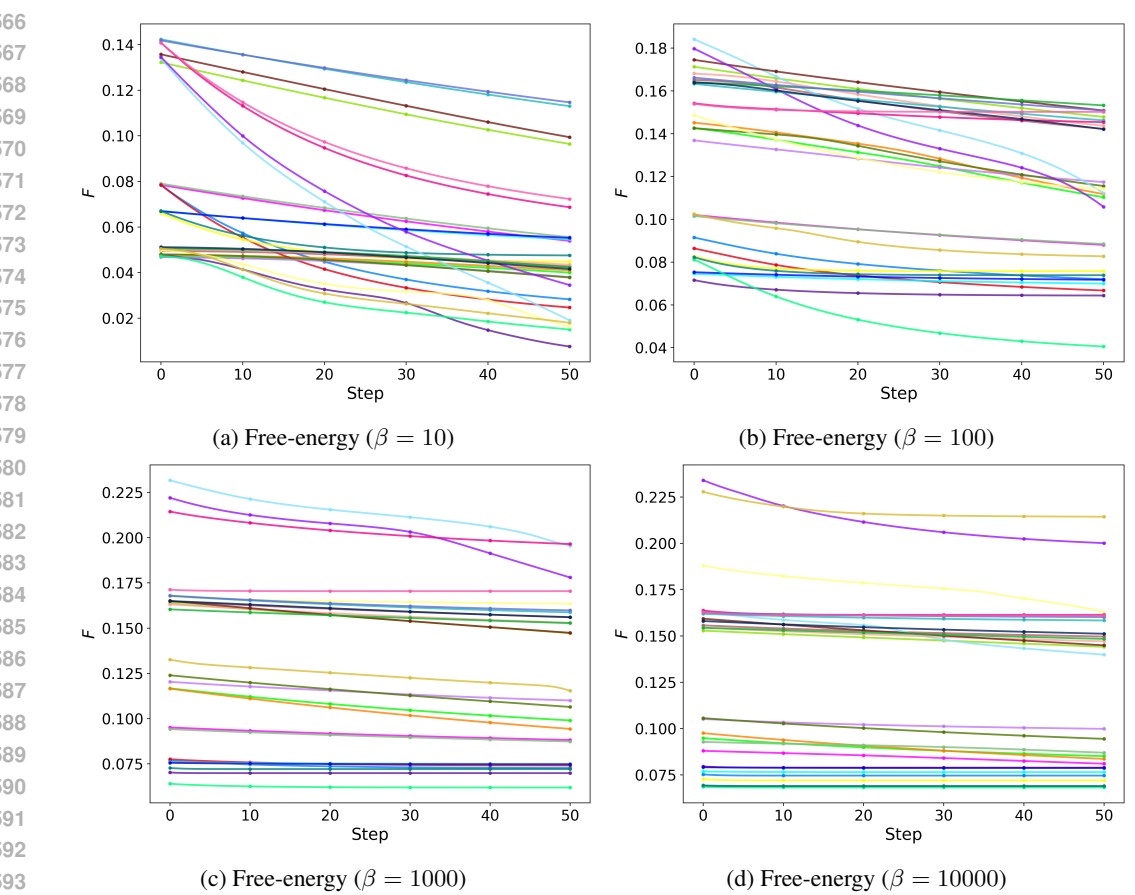

(a) Free-energy ($\beta = 10$)    (b) Free-energy ($\beta = 100$)

(c) Free-energy ($\beta = 1000$)    (d) Free-energy ($\beta = 10000$)

Figure 8: Free-energy trajectories for all 28 stable experiments, aggregated into four representative runs across the four values of $\beta$.

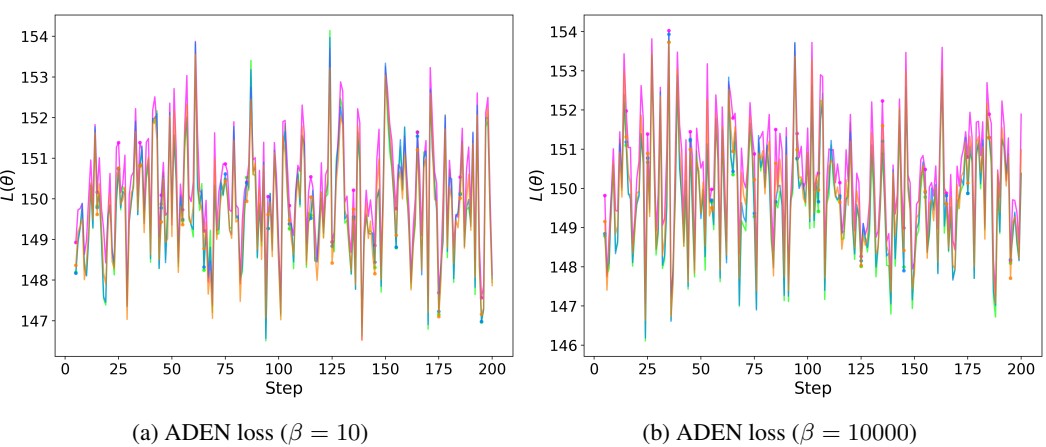

(a) ADEN loss ($\beta = 10$)    (b) ADEN loss ($\beta = 10000$)

Figure 9: Loss trajectories for the four flat (plateauing) experiments, shown for low and high values of $\beta$.

elevated learning rate destabilizes the early optimization steps, effectively preventing the model from entering the regime where the entropy-annealing procedure reveals informative gradients. Thus the gradients remain extremely small or incoherent throughout the early stages, causing the ADEN loss and free energy to appear "frozen."

Crucially, this behavior is *not* observed in any of the other 28 conditions, including runs with the same environment settings but different learning rates or initialization schemes. This strongly indicates that the flat-loss behavior arises from the interaction of (model size, learning rate, initialization), rather than from any inherent instability of ADEN or the underlying MEP formulation.

In conclusion, across a 32-run sweep, ADEN training is stable, reproducible, and insensitive to the majority of hyperparameter and initialization variations. The four constant-loss cases arise from interpretable interactions between autonomy strength, learning-rate, and model capacity, and they highlight that the ADEN objective behaves predictably even in autonomy-dominated regimes.

## L   NOTATION SUMMARY

| Symbol | Description |
|---|---|
| $N$ | Number of entities (data points). |
| $K$ | Number of clusters. |
| $\mathcal{X} = \{x_i\}$ | Entity feature vector set; typically $x_i \in \mathbb{R}^d$. |
| $\mathcal{Y} = \{y_j\}$ | Cluster representatives set (facility/UAV locations), $y_j \in \mathbb{R}^d$. |
| $Y \in \mathbb{R}^{Kd}$ | $Y = [y_1^\top \ y_2^\top \ \ldots \ y_K^\top]^\top$ vector of all representatives |
| $\rho(i)$ | Normalized importance weight of entity $i$ ($\rho(i) \geq 0$, $\sum_i \rho(i) = 1$). |
| $\mu(j\|i)$ | Hard assignment of entity $i$ to cluster $j$; takes values in $\{0, 1\}$. |
| $\pi(j\|i)$ | Soft-assignment probability for entity $i$ choosing cluster $j$. |
| $\Lambda$ | Feasible set of hard assignments $\mu$. |
| $\Lambda_\beta$ | Feasible set of soft assignments $\pi$. |
| $p(k\|j, i)$ | Local autonomy distribution: probability entity $i$ associates with $k$ when prescribed $j$. |
| $d(x_i, y_j)$ | Dissimilarity cost / cost of association between entity $i$ and cluster $j$. |
| $d_{\mathrm{avg}}(x_i, y_j)$ | Autonomy-aware average association cost. |
| $D$ | Objective in autonomy-aware clustering. |
| $H$ | Entropy of the soft-assignment distribution. |
| $\beta$ | Annealing parameter governing softness of assignments. |
| $\tau$ | Multiplicative annealing factor used to update $\beta$. |
| $F(Y)$ | Free-energy objective (DA formulation). |
| $\pi_Y^\beta(j \mid i)$ | Gibbs distribution / optimal assignment policy induced by $Y$ and $\beta$. |
| $\beta_{\mathrm{cr}}$ | Critical value of $\beta$ at which phase transition occurs. |
| $P_{\pi\rho}^Y$ | Diagonal matrix of effective cluster masses. |
| $\nabla_Y F$ | Gradient of the free-energy function w.r.t. cluster representatives. |
| $\mathbb{I}_d$ | Identity matrix of size $d \times d$ |
| $Q(s, a)$ | Q-value of a state–action pair in the MDP. |
| $\gamma$ | Discount factor; fixed to $\gamma = 0$ (unit horizon). |
| $d_\theta(x_i, y_j)$ | ADEN approximation of autonomy-aware average cost. |
| $\theta$ | Trainable weights of the ADEN network. |
| $L(\theta)$ | Regression loss used to train ADEN. |
| ADEN | Adaptive Distance Estimation Network. |
| $\epsilon$-greedy | Exploration rule used to sample cluster assignments in Algorithm 2. |
| $\eta_y$ | Learning rate for gradient steps in ADEN training. |
| $\tilde{y}_j$ | Perturbed cluster representative $y_j$. |
| $\kappa, \bar{\gamma}, \zeta, T$ | Parameters of softmax autonomy model in Section 5. |

