# OpenReview forum: "Autonomy-Aware Clustering: When Local Decisions Supersede Global Prescriptions"
_ICLR.cc/2026/Conference — Submitted to ICLR 2026_

### Official Review · Reviewer_ggqe · 2025-10-21

**Soundness:** 2
**Presentation:** 2
**Contribution:** 3
**Rating:** 4
**Confidence:** 3

**Summary:**

This paper presents the idea of autonomy-aware clustering to tackle real-world scenarios where entities deviate from the prescribed cluster assignments. By integrating reinforcement learning with deterministic annealing, the proposed method captures these autonomy effects, leading to satisfactory clustering results on the London Traffic dataset.

**Strengths:**

1. This paper proposes the entity autonomy problem in the existing passive clustering frameworks, which seems to be a practical research direction.
2. The proposed method is theoretically grounded, with detailed mathematical derivations and proofs.

**Weaknesses:**

1. In Fig. 1, it is not clear how such autonomy would occur in existing clustering methods. For example, for the given data, the classic K-means algorithm would not achieve such degraded results as illustrated in Figures c and d. I feel that a more intuitive example and explanation would help the readers to understand that the proposed autonomy problem is realistic and commonly encountered.
2. How could the proposed Adaptive Distance Estimation Network estimate the behavior of instances? In the fully supervised case, how can such estimations reflect real autonomy rather than random guesses? According to the experiments, the methods with or without prior autonomy knowledge only have a performance gap of 3-4%, which is a bit surprising.
3. In fact, such a result may indicate that the dataset used for evaluation is not representative and complicated enough. It is not convincing whether the method could generalize to other scenarios, such as social networks and recommender systems, especially considering the hyperparameters required.
4. In Eq. P1, $\rho(i)$ is not explained.

**Questions:**

My major concerns lie in two aspects: i) more examples and discussions are needed to prove that such an autonomy problem widely exists in real-world applications; ii) it is a bit unreasonable that the model could produce results close to ground truth even without any priors on the entity autonomy. In other words, one could change the "ground truth" and the model would produce the same but "much worse" results. These concerns need to be addressed.

---

> ### Author Response · Authors · 2025-11-21
>
> Thank you for your thoughtful summary and positive evaluation of our work. We truly appreciate the recognition of the practical relevance of autonomy-aware clustering and the acknowledgment of the theoretical grounding and methodological contributions in our approach. Below we address your questions and concerns
>
> *"**W:** In Fig. 1, it is not clear how such autonomy would occur in existing clustering methods. For example, for the given data, the classic K-means algorithm would not achieve such degraded results as illustrated in Figures c and d. I feel that a more intuitive example and explanation would help the readers to understand that the proposed autonomy problem is realistic and commonly encountered. **Q:** More examples and discussions are needed to prove that such an autonomy problem widely exists in real-world applications"*
>
> **A:** Thank you for the comment. It touches the core implication of autonomy on clustering outcomes. The solutions shown in Fig. 1(c)-(d) are in fact the **optimal** clustering solutions in the presence of local autonomy, not degraded ones. Recall that a clustering solution consists of the pair $\{y_j\}$ (cluster representatives) together with the assignment policy $\pi(j|i)$. The shift in the cluster representatives $y_j$ relative to Fig. 1(a) arises because the algorithm now accounts for the latent autonomy $p(k | j,i)$, which encodes the tendency of each data point to associate with clusters other than the one prescribed by $\pi$. In this example, several data points already exhibit a non-negligible probability of deviating from their prescribed cluster. As a result, the optimal solution rearranges the representatives $y_j$ to reduce the average autonomy-aware association cost $d_{\text{avg}}(x_i, y_j)$ defined in (P2). This restructuring is therefore the **correct optimal response** of the clustering system once autonomy is taken into account.
>
> To illustrate this further consider the following stylized example. Consider clustering data points into clusters A and B. Suppose that, whenever the policy assigns a data point to cluster A, it overrides that prescription and chooses cluster B instead, and whenever the policy assigns a data point to cluster B, it autonomously chooses cluster A instead. Under such autonomy, the ** optimal ** cluster representative $y_{\text{B}}$ will lie at the center of cluster A, and the **optimal** cluster representative $y_{\text{A}}$ will lie at the center of cluster B. Though this is counter intuitive from the perspective of classical clustering, it minimizes the true objective (P2) under autonomous deviations and achieves substantially lower cost than methods like $k$-means, which do not account for such behavior. Our intention in Fig. 1 was to highlight precisely this phenomenon: once autonomy is present, the optimal solution may differ drastically from the classical one.
>
> Thank you for the suggestion to include more intuitive explanation for the reader. We have revised the manuscript to include more details on the source of local autonomy and its implications on the design of communication network in decentralized sensing in Introduction Section 1 (highlighted in blue ink), and we have added Appendix A (with illustrative diagram and text) that explains the same. We believe all this along with existing details on local autonomy and its implications in our Introduction section will give reader a more intuitive understanding of problem we address in our work.
>
> ---
> *"**W:** In Eq. P1,  is not explained."*
>
> **A:** Thank you for the comment. In the Introduction, when presenting (P1), we intentionally omitted some notation to avoid disrupting the conceptual build-up to autonomy; the full definition of $\rho(i)$ was provided later in Section 3, where the formal problem formulation appears.
>
> Thank you for bringing out the unintended confusion this might cause. We have clarified this in the revised manuscript (marked in blue ink in Section 1). Specifically, $\rho(i)$ denotes the relative weight or importance of the $i$-th entity (data point) in the clustering objective—e.g., reflecting its sampling frequency, reliability, or domain-specific relevance—and ensures that entities contribute proportionally to the overall cost.

---

> ### Author Response · Authors · 2025-11-21
>
> *"**W:** How could the proposed Adaptive Distance Estimation Network estimate the behavior of instances? In the fully supervised case, how can such estimations reflect real autonomy rather than random guesses? According to the experiments, the methods with or without prior autonomy knowledge only have a performance gap of 3-4%, which is a bit surprising."*
>
> Thank you for your question. We would like to expand on two points regarding the role of the Adaptive Distance Estimation Network (ADEN) and the interpretation of behavior and random guesses.
>
> **1. ADEN is not a supervised-learning estimator of instance behavior** - Our paper does not introduce any supervised scheme for learning local autonomy. When the local autonomy model $p(k |j,, i)$ is known, no learning is required: the optimal policy and cluster representatives follow directly from the closed-form expressions in Eqs. (1) and (2) (or, Algorithm 1).
>
> The ADEN module is used only when the autonomy model is *unknown*, and its purpose is to learn average association cost $d_{\text{avg}}(x_i,y_j):=p\sum_k (k | j, i)d(x_i,y_j)$ by interacting with the environment in a reinforcement learning setting. Concretely, ADEN observes the *realized instantaneous distances* $d(x_i, y_k)$ that occur when the central policy assigns cluster $j$ to data point $i$ and the data point samples $k\sim p(\cdot|j,i)$ from its local autonomy, and learns $d_{\text{avg}}(x_i,y_j)$ from these observations. Note that this is analogous to how $Q$-functions are learned in deep RL methods. The model learned by ADEN captures the actual deviation patterns (autonomy) of the agents, rather than arbitrary or random behavior. Once ADEN has learned the average cost of association, the central policy and cluster centers are optimized accordingly (as described in Algorithm 2).
>
> **2. The quality of the learned autonomy model is objectively verifiable** (performance gap of 3-4%)  - Whether ADEN/RL structure in Algorithm 2 has captured meaningful autonomy---as opposed to producing random or uninformative estimates---can be assessed directly via the resulting objective function values. Solutions from our ADEN/RL Algorithm 2 efficiently models autonomy patterns, thus, yield significantly lower Free Energy $F(Y)$ in equation ($P(\beta)$). In contrast, ignoring autonomy or using an incorrect or random model leads to substantially higher Free Energy. Our simulations in Section 5 demonstrate this.
>
> ---
> *"**Q:** It is a bit unreasonable that the model could produce results close to ground truth even without any priors on the entity autonomy. In other words, one could change the "ground truth" and the model would produce the same but "much worse" results. These concerns need to be addressed."*
>
> **A:** Thank you for the comment. Our proposed Algorithm 2 is designed to track the quality of solutions produced by the ground-truth model (obtained using Algorithm 1). The simulations in Section 5 show that Algorithm 2 consistently achieves this across a wide range of autonomy levels for the two datasets considered. This is because Algorithm 2 effectively captures the influence of local autonomy $p(k|j,i)$ on clustering through repeated interactions with the data points—analogous to how a $Q$-function is learned in reinforcement learning without requiring explicit knowledge of the transition kernel $p(s' | s,a)$. Importantly, the method does not rely on, or make assumptions about, any priors on the local autonomy. Therefore, when the local autonomy is perturbed (thereby changing the ground truth), Algorithm 2 naturally adapts to the new autonomy level and produces clustering solutions that remain close to the corresponding ground truth, as demonstrated in our experiments.

---

> > ### Comment · Reviewer_ggqe · 2025-11-26
> >
> > Sorry for the typo. My intended question was "In the fully UNsupervised case, how can such estimations reflect real autonomy rather than random guesses?"
> >
> > Thanks for explaining the Q-learning-like behavior of the proposed ADEN module, which helps me better understand the method. However, since the method is only evaluated on one real-world scenario, such theoretical analyses are not convincing enough to fully demonstrate that the proposed framework could generalize to other scenarios I mentioned. Some additional experimental evidence is expected.

---

> ### Author Response · Authors · 2025-11-21
>
> *"**W:** In fact, such a result may indicate that the dataset used for evaluation is not representative and complicated enough. It is not convincing whether the method could generalize to other scenarios, such as social networks and recommender systems, especially considering the hyperparameters required."*
>
> **A:** Thanks for the comment regarding the generality of our proposed method. This observation indeed lies at the heart of our contribution. Our modeling of autonomy, the resulting problem formulation, solution methodology and the algorithms are intentionally designed to be generic - similar in spirit to established clustering techniques such as $k$-means, deterministic annealing, and Gaussian mixture models. Just as these classical methods are used widely across domains such as image segmentation, gene-expression analysis, and facility location, our proposed autonomy-aware clustering naturally applies to systems in which data points exhibit active or autonomous behavior, such as decentralized sensing, social networks, and recommender systems.
>
> As with any general-purpose methodology, some hyperparameter tuning is inevitable and is not unique to our framework. Existing clustering techniques require several such choices: $k$-means requires choosing the initialization strategy and the number of clusters; deterministic annealing requires selecting an annealing schedule and temperature parameters; and GMMs require covariance structure selection and EM initialization. Similar considerations arise in broader learning  pipelines: deep learning models require learning rates, batch sizes, and regularization, RL methods depend on exploration rates, target-network update rules, and discount factors. In this sense, the hyperparameters in our method play a role analogous to those in these well-established techniques.

---

> ### Author Response · Authors · 2025-12-03
>
> *"**Comment:** Sorry for the typo. My intended question was - "In the fully UNsupervised case, how can such estimations reflect real autonomy rather than random guesses?" Thanks for explaining the Q-learning-like behavior of the proposed ADEN module, which helps me better understand the method. However, since the method is only evaluated on one real-world scenario, such theoretical analyses are not convincing enough to fully demonstrate that the proposed framework could generalize to other scenarios I mentioned. Some additional experimental evidence is expected."*
>
> **A:** Thank you for the clarification. We are extremely glad that our explanation helped better understand our method and it brought forth the **Q-learning-like** (more generally, the **RL-type**) behaviour of the proposed ADEN module. It is correct that the proposed ADEN module (Algorithm 2) operates within the reinforcement learning paradigm, that differentiates it from any standard unsupervised or supervised setting; therefore, dependence on either of those paradigms does not arise.
>
> Thank you once again for your suggestion to incorporate additional experiments. As per your earlier mentioned scenarios, we have included extensive simulations on movie recommendation system using the popular MovieLens dataset (*Harper et. al. (2015). The movielens datasets: History and context. Acm transactions on interactive intelligent systems (tiis), 5(4), 1-19*). This setting naturally leverages clustering for recommendation and allows us to evaluate the framework on a large, high-dimensional dataset: the MovieLens data contains $1,000,209$ ratings from $6,040$ users on roughly $3,900$ movies, and the resulting user feature vectors lie in $\mathbb{R}^{10}$. We introduce $40$ different autonomy levels --- covering wide range of complex autonomy models. The results show that even when the autonomy is unknown, the proposed ADEN module (Algorithm 2) performs efficiently achieving a very small overall average error of $2.81$\% in comparison to the ground truth. See Appendix I for further details and an in-depth analysis of our experimental observations.
>
> To summarize, the revised manuscript now includes:
>
> 1) experiments on $18$ autonomy levels for the synthetic dataset (original manuscript),
>
> 2) experiments on $4$ autonomy levels for the UDT19 London traffic dataset (original manuscript),
>
> 3) experiments on $40$ autonomy levels for the MovieLens dataset (newly added in revised manuscript; see Appendix I), and
>
> 4) a stability and sensitivity analysis of Algorithm 2 in Appendix K (newly added in revised manuscript).
>
> These new additions together with existing results provide substantial empirical evidence supporting the efficacy, robustness, and generality of our framework. We thank the reviewer once again for the constructive suggestions.

---

### Official Review · Reviewer_mjED · 2025-10-26

**Soundness:** 3
**Presentation:** 4
**Contribution:** 4
**Rating:** 6
**Confidence:** 3

**Summary:**

The paper introduces the novel concept of Autonomy-Aware Clustering, addressing the critical limitation in traditional clustering where entities are assumed passive and strictly conform to assignments. In reality, entities exhibit local autonomy ($p(k|j,i)$), probabilistically overriding prescribed cluster memberships, which can significantly alter cluster centers and composition. The framework tackles this challenge in two stages. First, for the case where autonomy models are known, it adapts Deterministic Annealing (DA), leveraging its advantages in handling non-convexity and initialization sensitivity. Second, for the practical, unknown-autonomy case, it formulates the problem as a unit-horizon Markov Decision Process (MDP) and proposes a Reinforcement Learning (RL) framework to jointly learn the assignment policy and cluster representatives. A key component is the Adaptive Distance Estimation Network (ADEN), a Transformer-based attention model used within the RL loop for model-free learning of dependencies. Empirical results demonstrate that the autonomy-aware framework produces effective results.

**Strengths:**

The most notable originality lies in the formal definition and robust solution for autonomy-aware clustering, explicitly accounting for stochastic entity behavior conditional on the policy. The quality is high due to the integrated methodological approach, which uses the strong theoretical basis of Deterministic Annealing (DA) and augments it with a practical Reinforcement Learning (RL) framework incorporating a Transformer-based ADEN for model-free learning. The work's significance is demonstrated by compelling results showing the method is highly accurate and, interestingly, can sometimes use the RL exploration to achieve up to a 10% improvement over the explicitly known-model solution by escaping local minima. The paper's clarity is excellent, making the sophisticated methodology accessible.

**Weaknesses:**

The framework's primary reliance on the DA formulation limits its direct generality to resource allocation or other clustering problems that can be represented by a distance to a cluster representative, excluding non-centroid-based clustering methods. The implementation of the ADEN to estimate the average cost within the RL loop may introduce complexity and potential instability in training that is not fully analyzed, compared to simpler clustering cost functions. While the empirical results are strong, the paper lacks a direct comparison of the ADEN/RL method against modern deep clustering baselines (e.g., autoencoder-based clustering) on the same dataset, making it difficult to isolate the contribution of the autonomy modeling aspect from the benefits of using a deep Transformer model generally.

**Questions:**

What is the practical stability and computational overhead of training the Transformer-based ADEN within the RL loop, especially for large $K$ clusters or very high-dimensional data, and did the authors observe sensitivity to ADEN initialization or hyperparameter choices? Is there a theoretical path to extend this framework to non-centroid-based clustering methods, such as those relying on density or connectivity, given the current formulation's dependence on cluster representatives $y_j$? Is the reported 10% performance gain over the known-model solution an isolated case, or can the authors provide statistics on how often the learning-based approach effectively escapes local minima across a broader range of problem instances?

---

> ### Author Response · Authors · 2025-11-21
>
> Thank you for your thoughtful and encouraging evaluation of our work. We truly appreciate the recognition of the conceptual contribution, methodological rigor, and clarity of presentation. Below we address your questions and concerns.
>
> *"**Q:** Is there a theoretical path to extend this framework to non-centroid-based clustering methods, such as those relying on density or connectivity, given the current formulation's dependence on cluster representatives $y_j$? (**W:** The framework's primary reliance on the DA formulation limits its direct generality to resource allocation or other clustering problems that can be represented by a distance to a cluster representative, excluding non-centroid-based clustering methods.)"*
>
> **A:** Thank you for your insightful comment on the breadth of applicability of our framework. The current formulation is designed for clustering methods that operate through an association cost of the form $d(x_i, C_j)$, which already includes a broad family of clustering approaches. The squared-Euclidean cost function for $d(x_i,y_j)$ used in our experiments is one specific instance. Many other widely used clustering methods can be expressed in this form. For example: (i) Gaussian mixture models (GMMs): $d(x_i, C_j) = -\log p_j(x_i)$, where $p_j$ is the likelihood under component~$j$, (ii) L1 clustering: where $d(x_i, C_j)$ is the $l_1$ norm of $y_j-x_i$, (iii) KL-divergence: $d(x_i, C_j)$ is KL-divergence between probabilities $x_i$ and $y_j$, and (iv) Constrained clustering (COP-$k$-means, must-link/cannot-link):} the cost $d(x_i, C_j)$ incorporates penalties or infeasibility terms associated with violating must-link or cannot-link constraints. For all such methods, the autonomy model $p(k|j,i)$, its effect on the effective association cost $d_{\mathrm{avg}}(x_i, C_j) = \sum_k p(k|j,i)\, d(x_i, C_k)$, and our proposed RL-based solution strategy extend with minimal modification. Thus, the autonomy-aware formulation naturally covers a wide class of clustering paradigms for which cluster membership is mediated through a point-to-cluster cost.
>
> Extending the framework to clustering methods that do not rely on a prototype or representative---such as density-based (e.g., DBSCAN) or graph-based connectivity methods---is more nuanced, since these methods are not naturally expressed through $d(x_i, C_j)$. In such cases, autonomy would need to be modeled at the level of pairwise dissimilarities $d(x_i,x_\ell)$ or neighborhood structures rather than cost-to-cluster distances. Developing such extensions would likely require revisiting the modelling and mathematical formulation, and represents an interesting direction for future work.
>
> ---
> *"**W:** While the empirical results are strong, the paper lacks a direct comparison of the ADEN/RL method against modern deep clustering baselines (e.g., autoencoder-based clustering) on the same dataset, making it difficult to isolate the contribution of the autonomy modeling aspect from the benefits of using a deep Transformer model generally."*
>
> **A:** We appreciate the reviewer’s observation and the positive feedback regarding the empirical performance. Our work is motivated by a different objective than improving existing deep clustering architectures. Instead, we aim to introduce a new paradigm—clustering under local autonomy, where data points may probabilistically deviate from prescribed assignments due to latent behavioral factors. Within this setting, the role of ADEN in Algorithm 2 is not to perform representation learning (as in deep autoencoder-based clustering), but to estimate the autonomy-aware average association cost $d_{\mathrm{avg}}(x_i, y_j)$ in the RL framework, analogous to how deep networks are employed to approximate value functions in deep reinforcement learning.
>
> Because autonomy fundamentally changes the problem setting, standard deep clustering methods are not directly comparable. These methods are designed for high-dimensional feature extraction and noise reduction, but they assume that data points follow the prescribed assignment policy (passive entities) and do not model agent-level deviations. As a result, they do not account for autonomy and would yield suboptimal solutions in autonomy-driven settings.
>
> The primary objective of Algorithm 2 is to identify clustering solutions when (a) data points exhibit autonomy, and (b) the autonomy model is unknown and must be learned through interaction. In this sense, ADEN serves as an estimator tailored to this requirement rather than as a general-purpose clustering architecture. Exploring how autonomy-aware clustering can interface with modern deep clustering pipelines is an interesting direction for future work, and we thank you for highlighting this connection.

---

> ### Author Response · Authors · 2025-11-21
>
> *"**Q:** What is the practical stability and computational overhead of training the Transformer-based ADEN within the RL loop, especially for large  clusters or very high-dimensional data. **W:** The implementation of the ADEN to estimate the average cost within the RL loop may introduce complexity and potential instability in training that is not fully analyzed, compared to simpler clustering cost functions."*
>
> **A:** *Computational overhead:* ADEN is designed to add minimal computational overhead within the RL loop. Its cost is dominated by three components: (i) the projection layers $\mathcal{F}_X,\mathcal{F}_Y$, (ii) $L$ ADB (attention) layers, and (iii) the pairwise “distance head’’ used to produce the $S \times K$ autonomy-aware distances for each mini-batch.
>
> During training we use mini-batches of size $S \ll N$ (typically $S \in [32,64]$ in our experiments). This reduces the per-iteration cost from $O(NK)$ to $O(SK)$, which makes ADEN lightweight even for large datasets.
>
> *Complexity of one ADEN forward pass:* Let $d_h$ be the hidden dimension and $H$ the number of attention heads. The computational complexity is:
>
> 1- Projection layers: $O(B S d d_h + B K d d_h)$.
>
> 2- ADB blocks: Each multi-head attention in ADB processes queries of size $S$ and keys/values of size $K$, giving cost $O\bigl(L BH (S K d_h)\bigr)$, which is linear in both $S$ and $K$.
>
> 3- Pairwise distance head: After broadcasting, the pairwise tensor is of size $B S K \times 2 d_h$, and the MLP head scales as
>     $O\bigl(B S K d_h\bigr)$.
>
> For the settings used in our experiments ($d_h = 64$, $H = 4$, $L = 2$, $S \le 64$, $K \le 50$), a full forward pass is on the order of *a few million* floating-point operations—well within the regime of small modern models (ADEN has $\approx$150k parameters).
>
> The overhead of ADEN is substantially lower than that of common deep clustering backbones (e.g., convolutional autoencoders or graph encoders) that often contain $1$–$10$ million parameters. As a result, the RL loop using ADEN is roughly of the same order as running a small Transformer encoder.
>
> We have added the above analysis to the original manuscript (Appendix G) for better clarity on computational overhead of ADEN.
>
> *Stability:* About the stability of ADEN, please refer to our response to your next question.
>
> ---
> *"**Q:** Did the authors observe sensitivity to ADEN initialization or hyperparameter choices?"*
>
> **A:** Thank you for raising this important question. To directly evaluate the sensitivity of ADEN to initialization and hyperparameters, we conducted an extensive set of $32$ additional experiments covering multiple random seeds, two environment configurations (different autonomy levels $\kappa$), two ADEN architectures, and four training-configuration settings (different learning rates and initialization strategies). These results, together with loss and free-energy curves, are now documented in the new Appendix K.
>
> Across these $32$ runs, ADEN training was stable in $28$ cases: both the network regression loss and the free energy decreased smoothly and consistently to low values. These stable cases include runs with different random seeds, different initialization schemes (mean noise vs sample), different learning rates, and both ADEN model sizes. This demonstrates that ADEN is not sensitive to weight initialization or moderate hyperparameter variations.
>
> The observed robustness is also motivated by the maximum entropy principle (MEP). ADEN is trained under the $\beta$-annealing continuation scheme described in Algorithm 2: when $\beta \approx 0$, all autonomy-aware distances $d_{\mathrm{avg}}(x_i,y_j)$ collapse to nearly the same scale, making the learning problem extremely easy; as $\beta$ increases, the network is warm-started from the previous level, allowing it to refine its estimates gradually. This homotopy-like procedure promotes stability that would otherwise arise from training directly at high $\beta$. Consequently, training remains stable across different initializations and hyperparameter settings.
>
> Only $4$ out of $32$ runs showed a plateau in the loss. Importantly, these are not instability cases but arise under a very specific combination: the largest ADEN architecture, the highest learning rate, mean noise initialization, and high autonomy ($\kappa$ large).  In these cases, the loss remains nearly constant rather than diverging or becoming unstable. All other runs under the same environments but with different learning rates or initialization strategies converge normally, confirming that these plateaus reflect an unusually flat landscape rather than sensitivity or training instability.
>
> In summary, our empirical results show that ADEN is highly stable and exhibits remarkably low sensitivity to initialization and hyperparameter choices.

---

> ### Author Response · Authors · 2025-11-21
>
> "**Q:** Is the reported 10% performance gain over the known-model solution an isolated case, or can the authors provide statistics on how often the learning-based approach effectively escapes local minima across a broader range of problem instances?"
>
> **A:** We appreciate the reviewer’s insightful question. The 10% improvement mentioned in the paper is not intended as a general claim; rather, it highlights that Algorithm 2 can occasionally outperform the known-model solution (Algorithm 1; ground truth) due to the non-convexity of the underlying optimization landscape. Clustering with squared Euclidean distortions is known to exhibit many poor local minima, and avoiding these is a common challenge for any clustering algorithm.
>
> In one of the autonomy configurations used with the UDT19 dataset, we observed an average improvement of approximately 10% over Algorithm 1 (the ground truth). This arises because Algorithm 2 updates the cluster representatives using a *learned* estimate $d_{\theta}(x_i,y_j)$ of the true autonomy-aware cost $d_{\mathrm{avg}}(x_i,y_j)$, which is unknown. The resulting gradients are noisy, and this noise can occasionally help the algorithm escape unfavorable local minima that the ground-truth method (which uses exact gradients) gets trapped in. Similar phenomena—where optimization over a learned surrogate finds a better local minimum than direct optimization of the true objective—have been reported in the optimization literature (e.g., Ge, Rong, et al. "Escaping from saddle points—online stochastic gradient for tensor decomposition." Conference on learning theory. PMLR, 2015.; Neelakantan, Arvind, et al. "Adding gradient noise improves learning for very deep networks. arXiv 2015." arXiv preprint arXiv:1511.06807 (2020); Wu, Jingfeng, et al. "On the noisy gradient descent that generalizes as sgd." International Conference on Machine Learning. PMLR, 2020.).
>
> The key takeaway from our experiments is that Algorithm 2 closely tracks the behavior of Algorithm 1, even though it has no prior access to the autonomy model. Its primary purpose is to learn and incorporate autonomy when it is unknown, not to outperform the ground-truth solution. However, due to non-convexity and noisy gradient updates, occasional improvements are possible, as observed in this instance.

---

### Official Review · Reviewer_no6m · 2025-10-31

**Soundness:** 3
**Presentation:** 3
**Contribution:** 3
**Rating:** 6
**Confidence:** 3

**Summary:**

This paper introduces autonomy-aware clustering, a new formulation of the clustering problem that explicitly accounts for cases where individual entities may override their prescribed cluster assignments. For example, sensors sending data to a non-assigned base station or users acting outside expected preference groups. The authors formalize this idea by introducing a local autonomy model  that captures the probability of an entity originally assigned to cluster jj actually joining cluster kk. They first extend Deterministic Annealing clustering to this setting when the autonomy model is known, and then develop a reinforcement-learning-based approach (with a Transformer-based Adaptive Distance Estimation Network, ADEN) to learn the autonomy dynamics when unknown. Experiments on synthetic data and a decentralized sensing application (UAV placement over traffic sensors) show that the proposed method achieves near-optimal solutions even without explicit autonomy models, outperforming baselines that ignore autonomy.

**Strengths:**

* The paper raises a genuinely interesting and underexplored issue — that entities in clustering may act autonomously rather than passively following their assigned clusters. This perspective is both conceptually fresh and practically relevant, especially for systems with distributed agents (sensors, users, robots, etc.).

* Casting clustering with unknown autonomy as a one-step MDP is a clever move, allowing the use of RL techniques without explicit autonomy models.

* The synthetic and real-world experiments convincingly show that ignoring autonomy leads to significant degradation in performance, while the proposed model maintains small optimality gaps.

* The paper manages to combine analytical results (e.g., β-annealing behavior) with practical deep-learning components (ADEN) in a coherent framework.

**Weaknesses:**

- While the framing is fresh, it remains unclear how far this differs from existing probabilistic or soft-clustering models (e.g., mixture models, stochastic EM, or clustering with noisy assignments). In those methods, points also have probabilistic memberships, which may implicitly capture similar uncertainty. The paper could better clarify how “local autonomy” goes beyond mere stochasticity in assignments.

- The autonomy term is elegant mathematically but may be hard to interpret or estimate in real data. If autonomy stems from unobserved confounders, does the model risk overfitting to noise rather than uncovering meaningful autonomy?

- In many practical domains, deviations from assigned clusters could also be viewed as data noise, mislabeling, or temporary network failures, phenomena often handled by preprocessing rather than by modifying the clustering objective. The paper could better articulate when modelling autonomy is essential rather than optional.

* The RL + transformer-based approach (ADEN) seems quite heavy relative to the conceptual simplicity of the problem. The added learning complexity may limit adoption in standard clustering pipelines.

**Questions:**

1. How does this framework differ fundamentally from soft or probabilistic clustering methods (e.g., GMMs, fuzzy c-means) that already allow stochastic assignments?
    What does the autonomy layer model that those do not?

2. In practice, how can one tell whether observed deviations reflect genuine _autonomy_ versus noisy or corrupted data?

3. Does the method require retraining for every new dataset, or can the learned ADEN generalise across settings with similar autonomy patterns?

4. How sensitive is the method to the choice of annealing schedule or β-step size?

---

> ### Author Response · Authors · 2025-11-21
>
> Thank you for your thoughtful and encouraging assessment of our work. We are grateful for the recognition of both the conceptual motivation behind autonomy-aware clustering and the technical contributions of approach. We appreciate your insights and are glad to respond to your questions and concerns below.
>
> *"**W:** While the framing is fresh, it remains unclear how far this differs from existing probabilistic or soft-clustering models (e.g., mixture models, stochastic EM, or clustering with noisy assignments). In those methods, points also have probabilistic memberships, which may implicitly capture similar uncertainty. The paper could better clarify how “local autonomy” goes beyond mere stochasticity in assignments. **Q:** How does this framework differ fundamentally from soft or probabilistic clustering methods (e.g., GMMs, fuzzy c-means) that already allow stochastic assignments? What does the autonomy layer model that those do not?"*
>
> **A:** Thank you so much for raising this subtle and important distinction. While soft or probabilistic clustering methods indeed allow stochastic memberships, the nature and source of that stochasticity are quite different from what we model as autonomy. In standard probabilistic clustering, the uncertainty arises at the level of the assignment policy $\pi$: a data point is assigned to clusters with certain probabilities because multiple cluster representatives explain its features $x_i$ comparably well. Thus, the stochasticity reflects *algorithmic uncertainty based solely on feature similarity*.
>
> In contrast, autonomy captures a different phenomenon: it models deviations that stem from latent, unobserved behavioural factors that are *not* encoded in $x_i$. A data point may override the prescribed assignment due to local conditions or constraints, even when feature-based prescriptions remain clear. In this sense, the stochasticity arises at the level of the data point itself, not at the level of the assignment policy. Autonomy therefore complements, rather than replaces, the stochasticity modeled by soft cluster assignments.
>
> This distinction appeared briefly in the original manuscript; we now elaborate on it more clearly in the revised manuscript (please see Section 2; text in blue ink). We also emphasize that our framework naturally supports policy-level stochasticity as well: the optimal assignment policy is a Gibbs distribution (Eq. 1), reducing to hard assignments only as $\beta \to \infty$. Autonomy therefore augments, rather than replaces, the probabilistic structure of soft clustering methods.
>
> ---
> *"**W:** The autonomy term is elegant mathematically but may be hard to interpret or estimate in real data. If autonomy stems from unobserved confounders, does the model risk overfitting to noise rather than uncovering meaningful autonomy? **Q:** In practice, how can one tell whether observed deviations reflect genuine autonomy versus noisy or corrupted data?"*
>
> **A:** Thank you for raising this thoughtful question, as it touches an important practical consideration. In many real systems, the assignment policy does not need to explicitly distinguish whether deviations arise from genuine autonomy or from certain forms of environmental noise: for instance, a data point prescribed to join cluster $j$ may instead associate with some $k \neq j$. Whether this deviation is due to local behavioural autonomy or due to noise in the communication channel, the clustering solution must adapt to the *observed* behaviour.
>
> However, there are also scenarios where deviations are known *a priori* to stem purely from channel noise while the data points themselves are passive—i.e., they reliably follow the prescribed assignment, but the noisy communication channel reports an incorrect cluster assignment to the central planner. In such cases, deploying an autonomy-aware approach is not appropriate. The appropriate strategy here is to use standard clustering techniques that do not consider autonomy.
>
> In many realistic deployments, the impact of noise on the autonomy model is naturally limited. Because the number of cluster representatives is small relative to the data points and these representatives are typically well separated (i.e., $d(y_j, y_k) \gg \varepsilon$), modest noise is unlikely to substantially change the autonomy distribution. For noise to meaningfully affect $p(k \mid j,i)$, it would need to be large enough to overwhelm the inherent separation among representatives—at which point the issue would compromise not just autonomy-aware clustering but the reliability of the sensing or communication system itself.
>
> Finally, a more formal analysis of how noise in the autonomy model $p(k \mid j,i)$ affects the clustering solutions can be carried out using tools from robust reinforcement learning, where one studies the impact of noise in the transition kernel $p(s'| s,a)$. This constitutes an interesting direction for future work.

---

> ### Author Response · Authors · 2025-11-21
>
> *"**W:** In many practical domains, deviations from assigned clusters could also be viewed as data noise, mislabeling, or temporary network failures, phenomena often handled by pre-processing rather than by modifying the clustering objective. The paper could better articulate when modeling autonomy is essential rather than optional."*
>
> **A:** Thank you for the comment. We addressed the relationship between noise, network effects, and local autonomy in the previous response. Regarding the remaining point, autonomy-aware clustering becomes essential whenever data points are active and may autonomously override the assignments prescribed by a policy. Our current framework already captures several such scenarios that may exhibit noisy communication channels, as previously discussed. Extensions along the lines of robust RL could further broaden the framework to handle more general noisy perturbations to autonomy. In summary, regardless of the presence or absence of noise, autonomy-aware clustering is required whenever entities possess the ability to deviate from their assigned clusters.
>
> ---
> *"**W:** The RL + transformer-based approach (ADEN) seems quite heavy relative to the conceptual simplicity of the problem. The added learning complexity may limit adoption in standard clustering pipelines."*
>
> **A:** We appreciate this perspective and understand the concern about added learning complexity. While the overall goal of clustering may appear simple, incorporating local autonomy fundamentally alters the structure of the problem. In particular, when the autonomy model $p(k|j,i)$ depends jointly on the data features $\{x_i\}$ and evolving cluster representatives $\{y_j\}$, the resulting optimization landscape becomes significantly more intricate than in classical clustering. Because the autonomy model is typically *not* known to the central decision-maker and must be inferred through interaction, an RL formulation becomes a natural tool to capture this latent behaviour. ADEN was designed and incorporated within the RL loop to learn the intricate and complex effect of local autonomy on the average association cost $d_{\text{avg}}(x_i,y_j)$. This is akin to use of any deep network to approximate the $Q$-function in deep RL methods. In our experiments, ADEN contains roughly 150k parameters, which is modest compared to many deep architectures (such as auto-encoders or deep graph based networks) commonly used in contemporary deep clustering pipelines for simpler settings where the data points are passive (do not even exhibit autonomy).
>
> For scenarios where it is known *a priori* that autonomy levels are very small and only weakly dependent on the feature vectors ${x_i}$ and ${y_j}$, ADEN can indeed be replaced by a simpler network, provided it can reliably estimate the average autonomy-aware association cost $d_{\text{avg}}(x_i,y_j)$. In our current exposition (simulation Section 5), we choose ADEN because it performs well even in medium- to high-autonomy regimes, where the autonomy distribution $p(k | j,i)$ also depends explicitly on both the data features $x_i$ and the cluster representatives $y_j$. In such settings, richer architectures offer clear benefits, but the framework itself remains flexible: the complexity of the estimator can be adapted to the autonomy structure of the application.
>
> ---
> *"**Q:** Does the method require retraining for every new dataset, or can the learned ADEN generalise across settings with similar autonomy patterns?"*
>
> **A:** We thank the reviewer for this question. In our framework, retraining is required *only* when the underlying local autonomy model $p(k| j, i)$ changes across problem instances. ADEN is designed to learn the expected autonomy-adjusted distance $d_{\mathrm{avg}}(x_i, y_j)$, as a function of the entity feature tensor $\mathbf{X}$ and the cluster representative tensor $\mathbf{Y}$ (see Appendix G for details). Crucially, these expected distances are determined by the autonomy distribution $p(k | j, i)$, which may or may not depend on $\{x_i\}$ and $\{y_j\}$.
>
> If the autonomy model $p(k |j, i)$ remains the same across multiple datasets or tasks, then ADEN can be directly reused without retraining. In this case, the learned mapping from $(\mathbf{X},\mathbf{Y})$ to $d_{\mathrm{avg}}(x_i,y_j)$ transfers across problem instances, allowing the central policy to adapt cluster assignments efficiently.
>
> If the autonomy model changes—e.g., agents adopt different behavioral rules, incentives, or environment-driven deviation patterns—then the expected distances change accordingly. Because ADEN explicitly models these autonomy-induced effects, a change in local autonomy distribution $p(k | j, i)$ requires retraining to correctly capture the new deviation structure.

---

> ### Author Response · Authors · 2025-11-21
>
> *"**Q:** How sensitive is the method to the choice of annealing schedule or β-step size?"*
>
> **A:** We appreciate the reviewer’s question regarding the annealing schedule, as it is an important component of the algorithm design. The behavior of the annealing parameter $\beta$ and its influence on the cluster representatives $\{y_j\}$ are characterized in Theorems 2 and 3, together with the discussion in Appendix E and the .mp4 file uploaded earlier as supplementary material. These results describe how the solution changes only at certain *critical* values of $\beta$, and remains essentially stable between them.
>
> Motivated by this structure, we use a geometric annealing schedule $\beta_{k+1} = \tau \beta_k$ with $\tau>1$. When the autonomy model is known explicitly, one can compute the critical values $\beta_{\mathrm{cr}}$s and move directly between them. In the more realistic setting where autonomy is unknown (or computing $\beta_{\text{cr}}$ is expensive), a geometric schedule provides an efficient surrogate. Linear or logarithmic schedules tend to be less effective in our setting because they spend many iterations in regions where the solution does not change appreciably.
>
> In practice, the key requirement is that the multiplicative factor $\tau$ should not cause the schedule to skip over a phase transition region (i.e., an interval containing a $\beta_{\text{cr}}$). Very small values of $\tau$ (e.g., $\tau$ roughly 1.001) guarantee this but lead to longer runtimes, while very large values (e.g., $\tau>1.5$) risk jumping over important phase transitions and converging to poorer minima. Across all our simulations, values in the range $\tau \in [1.01, 1.1]$ consistently offered a good balance between computational efficiency and solution quality.

---

> > ### Comment · Reviewer_no6m · 2025-11-26
> >
> > Thank you for your very thorough response! I've gained further understanding about your method and its relevance. I do understand now how autonomy aware clustering deals with a scenario where stochasticity arises from the inherent system and not from the features itself which as I understand requires modelling a separate or joint latent space with the feature latent space.
> >
> > Overall, you have answered my questions very well. I believe though similar to reviewer mjED that one more experiment would greatly add value to the method and the proposed framework.
> >
> > Therefore I'm leaning between a 6-8 with my score. I will however for now update my confidence, since I understand the relevance of the paper much better thanks to the authors responses.

---

> > > ### Author Response · Authors · 2025-12-03
> > >
> > > *" **Comment:** Thank you for your very thorough response! I've gained further understanding about your method and its relevance. I do understand now how autonomy aware clustering deals with a scenario where stochasticity arises from the inherent system and not from the features itself which as I understand requires modelling a separate or joint latent space with the feature latent space. Overall, you have answered my questions very well. I believe though similar to reviewer mjED that one more experiment would greatly add value to the method and the proposed framework. Therefore I'm leaning between a 6-8 with my score. I will however for now update my confidence, since I understand the relevance of the paper much better thanks to the authors responses."*
> > >
> > > **A:** We sincerely thank you for the thoughtful follow-up and the careful consideration of our earlier responses. We are glad that the clarifications helped convey the relevance of our work, and we truly appreciate your positive feedback, updated confidence, and assessment of our work.
> > >
> > > We also appreciate the suggestion regarding additional experiments. We have added extensive simulations on movie recommendation system using the popular MovieLens dataset (*Harper et. al. (2015). The movielens datasets: History and context. Acm transactions on interactive intelligent systems (tiis), 5(4), 1-19*). This setting naturally leverages clustering for recommendation and allows us to evaluate our framework on a large, high-dimensional dataset: the MovieLens data contains $1,000,209$ ratings from $6,040$ users on roughly $3,900$ movies, and the resulting user feature vectors lie in $\mathbb{R}^{10}$. We introduce $40$ different autonomy levels --- covering wide range of complex autonomy models. The results show that even when the autonomy is unknown, the proposed ADEN module (Algorithm 2) performs efficiently achieving a very small overall average error of $2.81$\% in comparison to the ground truth. See Appendix I for further details and an in-depth analysis of our experimental observations. We have also added a stability and sensitivity analysis of Algorithm 2 in Section K in the revised manuscript. These new additions together with existing results further strengthen the experimental demonstration of our framework. We thank you once again for the constructive suggestions.

---

### Official Review · Reviewer_2opo · 2025-10-31

**Soundness:** 1
**Presentation:** 1
**Contribution:** 2
**Rating:** 0
**Confidence:** 3

**Summary:**

## Summary

The paper presents autonomy-aware clustering, which allows entities (data points) to probabilistically deviate from their assigned clusters according to a probability distribution.
It extends Maximum Entropy Principle–based Deterministic Annealing (MEP-DA) by local autonomy distributions into the objective function. When the autonomy distribution is known, the authors provide analytical update rules and analyze convergence behavior. When it is unknown, they propose a neural network (ADEN) to estimate average costs through supervised learning, alternating between model training and cluster updates until an annealing threshold is reached.  The authors evaluate their method on synthetic and real-world traffic data. The results indicate that accounting for local autonomy improves clustering performance compared to ignoring it.

## Contribution

While I have reservations about whether local autonomy is a good model for decentralized sensing, I acknowledge that it could be an interesting problem to study. The theoretical contribution seems solid, albeit incremental.

**Strengths:**

- Local entity autonomy could be an interesting problem, and decentralized sensing is a plausible scenario.
- The theoretical contribution seems solid, albeit incremental.

**Weaknesses:**

## Soundness

The paper makes several unsubstantiated claims and has major methodological flaws:

- The authors do not justify why local autonomy is a better model for decentralized sensing than other approaches.
- The authors claim to use Reinforcement Learning but this is not justified, in my opinion (see detailed discussion below).
- The MDP the authors claim to solve is never formalized.
- The experimental setup has major flaws: the problems are contrived and artificial, the baselines are not clearly defined.
- Some numerical results are missing (e.g., UDT19 dataset) and the empirical evaluation has major flaws. How can an algorithm achieve "10% improvement over ground truth"?
- The claim "These gaps can be further reduced through standard hyperparameter tuning and extended training" is not supported by any evidence.
- The problem formulation is presented as major contribution, but it is a straightforward extension of MEP-DA.

## Presentation

The presentation is poor and needs significant improvement:

- The writing is convoluted and hard to follow. The paper is full of jargon and undefined terms (e.g., "entity").
- The figures/tables are low quality and hard to read.
- The related work section is weak and does not contextualize the work well. Yes, other works do not use local autonomy, but how do they methodologically differ from this work? Why is local autonomy a better model?
- The conclusion is completely missing.
- Limitations are completely missing.

I want to discuss the Reinforcement Learning aspect in more detail, as it is a major red flag for me. From my understanding, the first loop in Algorithm 2 performs the following steps:

1. Assign new distances to all data points based on the current cluster centroids.
2. Sample assignments using the currently learned soft assignment distribution $\pi_\theta$.
3. Observe actual assignment outcomes using the autonomy distribution and update empirical costs.
4. Update an exponential moving average of the costs.
5. Update the neural network parameters $\theta$ using supervised learning to minimize the squared error between the predicted costs and the empirical costs.

This is not Reinforcement Learning. It does not use an MDP formulation. There is no action space, state space, transition dynamics, or reward function. There is also no temporal component: While the training loop is run repeatedly, there is no temporal difference learning despite the claim of a "straightforward Q-learning–style stochastic iterative update". Instead, Algorithm 2 is supervised learning with an alternating optimization scheme. The authors should clarify this, or correct my misunderstanding by formally defining the MDP they are solving and the Bellman equation they are using.

**Questions:**

- Why would I model the decentralized sensing problem with local autonomy instead of modeling interference/noise directly?
- Can the authors clarify why Algorithm 2 is using Reinforcement Learning?
- What MDP is Algorithm 2 solving? The authors should be able to formalize the action space, state space, transition dynamics, and reward function.
- What baseline is used in the results? MEP-DA w/o the local autonomy distribution?
- In the results, the paper claims that "Notably, when $\kappa=0.1$ , $T = 0.1$, ADEN matches the performance of the model-based baseline (ground truth), and for $\kappa=0.5$, T = 0.1 it achieves approximately a 10% improvement over the ground truth, despite the absence of an explicit autonomy model".
  - What is the model-based baseline (ground truth)? Algorithm 1 with the true autonomy distribution?
  - How is it possible to outperform the ground truth?
- Where are the numerical results for the UDT19 dataset?
- What are limitations of the proposed method? When would I use it over other methods and vice versa?

---

> ### Author Response · Authors · 2025-11-21
>
> Thank you for the summary of our work and for recognizing the relevance of local entity autonomy. We appreciate your acknowledgment of the soundness of the theoretical contributions of our work. Below, we address your questions and concerns in detail.
>
> *"**Q:** Why would I model the decentralized sensing problem with local autonomy instead of modeling interference/noise directly?"*
>
> **A:** Thank you for raising this point. Local autonomy and interference/noise correspond to different aspects of the system, and our framework does not treat autonomy as a proxy for physical-layer noise. Rather, interference, fading, congestion, and security-related factors are *possible drivers* of autonomy. The autonomy term $p(k | j,i)$ is intended to provide a behavioral abstraction capturing the probability that a sensor deviates from its nominally assigned UAV under heterogeneous local conditions that cannot be reduced to a simple analytical model.
>
> In decentralized sensing networks, even with static UAV positions, a sensor at $x_i$ may communicate with a UAV $k \neq j$ for several reasons: degraded SINR from interference, fading, or blockage; congestion or queue buildup at the nominal UAV; energy-conserving heuristics favoring a lower-power link; scheduling conflicts; or security/privacy constraints. For example, if UAV $j$ is temporarily untrusted, compromised, or operating under limited-certification modes, a sensor may route its data to a more secure UAV $k$, even if $j$ is geographically closer. Such deviations are not easily captured by standard interference/noise models, but they naturally get captured in the autonomy model $p(k|j,i)$.
>
> Explicitly modeling all contributing factors—interference, shadowing, blockage, queue dynamics, battery constraints, congestion control, and security risks—would require detailed *environment-specific* models that are typically unavailable in practice. Aggregating their combined impact into the autonomy distribution $p(k | j,i)$ yields a formulation that is more robust, more general, and readily applicable across diverse operating conditions, while also being amenable to RL-based learning when the autonomy behavior is unknown. In the revised manuscript, we greatly elaborate on the local autonomy and its implications in decentralized sensing, and provide a more intuitive explanation (using schematics) that helps the reader understand the applicability of autonomy-aware clustering in such settings (see Section 1 and Appendix A for details).
>
> ---
> *"**Q:** What MDP is Algorithm 2 solving? The authors should be able to formalize the action space, state space, transition dynamics, and reward function."*
>
> **A:** Thank you for the suggestion to further clarify the MDP underlying Algorithm 2. The original manuscript discusses the unit-horizon MDP viewpoint of autonomy-aware clustering in Section 1. We agree that making this correspondence explicit in the later sections will help readers who wish to explore the MDP/RL interpretation more deeply. Accordingly, the revised manuscript now includes a concise formal description of the MDP structure in Section 4 (highlighted in blue), and further discussions in Section 6.
>
> In this MDP formulation, the state space comprises the data points $\{x_i\}$ together with the current cluster representatives $\{y_j\}$; the action space consists of the clusters $\{C_j\}$; the autonomy distribution $p(k | j,i)$ specifies the transition probabilities; the instantaneous cost is the cluster–association cost $d(x_i,C_j)$; and the discount factor is $\gamma=0$ due to the unit-horizon structure. The cluster assignment policy $\pi(j|i)$ determines the action taken at each state. This viewpoint directly motivates the use of reinforcement learning: it allows us to learn optimal assignments and autonomy-aware costs even when the transition kernel (i.e., the autonomy model) is unknown. See Section 4 for the formal MDP structure and the related value function.
>
> The key distinction of our framework from standard RL is that the learned estimate $d_{\theta}(x_i,y_j)$ is used not only to update the assignment policy $\pi(j | i)$, but also to construct the estimated free-energy function $F(Y)$ in ($\hat{P}(\beta)$) , which is then optimized over the cluster representatives $Y$. Thus, as described in Algorithm 2, RL motivates a principled and effective mechanism for learning the autonomy-aware clustering solutions.

---

> ### Author Response · Authors · 2025-11-21
>
> *"**Q:** Can the authors clarify why Algorithm 2 is using Reinforcement Learning?"*
>
> **A:** We thank the reviewer for asking about the role of reinforcement learning in Algorithm 2. Our use of RL follows directly from the unit-horizon MDP viewpoint of autonomy-aware clustering introduced in the paper. This viewpoint allows us to borrow the algorithmic structure of techniques (such as RL) developed for solving MDPs, while adapting them to the clustering setting.
>
> Concretely, Algorithm 2 mirrors the components of a standard RL pipeline. We begin by sampling a data point, analogous to sampling an initial state. The cluster assignment $\epsilon$-greedy policy plays the role of an $\epsilon$-greedy action selection mechanism. Once a cluster $j \sim \pi(\cdot | i)$ is prescribed, the data point may associate with a different cluster $k \sim p(\cdot |j,i)$, paralleling a transition from state $s$ to $s'$ under the dynamics $p(s'|s,a)$.
>
> The function $d_{\theta}(x_i,y_j)$ serves as a learned estimate of the autonomy-aware cost, and is updated by minimizing the loss $L(\theta)$ in a manner similar to updating a value function (or $Q$-function) in deep RL. These updates occur within an annealing loop over the entropy parameter $\beta$, analogous to the annealing strategies used in $G$-learning and other entropy-regularized RL methods.
>
> The key distinction from standard RL is that the learned estimate $d_{\theta}(x_i,y_j)$ is used not only to update the assignment policy $\pi(j \mid i)$, but also to construct the estimated free-energy function $F(Y)$ in ($\hat{P}(\beta)$), which is then optimized over the cluster representatives $Y$. Thus, as described in Algorithm 2, RL provides a principled and effective mechanism for learning the autonomy-aware clustering solutions.
>
> ---
> *"**Q:** What baseline is used in the results? MEP-DA w/o the local autonomy distribution?"*
>
> **A:** Thanks for seeking clarification on this. We have used the two baselines: (A) solutions from Algorithm 1 that assumes the complete knowledge of autonomy (we also refer to this as the ground truth), and (B) solutions from deterministic annealing - that fails to incorporate autonomy in clustering solutions. While the baseline (B) is used to concretize the dire consequences of failing to model and incorporate autonomy in clustering solutions, whenever the data points are active, the baseline (A) is used to demonstrate the efficacy of our Algorithm 2 in providing clustering solutions even when autonomy is entirely unknown.
>
> ---
> *"**Q:** In the results, the paper claims that "Notably, when $\kappa=0.1$ , $T = 0.1$, ADEN matches the performance of the model-based baseline (ground truth), and for $\kappa=0.5$, T = 0.1 it achieves approximately a 10% improvement over the ground truth, despite the absence of an explicit autonomy model". What is the model-based baseline (ground truth)? Algorithm 1 with the true autonomy distribution? How is it possible to outperform the ground truth?"*
>
> **A:** Yes, Algorithm 1 acts as the ground truth to demonstrate the efficacy of Algorithm 2. We observe that Algorithm 2 outperforms Algorithm 1 in one of the instances on UDT19 dataset (as mentioned above). This improvement mentioned in the paper is not intended as a general claim; rather, it highlights that Algorithm 2 can occasionally outperform the known-model solution due to the non-convexity of the underlying optimization landscape. Clustering with squared Euclidean distortions is known to exhibit many poor local minima, and avoiding these is a common challenge for any clustering algorithm.
>
> Further, the Algorithm 2 updates the cluster representatives using a *learned* estimate $d_{\theta}(x_i,y_j)$ of the true autonomy-aware cost $d_{\mathrm{avg}}(x_i,y_j)$, which is unknown. The resulting gradients are noisy, and this noise can occasionally help the algorithm escape unfavorable local minima that the ground-truth method (which uses exact gradients) gets trapped in. Similar phenomena—where optimization over a learned surrogate finds a better local minimum than direct optimization of the true objective—have been reported in the optimization literature (Ge, Rong, et al. "Escaping from saddle points—online stochastic gradient for tensor decomposition." Conference on learning theory. PMLR, 2015.; Neelakantan, Arvind, et al. "Adding gradient noise improves learning for very deep networks. arXiv 2015." arXiv preprint arXiv:1511.06807 (2020); Wu, Jingfeng, et al. "On the noisy gradient descent that generalizes as sgd." International Conference on Machine Learning. PMLR, 2020.)
>
> The takeaway from our experiments is that Algorithm 2 closely tracks the behavior of Algorithm 1, even though it has no prior access to the autonomy. Its primary purpose is to learn and incorporate autonomy in clustering solutions when it is unknown, not to outperform the ground-truth. However, due to non-convexity and noisy gradients, occasional improvements are possible, as observed in this instance.

---

> ### Author Response · Authors · 2025-11-21
>
> *"**Q:** Where are the numerical results for the UDT19 dataset?"*
>
> **A:** Thank you for seeking clarification on the results for UDT19 dataset. The simulation setup and numerical results for the UDT19 dataset are provided and discussed in lines~421--475 of the original manuscript, and are illustrated in Figure 2. In particular, we simulate four variants of the autonomy model $p(k|j,i)$, governed by parameters $\kappa$ and $T$ (as described in Section 5), and compare the performance of Algorithm 2 with the ground-truth solution obtained from Algorithm 1.
>
> These four variants span a range of autonomy levels. At low ($\kappa=0.1,T=0.1$) and medium ($\kappa=0.5,T=0.1$) autonomy levels (corresponding to a more concentrated distribution achieved with larger $T=0.1$), Algorithm 2 performs comparably to, and occasionally even better than, the ground truth. While this improvement is not claimed to be general, it can occasionally occur in highly nonconvex clustering landscapes where the noisy gradient steps in Algorithm 2 help it escape poor local minima that Algorithm 1 converges to.
>
> At higher autonomy levels (smaller $T=0.01$, yielding a more uniform autonomy distribution), Algorithm 2 continues to track the ground-truth performance, with average gaps of 18.37% and 24.82%. These gaps can be further reduced with additional training. Overall, the four autonomy variants collectively demonstrate the ability of the RL-based method to learn autonomy effects and incorporate them effectively in determining the final clustering solutions, while tracking the performance of the ground truth.
>
> ---
> *"**Q:** What are limitations of the proposed method? When would I use it over other methods and vice versa?"*
>
> **A:** Thank you for the comment on scope and applicability our proposed framework. Broadly, our work has two significant contributions. First, is the observation, ideation, and modeling of autonomy of data points that, to the best of our knowledge, has not been studied in the existing clustering literature. This is followed by another significant contribution of an RL-based framework that could determine clustering solutions even when autonomy is not explicitly known. The proposed framework is as general as clustering techniques such $k$-means++, and deterministic annealing.
>
> Our proposed framework should be used to determine clustering solutions in scenarios where the data points are active, i.e., they exhibit autonomy and can possibly override the cluster prescription from a global assignment policy. This includes several broad application areas such as recommendation systems, decentralized sensing, mobile or robotic systems. However, if we know that the data points are passive, and they would comply with the cluster prescription of the clustering technique, then such scenarios need not be studied under autonomy-aware clustering, instead can be directly dealt with existing clustering techniques. While not particular to our solution methodology, one challenge of the class of autonomy-aware clustering problems is the computational overhead of learning the local autonomy when an explicit model is not given, where large datasets with complex autonomy models $p(k|j,i)$ may require extensive training for viable clustering solutions.

---

> ### Author Response · Authors · 2025-11-21
>
> *"**W:** The authors do not justify why local autonomy is a better model for decentralized sensing than other approaches."*
>
> **A:** Thank you for raising this point. Our intention is not to claim that local autonomy is the only or definitively best way to model decentralized sensing. Rather, local autonomy provides a **general and flexible framework** that applies broadly to settings in which data points (entities) may act autonomously and override prescribed assignments. Decentralized sensing is one such application—used primarily to illustrate how autonomy naturally arises—not the sole target of our method.
>
> In decentralized sensing, deviations from intended assignments may occur for many heterogeneous reasons: congestion at a UAV, intermittent fading or blockage, interference variations, energy constraints, scheduling conflicts, or security and trust considerations. Explicitly modeling each of these factors requires environment-specific assumptions, detailed channel or interference models, or multi-layer system models that are often unavailable. Moreover, these effects are frequently coupled in ways that are difficult to capture analytically.
>
> Local autonomy offers a principled abstraction that **aggregates** such sources of deviation into a **single** probabilistic behavior model, without requiring explicit modeling of each underlying physical or operational mechanism. This makes the approach ** environment-agnostic, **scalable** - avoids constructing increasingly complex system-specific models as the environment changes, and **broadly applicable** , because the same autonomy formulation also captures behaviors in recommender systems, robot task switching, and other domains with active data points.
>
> Our goal in this work is therefore to propose a **general** formulation of autonomy-aware clustering that can accommodate decentralized sensing as one instance, rather than to engineer a specialized method tailored solely to that setting. Subsuming diverse behaviors under the autonomy model provides a unifying and practical approach that remains robust across different applications and environments.
>
> ---
> *"**W:** The experimental setup has major flaws: the problems are contrived and artificial, the baselines are not clearly defined."*
>
> **A:** Our objective in this work is to introduce autonomy in the context of clustering framework. In our work, to evaluate the behavior of Algorithm 2, which arises from our proposed framework, we consider two complementary experimental setups: (i) a synthetic dataset, and (ii) the UDT19 urban traffic dataset, across a diverse and representative setting of autonomy levels. Across a broad range of autonomy levels, in both of the setup, our experiments show that Algorithm 2 consistently tracks the performance of the ground-truth solution (Algorithm 1). The key distinction is that Algorithm 1 operates with complete knowledge of the autonomy model, whereas Algorithm 2 does not require such information.
>
> To our knowledge there are no established literature that addresses the autonomy in the context of clustering as we have introduced. However, to demonstrate the effectiveness of the proposed Algorithm 2, we consider two baselines as detailed in our earlier response. More precisely, we consider (i) Algorithm 1 that assume complete autonomy information (ground truth), and (ii) deterministic annealing algorithm that completely ignores autonomy while clustering. The latter brings out the dire consequences of ignoring autonomy, whereas the former brings out the efficacy of the RL-based Algorithm 2.
>
> ---
> *"**W:** The claim "These gaps can be further reduced through standard hyperparameter tuning and extended training" is not supported by any evidence."*
>
> **A:** Thank you for raising this point. Our statement regarding hyperparameter tuning was intended as a general remark rather than a specific empirical claim. Algorithm 2 is implemented within a standard deep learning/RL/MEP pipeline, where performance is known to depend on choices such as learning rate schedules, batch sizes, network capacity, training duration, and annealing rates. While our results do not rely on additional tuning to support the main conclusions, it is reasonable and consistent with typical deep RL practice to expect that more extensive tuning could potentially reduce the observed gaps. To prevent any unintended overstatement, we have revised the manuscript to phrase this as a possibility rather than a guarantee. Importantly, none of our core findings depend on this point.
>
> In our revised manuscript, we performed a series of simulations assessing the effect of hyperparameter tuning and it is evident that a good choice of hyperparameter can improve the ADEN training (please see Appendix K). For instance, it is evident from Figures 7 and 8, the choice of hyperparameters can improve the convergence rates and final loss value.

---

> ### Author Response · Authors · 2025-11-21
>
> *"**W:** The problem formulation is presented as major contribution, but it is a straightforward extension of MEP-DA."*
>
> **A:** Thank you for your comment. While our formulation indeed builds on the resource-allocation structure that MEP-DA addresses, the introduction of *local autonomy* substantially changes both the analytical landscape and the solution methodology in ways that are not present in the original DA framework.
>
> First, when the autonomy model is not available, the problem becomes inherently a *learning* problem rather than a purely analytical clustering problem. In this setting, the autonomy term couples the assignments across clusters in a nontrivial manner, and the algorithm must learn the autonomy-aware average cost $d_{\mathrm{avg}}(x_i,y_j)$ through repeated interaction with the data points. This requires dedicated learning modules such as ADEN, along with an RL-based pipelines that do not arise in the classical MEP-DA formulation. These learning based methods introduces additional complexities (such as exploration-exploitation, network architecture, and stability considerations) not found in DA framework.
>
> Second, when the autonomy distribution depends on the cluster prescriptions $\{y_j\}$, the learning dynamics become significantly more complex: the assignment probabilities, effective association costs, and gradient updates are intertwined through the autonomy mechanism. This structural coupling between clusters is absent in standard DA, where clusters evolve largely independently apart from the shared temperature parameter. In contrast, in the autonomy-aware setting, changes to one representative $y_j$ influence the effective behavior of other clusters through the autonomy distribution, requiring a qualitatively different analysis and algorithmic treatment.
>
> Third, the nature of phase transitions changes fundamentally. In classical MEP-DA, phase transitions occur when the Hessian associated with a *single* cluster loses rank, and the directions of degeneracy are localized to that cluster. In our setting, autonomy induces strong cross-cluster coupling, so phase transitions occur when the Hessian of the *full* system of representatives loses rank. This global coupling leads to significantly more subtle and non-trivial to characterize phase-transition behavior that we demonstrate in Theorems 2 and 3.
>
> Finally, we note that we go beyond existing DA literature by providing a full convergence analysis of Algorithm 1 (Theorem 1), which — despite being central to our autonomy-aware extension — has no direct analogue in the existing DA literature. Both the convergence result and the phase-transition analysis require nontrivial arguments owing to the autonomy-induced coupling structure.
>
> For these reasons, while autonomy-aware clustering builds upon the MEP-DA perspective, the resulting problem and solution approach differ substantially in technical depth and analytical requirements.
>
> ---
> *"**W:** The writing is convoluted and hard to follow. The paper is full of jargon and undefined terms (e.g., "entity")."*
>
> **A:** Thank you for the comment. In our formulation, the term *entity* is used intentionally as a general descriptor for the items being clustered (e.g., sensors, users, or data points), and its meaning is specified in and clear from the context of the optimization problem (P1). We believe the usage is appropriate for the range of applications considered. If any specific parts seems ambiguous, we would be happy to address them directly.
>
> ---
> *"**W:** The figures/tables are low quality and hard to read."*
>
> **A:** Thank you for the comment. We have reviewed all the figures and tables in the manuscript to ensure that all of them are high resolution. In the revised manuscript, we have increased the marker size for data points and cluster representatives in the Figure 4 in Appendix F. If there are specific figures or tables you still find particularly unclear, we would be happy to refine those further.
>
> ---
> *"**W:** The conclusion is completely missing. Limitations are completely missing."*
>
> **A:** Thank you for the suggestion. We have revised the manuscript with a section on Analysis and Discussion (Section 6) that also cover the above highlighted aspects.

---

> ### Author Response · Authors · 2025-11-21
>
> *"**W:** The related work section is weak and does not contextualize the work well. Yes, other works do not use local autonomy, but how do they methodologically differ from this work? Why is local autonomy a better model?"*
>
> **A:** Thank you for the comment. Local autonomy is not just a better model, albeit a **necessary modeling requirement** when the data points are autonomous, and can possibly override the cluster prescriptions of the assignment policy. More precisely, the proposed local autonomy is not competing with the existing clustering framework, instead it extends the class of clustering problems to cover a broader class of practical problems.
>
> To the best of our knowledge, prior works do not even consider stochasticity (autonomy) at the level of the data-point; however, some of the methods do allow stochasticity at the level of assignment policy (such as Gaussian mixture models), but this is different from the stochasticity at the level of data points. Thus, in this work, we expand the paradigm of clustering to cover a much broader class of problem settings, which to our knowledge were not even addressed previously.
>
> Most prior clustering frameworks — including $k$-means, DA, EM/GMMs, spectral methods, and deep clustering approaches — assume passive data points whose assignments are fully determined by distances, likelihoods, or learned representations. Methodologically, these approaches optimize cluster representatives or latent embeddings under the assumption that the assignment mechanism is deterministic (or soft-deterministic) and fully controlled by the algorithm.
>
> As mentioned in our response to the comment on *extension of MEP-DA*, the proposed formulation introduces local autonomy, where an entity may probabilistically deviate from its prescribed assignment due to latent behavioral, operational, or environmental factors. This fundamentally alters the objective: the effective cost becomes autonomy-adjusted (P(2)) and depends not only on distances between feature vectors but also on latent transition behavior. Addressing this requires (i) new analytical updates (equations (1), (2), and Algorithm 1) together with convergence analysis (Theorem 1) and theoretical analysis of algorithmic characteristics (Theorems 2–3) to minimize the autonomy-aware free energy $P(\beta)$, and (ii) when autonomy is unknown, a model-free RL-based algorithm (Algorithm 2) to estimate autonomy-adjusted association costs—components that do not arise in standard clustering methods.
>
> Finally, while Algorithm 2 uses elements from both deterministic annealing (e.g., annealing schedules) and reinforcement learning, it extends them to cover this new and broader class of problem setting. For instance, in classical deep RL, the value function learned through interaction with the environment is used to compute an optimal policy. In contrast, in our framework the learned autonomy-aware average association cost (analogous to a $Q$-function as indicated in our earlier response) is further used to optimize the cluster representatives $y_j$, an additional optimization layer not present in standard RL pipelines.
>
> ---
> *"**W:** ... the first loop in Algorithm 2 performs the following steps: Assign new distances to all data points based on the current cluster centroids. Sample assignments using the currently learned soft assignment distribution. Observe actual assignment outcomes using the autonomy distribution and update empirical costs. Update an exponential moving average of the costs. Update the neural network parameters  using supervised learning to minimize the squared error between the predicted costs and the empirical costs. This is not Reinforcement Learning. There is no action space, state space, transition dynamics, or reward function. There is also no temporal component..."*
>
> **A:** Thank you for seeking clarification on this point. We have responded to this in your questions on *what MDP is Algorithm 2 solving* and *why Algorithm 2 is using Reinforcement Learning* in the above thread. In the revised manuscript, we have formalized the unit horizon MDP viewpoint of autonomy-aware clustering at the beginning of Section 4 (marked in blue ink). We also added a sub-section to the Section 6, that draws parallel between our proposed Algorithm 2, and standard RL pipelines. We have provided explicit equivalence between the average cost of association $d_{\text{avg}}(x_i,y_j)$ and the $Q$-function in RL literature, that motivates approach used in Algorithm 2 to estimate the former. Unit-horizon (one-step) MDP formulations are widely used in other contexts too; for example in contextual bandits, meta-learning, AutoML (algorithm selection), and architecture search — whenever the task involves a single state–action–reward interaction with unknown stochastic effects. Such problems naturally benefit from RL tools/pipelines.

---

> > ### Comment · Reviewer_2opo · 2025-11-25
> >
> > >  Why would I model the decentralized sensing problem with local autonomy instead of modeling interference/noise directly?"
> >
> > Good illustrative example. Thanks for the clarification. This is crucial, as the authors propose a new problem formulation.
> >
> > >  What baseline is used in the results? MEP-DA w/o the local autonomy distribution?
> >
> > Thanks for the clarification.
> >
> > > What is the model-based baseline (ground truth)? Algorithm 1 with the true autonomy distribution? How is it possible to outperform the ground truth?
> >
> > I would strongly encourage the authors to ditch the “ground truth” terminology because it is confusing. To me, the ground truth would be the actual next assignment a point takes through its local autonomy. What the authors are referring to is a model-based baseline, which has access to the autonomy distributions. Without this context (only on page 9), the claim in the abstract is easily misunderstood.
> >
> > >  Where are the numerical results for the UDT19 dataset?
> >
> > Thank you for clarifying this. Can the authors please provide a table (such as table 1) with the numerical results for all combinations? It is enough if this is in the Appendix, but it should be provided.
> >
> > >  What are limitations of the proposed method? When would I use it over other methods and vice versa?
> >
> > The authors describe the following limitations:
> >
> > When local autonomy is not given, the algorithm incurs computational overhead.
> > The model may fail to learn the autonomy model if it is not in the function class that the network can approximate.
> >
> > Of this, only the second point is a limitation specific to this method. This reads incomplete, as for example, their algorithm does not consider temporal dependencies $\gamma=0$. For example, it is also not discussed how their algorithm scales to high-dimensional observations or large datasets. Correctly specifying limitations is a crucial part of a paper, which I feel has still not been fully addressed.
> >
> > > The authors do not justify why local autonomy is a better model for decentralized sensing than other approaches.
> >
> > Thanks, see answer 1. Has been addressed
> >
> > > The experimental setup has major flaws: the problems are contrived and artificial, the baselines are not clearly defined.
> >
> > I thank the authors for clarifying the baselines (see above). However, I still think that the datasets provided are too small and too toy-like to evaluate how this algorithm would perform in real-world settings, e.g., real-world food delivery management. Can the authors elaborate more on the specifics of the UDT19 London Traffic dataset? How many samples does it have? How many dimensions does a sample have? I’ve checked Loder et al., 2019, but could not find this.
> >
> > > The claim "These gaps can be further reduced through standard hyperparameter tuning and extended training" is not supported by any evidence.
> >
> > Thanks for clarification and the additional experiments. This has been addressed
> >
> > > The problem formulation is presented as major contribution, but it is a straightforward extension of MEP-DA.
> >
> > Thanks for the clarification. This helps substantially. I have one question related to the reply: “This requires dedicated learning modules such as ADEN, along with an RL-based pipelines that do not arise in the classical MEP-DA formulation.” Is “RL” the only possible framework for tackling autonomy-aware clustering, or are there other possible approaches that could be used?
> >
> > > Writing, figures, conclusion, limitations
> >
> > I still think the paper could do a better job of introducing terminology and notation in some places, e.g., that an entity is just a data point, or consistent notation for the assignment policy ($\pi$ vs $\mu$). However, the most important criticism (conclusion, limitations) has been addressed.
> >
> > > The related work section is weak and does not contextualise the work well. Yes, other works do not use local autonomy, but how do they methodologically differ from this work? Why is local autonomy a better model?
> >
> > Thank you for better contextualising your work. This has been addressed.

---

> > > ### Comment · Reviewer_2opo · 2025-11-25
> > >
> > > ## Regarding the "RL" problem formulation and MDP
> > >
> > > First, I want to thank the authors for specifying the MDP that their algorithm is solving. This is crucial, as the RL framework is the core aspect of their methodological contribution. However, I still have major issues with their formalization and would argue that the author's claim of using "RL" is, in fact, an overclaim. Let's review the author's MDP formalization:
> > >
> > > - **State space (dataset with cluster representatives):** $\mathcal S = \{x\_i\}\_{i=1}^N \times \{y\_j\}\_{j=1}^K$
> > > - **Action space (cluster assignments):** $\{C\_j\}\_{j=1}^K$
> > > - **Reward function ($c : \mathcal S \times \mathcal A \times \mathcal A$):** $c(x\_i, C\_j, y\_k) := d(x\_i, y\_k)$
> > > - **Transition function ($p(s' \mid s, a)$):** $k \sim p(\cdot \mid j, i)$
> > > - **Initial state distribution:** "We begin by sampling a data point, analogous to sampling an initial state"
> > > - **Discount factor:** $\gamma = 0$
> > >
> > > There are multiple errors and ill-defined functions in this formulation, which I will demonstrate below:
> > >
> > > - The state and action space definitions are OK.
> > > - The reward function simplifies to depending solely on a state $s \in \mathcal S$, as defined above, which is an acceptable and common simplification.
> > > - The transition function contradicts the stated state space definition. The authors define states as (data point, cluster assignment) pairs, but their transition function specifies only a distribution over assignments, omitting the data point component. This makes the transition function mathematically incomplete.
> > > - The initial state violates the author's own state space definition because it ignores the cluster representative dimension of the state space.
> > > - Setting the discount factor to zero makes the author's formulation a non-RL problem, as the temporal dimension is crucial to RL. In particular, the authors define Equation (5) with an infinite sum, which is pointless given $\gamma = 0$. Additionally, in Equation (6), the authors define a pointless formulation of the Bellman Equation $Q(s,a)$ given that $\gamma = 0$. There is no need for a value function without delayed rewards or credit assignment, both of which are nonexistent with $\gamma = 0$.
> > >
> > > **To summarize:** the authors employ ill-defined notation to create an RL formulation of a problem that does not exist. In my opinion, this presents a serious methodological flaw that warrants rejection on its own.

---

> > > > ### Author Response · Authors · 2025-11-29
> > > > **Clarification on the RL problem formulation and MDP**
> > > >
> > > > Thank you for the follow-up comments on the RL formulation. We believe the following clarifications, specifically, **on the definition of the state space $\mathcal{S}$** and **why $\gamma=0$ does not invalidate our RL viewpoint** will address all the concerns.
> > > >
> > > > ---
> > > > *" **Comment:** State space (dataset with cluster representatives): $\mathcal{S}$ = $\\{x_i\\}^N_{i=1} $ $\times \\{y_j\\}_{j=1}^K$* "
> > > >
> > > > **A:** Thank you for comment. The above is an incorrect interpretation of the unit horizon MDP stated in the beginning of Section 4. The state is **NOT** $(x_i,y_j)$ pair; i.e., the state space $\mathcal{S}$ is **NOT the cross product** $\\{x_i\\}^N_{i=1}\times \\{y_j\\}_{j=1}^K$.
> > > >
> > > > We re-iterate the definition of the state space here: $\mathcal{S}=\\{x_1,...,x_N,y_1,...,y_K\\}$.
> > > >
> > > > Note that the MDP we have defined is a unit horizon, i.e., there is only one transition from initial state to the final state. The initial states $s_0$ are sampled from the distribution $\bar{\rho}(\cdot)$, where $\bar{\rho}(x_i)=\rho(i)$ for all $1\leq i\leq N$, and $0$ otherwise. This implies that the possible initial states can be elements only from the set $\\{x_1,...,x_N\\}$.
> > > >
> > > > From the definitions of the action space, and the autonomy model $p(k|j,i)$, the final state can only be from the set $\\{y_1,...,y_K\\}$.
> > > >
> > > > To avoid any possible ambiguity, we have revised the MDP formalism in Section 4 and added further clarification regarding the definition of the state space. We believe this resolves the source of the misunderstanding and also addresses the subsequent questions that were based on this interpretation. For completeness, we provide additional details below.
> > > >
> > > > ---
> > > > *" **Comment:** The transition function contradicts the stated state space definition. The authors define states as (data point, cluster assignment) pairs, but their transition function specifies only a distribution over assignments, omitting the data point component. This makes the transition function mathematically incomplete."*
> > > >
> > > > **A:** *We hope that the clarification on the definition of the state space obviates this concern.* Note that the transition function **does not** contradicts the state space definition provided in Section 4. The state space is $\mathcal{S}:=\\{x_1,...,x_N,y_1,...,y_K\\}$. Further, the action space $\mathcal{A}=\\{C_1,...,C_K\\}$. Under these $\mathcal{S}$ and $\mathcal{A}$, the transition kernel $p(s'|a,s)$ would be $p(y_k|C_j,x_i)$. However, for notational simplicity and to match the autonomy model described in earlier sections of the manuscript we denote this as $p(k|j,i)$.
> > > >
> > > > ---
> > > > *" **Comment:** The initial state violates the author's own state space definition because it ignores the cluster representative dimension of the state space."*
> > > >
> > > > **A:** We hope the above clarification regarding the state space clarifies this concern.
> > > >
> > > > ---

---

> > > > > ### Author Response · Authors · 2025-11-29
> > > > >
> > > > > *" **Comment:** Setting the discount factor to zero makes the author's formulation a non-RL problem, as the temporal dimension is crucial to RL. In particular, the authors define Equation (5) with an infinite sum, which is pointless given $\gamma=0$. Additionally, in Equation (6), the authors define a pointless formulation of the Bellman Equation $Q(s,a)$ given that $\gamma=0$.  There is no need for a value function without delayed rewards or credit assignment, both of which are nonexistent with $\gamma=0$."*
> > > > >
> > > > > **A:** Thank you for this follow-up comment. This is important and we clarify in detail. We emphasize that setting the discount factor to $\gamma = 0$ **does not invalidate our RL viewpoint**. The connection we draw between autonomy-aware clustering (AAC) and a **unit-horizon MDP** is intentional and enables us to **leverage the standard RL pipeline** and arrive at the Algorithm 2 in the absence of an explicit autonomy model (model-free RL). This idea of viewing AAC as an MDP is one of the  novelties of our work.
> > > > >
> > > > > Importantly, **RL methods remain well defined for $\gamma = 0$**. In *Reinforcement Learning: An Introduction (Sutton \& Barto, 2nd ed.*, Chapter 3 explicitly allows $0\leq \gamma \leq 1$ (Eq. 3.8), with $\gamma = 0$ corresponding to a *myopic* agent. All standard RL mechanisms and structures - exploration strategies, TD learning, policy-gradient methods, and value-function approximation techniques - remain applicable in this setting.
> > > > >
> > > > > Moreover, in Chapter 2 of the same book, the foundational examples motivating RL - multi-armed and contextual bandits - are *unit-horizon MDPs* with $\gamma = 0$ and value $V_\pi(s) = \mathbb{E}[r(s,a)]$, which is precisely the structure we adopt. The book uses these formulations to introduce key RL concepts such as $\epsilon$-greedy action (Sec. 2.2) and incremental value-function updates (Eq. 2.4).
> > > > >
> > > > > In another RL text, *Szepesvári, Csaba. Algorithms for reinforcement learning. Springer nature, 2022*, Chapter 2 formalizes the MDP structure with $0\leq \gamma\leq 1$. In fact, in their exposition of the least-squares temporal difference (LSTD) method - a widely used RL algorithm - they use the case $\gamma = 0$ in their discussion on choice of value function approximator (see page 42); and subsequently tailor discussion around the same loss function as in (7) in our paper. This shows that **standard, well-established RL algorithms explicitly include and remain valid in the $\gamma=0$ regime**. Therefore, our formulation is fully consistent with canonical RL theory and practice, and our use of the RL framework for the AAC problem is both mathematically sound and methodologically appropriate.
> > > > >
> > > > > Again, the connection we establish between autonomy-aware clustering and a unit-horizon MDP is deliberate, and it allows us to *leverage (and extend) the standard RL pipeline* to learn the clustering solutions in the absence of an autonomy model. To make this correspondence explicit, we restate below the core components of a conventional RL workflow together with their AAC analogues:
> > > > >
> > > > > 1) Sample an initial state $s_0$ (AAC: a data point $i \sim \rho(\cdot )$).
> > > > >
> > > > > 2) Select an $\epsilon$-greedy action $a_0$ (AAC: a cluster $j$ sampled under the $\epsilon$-greedy softmax policy $\pi(\cdot |i)$).
> > > > >
> > > > > 3) The environment transitions to $s_1 \sim p(\cdot|a_0,s_0)$ (AAC: the data point associates to $k \sim p(\cdot \mid j,i)$ under autonomy).
> > > > >
> > > > > 4) Receive instantaneous cost $c(s,a,s')$ (AAC: instantaneous cost is $d(x_i,y_k)$).
> > > > >
> > > > > 5) Update the value estimate using $Q$-learning or deep-RL-style regression (AAC: the analogous update of $d_\theta(x_i,y_j)$ (estimator of $d_{\text{avg}}(x_i,y_j)$, as described in Section 4).
> > > > >
> > > > > Thus, while $\gamma > 0$ is needed for long-horizon problems, it is not relevant to our single-step setting.
> > > > >
> > > > > Finally, defining the infinite-horizon return (Eq. 5) and the Bellman equation (Eq. 6) in the paper is **NOT** pointless; it completes the earlier requested formalization. In fact, building up on (5) and (6), we show that under $\gamma = 0$, the Bellman equation in (6) reduces to
> > > > >
> > > > > $$Q(s,a) = \mathbb{E}\big[d(x_i,y_k)\big] = d_{\mathrm{avg}}(x_i,y_j),$$
> > > > >
> > > > > which establishes the equivalence between the AAC average cost $d_{\mathrm{avg}}(x_i,y_j)$ and the $Q$-function. This equivalence motivates our use of deep-RL-style updates to learn $d_{\text{avg}}(x_i,y_j)$ (estimated as $d_{\theta}(x_i,y_j)$) and clarifies how the optimal policy in soft $Q$-learning reduces to our assignments $\pi_Y^{\beta}(j|i)$ in Eq. (1); further reinforcing the equivalence claim.
> > > > >
> > > > > **In summary**, our formulation follows standard RL principles. Unit-horizon MDPs with $\gamma = 0$ are a well-defined part of the RL framework and study considered in standard RL texts. The AAC problem fits this structure, and our use of RL framework and pipeline is justified. Consequently, *the assertion that our use of RL is an overclaim is not supported when viewed in light of standard RL formalism.*
> > > > >
> > > > > ---

---

> > > ### Author Response · Authors · 2025-11-29
> > >
> > > *" **Comment:** I would strongly encourage the authors to ditch the “ground truth” terminology because it is confusing. To me, the ground truth would be the actual next assignment a point takes through its local autonomy. What the authors are referring to is a model-based baseline, which has access to the autonomy distributions. Without this context (only on page 9), the claim in the abstract is easily misunderstood."*
> > >
> > > **A:** Thank you for the comment. We have revised the manuscript, to provide more clarity around the term *ground truth*. We have updated the Abstract and Introduction to explicitly state that *ground truth* refers to the model-based baseline obtained when the autonomy distribution is known (highlighted in blue ink in the revised manuscript). We believe that this clarification makes the intended meaning precise and avoids any possible misunderstanding, while allowing us to maintain consistent terminology throughout the work.
> > >
> > > ---
> > > *" **Comment:** Can the authors please provide a table (such as table 1) with the numerical results for all combinations? It is enough if this is in the Appendix, but it should be provided."*
> > >
> > > **A:** Thank you for the suggestion. We have added the Table 3 to the Appendix I summarizing the details.
> > >
> > > ---
> > > *" **Comment:** (When local autonomy is not given, the algorithm incurs computational overhead. The model may fail to learn the autonomy model if it is not in the function class that the network can approximate.) Of this, only the second point is a limitation specific to this method. This reads incomplete, as for example, their algorithm does not consider temporal dependencies. For example, it is also not discussed how their algorithm scales to high-dimensional observations or large datasets. Correctly specifying limitations is a crucial part of a paper, which I feel has still not been fully addressed."*
> > >
> > > **A:** Thank you for the comment. We would like to clarify that the absence of temporal dependencies in our formulation is **not** a limitation, but rather a deliberate **feature** of the autonomy-aware clustering (AAC) framework. AAC is viewed as a **unit-horizon MDP**, in which each interaction consists of a single prescription–deviation step with no temporal evolution. Within this viewpoint, temporal dependencies cannot arise by design, and deliberately introducing them would fundamentally completely change the problem away from the original task of clustering.
> > >
> > > Regarding **method-specific limitations**: the primary challenge remains learning the autonomy-adjusted cost when the autonomy model is unknown, especially if the true autonomy mechanism lies outside the function class that ADEN can approximate. We further note that the current framework does not directly extend to clustering methods based on pairwise distances or connectivity structure (e.g., density-based or graph-based clustering), and that incorporating robustness to noise in the autonomy mechanism or in the feature vectors is a non-trivial extension closely related to robust RL and existing transformer based techniques. These aspects constitute meaningful directions for future work, and have been mentioned as future directions in the Section 6 (Analysis and Discussion)
> > >
> > > The challenges such as very large datasets or high-dimensional observations do exist, but are **not specific** to our method; these are shared with most clustering and RL-based approaches. Standard tools such as dimensionality reduction, sampling, or distributed computation can be incorporated into our framework when needed.
> > >
> > > ---
> > > *" **Comment:** I still think the paper could do a better job of introducing terminology and notation in some places, e.g., that an entity is just a data point, or consistent notation for the assignment policy ($\pi$ vs $\mu$). However, the most important criticism (conclusion, limitations) has been addressed."*
> > >
> > > **A:** Thank you for the comment and for acknowledging the concluding section added to the revised manuscript. We use the term *entity* as a generic abstraction because our work addresses a broad class of optimization problem that occurs in several different application areas. The distinction between hard associations ($\mu(j|i)$) and soft associations ($\pi(j|i)$) is already defined in the statements of (P(1)), (P(2)), and (P($\beta$)); however, we agree that consolidating notation can further aid readability. To this end, we have added a brief notation summary to the Appendix L to make these definitions easier to reference. We have also added this information at the beginning of the Section 3 (Problem Formulation) of the revised manuscript (highlighted in blue ink).

---

> > > ### Author Response · Authors · 2025-11-29
> > >
> > > *" **Comment:** I thank the authors for clarifying the baselines (see above). However, I still think that the datasets provided are too small and too toy-like to evaluate how this algorithm would perform in real-world settings, e.g., real-world food delivery management. Can the authors elaborate more on the specifics of the UDT19 London Traffic dataset? How many samples does it have? How many dimensions does a sample have? I’ve checked Loder et al., 2019, but could not find this."*
> > >
> > > **A:** Thank you for the comment. While a major contribution of our work lies in proposing a general autonomy-aware clustering framework---including its formulation, theoretical guarantees, and learning algorithms---we also demonstrate its practical relevance on a real, city-scale dataset widely used in the transportation literature. In particular, we use the UDT19 London Traffic dataset to design autonomy-aware decentralized sensing networks.
> > >
> > > The UDT19 London Traffic dataset is curated from Transport for London (TfL) traffic sensor feeds and processed by Loder et al. (2019). It contains **5719** traffic sensors distributed across Greater London, each providing multiple spatio-temporal attributes such as geographic coordinates, average speed, flow, and occupancy. For our clustering and UAV-placement experiments, we use the **GPS coordinates** (relevant features for sensing network design) of these sensors (2-dimensional features per sample), and we synthetically vary autonomy levels (mimicking underlying reasons such as congestion, obstruction, low battery) to evaluate performance under different deviation patterns. Although each sample is low-dimensional, the dataset is far from toy-like: it consists of **a few thousand entities** with **substantial spatial heterogeneity** across the London road network (as illustrated in Figure 2), making it a realistic and challenging clustering instance.
> > >
> > > Thank you for highlighting the missing information. We have added the details of the dataset in the Appendix I and made a note of the same in Section 5.
> > >
> > > We also emphasize that challenges such as very high-dimensional features or extremely large datasets are not specific to our autonomy-aware formulation; they are common and shared with clustering and RL-based methods. Standard approaches such as PCA, dimensionality reduction, mini-batching, or parallelization can be incorporated into our framework when needed. The unique complexity introduced by our setting arises from the autonomy term $p(k|j,i)$, especially when it is unknown and depends on the representatives $\\{y_j\\}$ and entity features $\\{x_i\\}$. Our experiments show that Algorithm 2 handles this complexity effectively across both datasets considered in our simulations. Please see details in Section 5 and Appendix I.
> > >
> > > ---
> > > *" **Comment:** I have one question related to the reply: “This requires dedicated learning modules such as ADEN, along with an RL-based pipelines that do not arise in the classical MEP-DA formulation.” Is “RL” the only possible framework for tackling autonomy-aware clustering, or are there other possible approaches that could be used?"*
> > >
> > > **A:** Thank you for the question. We do not claim that RL is the only framework for addressing autonomy-aware clustering. Rather, RL provides a principled and practical approach for the *unknown-autonomy* setting, where the autonomy distribution influences the effective assignments but is not available in closed form. In such cases, the problem naturally becomes one of learning the autonomy-adjusted association costs through interaction, and (unit-horizon MDP viewpoint and subsequently) RL offers a well-established structure to do this.
> > >
> > > Other approaches could indeed be constructed by first learning or estimating the autonomy model - for example, by fitting $p(k|j,i)$ from observed assignment deviations - and then applying deterministic annealing or another clustering method ($k$-means) directly considering the estimated $d_{\text{avg}}(x_i,y_j)$ in the cost function. However, such two-stage procedures still require (i) collecting sufficient data to reliably estimate the autonomy distribution, and (ii) handling cases where autonomy depends on the evolving cluster representatives. Our RL-based approach integrates these steps into a single framework, learning the autonomy-adjusted costs directly and updating the cluster representatives accordingly, without requiring an explicit intermediate estimation of $p(k|j,i)$

---

### Author Response · Authors · 2025-12-03
**Authors' Closing Remarks**

We thank all the reviewers for their time and feedback. We sincerely appreciate the thoughtful questions, which helped us clarify several aspects of our framework. We have incorporated all suggested clarifications, added new experiments (MovieLens and stability analysis), incorporated a discussion section, and included the MDP formalization for improved clarity. We hope the updated manuscript addresses all concerns and demonstrates the contributions and generality of autonomy-aware clustering.

---

### Meta-Review · Area_Chair_KXD6 · 2026-01-05

**Summary:**

This work proposes a framework for autonomy-aware clustering. The key is to allow the nodes to deviate from their assigned cluster membership according to an autonomy distribution.

Strengths:

The reviewers acknowledge that the setting of autonomy-aware clustering could be an interesting setting. Some praised the recasting of the problem as MDP. Strong empirical results were also noticed. The reviewers also commended the fact that theoretical justifications were given.

Weaknesses:

Reviewers are also critical about the autonomy-aware model’s necessity (against existing probabilistic soft clustering models), the clarity of the MDP, and the claim that the method is reinforcement learning (RL). Multiple reviewers raised concerns on the complexity of transformer based ADEN method. A reviewer is also concerned about the applicability in real world scenarios as the method is only evaluated by one single real world data.

**Reviewer Concerns:**

The major concerns of Reviewer 2opo regarding relations to RL were not fully resolved. The authors have added explanations in the manuscript and clarified that gamma=0 in the RL formulation. But the reviewer was not convinced that this is a meaningful RL formulation.

Reviewer no6m's concerns were on differences with probabilistic clustering and complexity. These were explained in the rebuttal. The reviewer leans to score between 6-8.

Reviewer ggqe has concerns on the limited real world experiment scope. The reviewer expressed that this was not sufficiently demonstrated. The authors further provided a MovieLens experiment. But as MovieLens is a standard clustering problem, how it is relevant to autonomy-aware clustering is not entirely unclear.

**Reviewer Scores:**

Reviewer 2opo (rating 0) had strong objection to positioning the work as RL. The push on clarifying the MDP and the RL formulation was necessary. After the discussion, the MDP and RL with gamma=0 were added. The rebuttal argued that setting gamma=0 appeared in classical RL textbooks. This is indeed true, but also limited its relation to true power of RL. The AC believes that Reviewer 2opo would not lift the score to acceptance level.

Reviewer ggqe's (rating 4) concern on the limited scope of experiments was responded by including a MovieLens experiment. However, given that MovieLens is a classical dataset, how much the concept of autonomy-aware clustering would help perhaps needs more understanding. The reviewer may not lift their score.

---

### Decision · Program_Chairs · 2026-01-26

Reject